# SPARSE MoEs MEET EFFICIENT ENSEMBLES

## ABSTRACT

Machine learning models based on the aggregated outputs of submodels, either at the activation or prediction levels, lead to strong performance. We study the interplay of two popular classes of such models: ensembles of neural networks and sparse mixture of experts (sparse MoEs). First, we show that the two approaches have complementary features whose combination is beneficial. Then, we present *partitioned batch ensembles*, an efficient ensemble of sparse MoEs that takes the best of both classes of models. Extensive experiments on fine-tuned vision Transformers demonstrate the accuracy, log-likelihood, few-shot learning, robustness, and uncertainty improvements of our approach over several challenging baselines. Partitioned batch ensembles not only scale to models with up to 2.7B parameters, but also provide larger performance gains for larger models.

## 1 INTRODUCTION

Neural networks typically use all their parameters to process an input. Sustaining the growth of such models—reaching today up to 100B parameters (Brown et al., 2020)—is challenging, e.g., due to their high computational and environmental costs (Strubell et al., 2019; Patterson et al., 2021). In this context, sparse mixtures of experts (sparse MoEs) employ *conditional computation* (Bengio et al., 2013) to combine multiple submodels (experts) and route examples to certain experts (Shazeer et al., 2017; Lepikhin et al., 2021; Fedus et al., 2021; Riquelme et al., 2021; Yang et al., 2021). Conditional computation can decouple the growth of the number of parameters from the training and inference costs, by only activating a subset of the overall model in an input-dependent fashion.

Paralleling this trend, the deployment of ML systems in safety-critical fields, e.g., medical diagnosis (Dusenberry et al., 2020b) and self-driving cars (Levinson et al., 2011), has motivated the development of reliable deep learning, e.g., for *calibrated and robust predictions* (Ovadia et al., 2019). Among the approaches, ensembles of neural networks have remarkable performance for calibration and accuracy under dataset shifts (Ovadia et al., 2019). These methods improve reliability by aggregating the predictions of individual submodels (ensemble members).

While sharing conceptual similarities, these two classes of models—MoEs and ensembles—have different properties. Sparse MoEs adaptively combine their experts depending on the inputs, and the combination generally happens at internal activation levels. Ensembles typically combine several models in a static way and at the prediction level. Moreover, these two classes of models tend to be benchmarked on different tasks: few-shot classification for MoEs (Riquelme et al., 2021) and uncertainty-related evaluation for ensembles (Ovadia et al., 2019; Gustafsson et al., 2020).

CONTRIBUTIONS: In this paper, we study the interplay between sparse MoEs and ensembles. This results in two sets of contributions.

**Contribution 1: Complementarity of MoEs and ensembles.** We show that sparse MoEs and ensembles have complementary features and benefit from each other. Specifically:
• The adaptive computation in sparse MoEs and the static combination in ensembles are orthogonal, with additive benefits when associated together. Their association results in insightful performance versus FLOPs trade-offs while varying the ensemble size and sparsity.
• In sparse MoEs, combining models at the prediction level leads to improved uncertainty estimates.
• Over tasks where either sparse MoEs or ensembles are known to perform well, naive—and computationally expensive—ensembles of MoEs provide the best predictive performance. Our benchmarking effort includes the first evaluation of sparse MoEs on uncertainty-related vision tasks, which builds upon the empirical work of Riquelme et al. (2021).

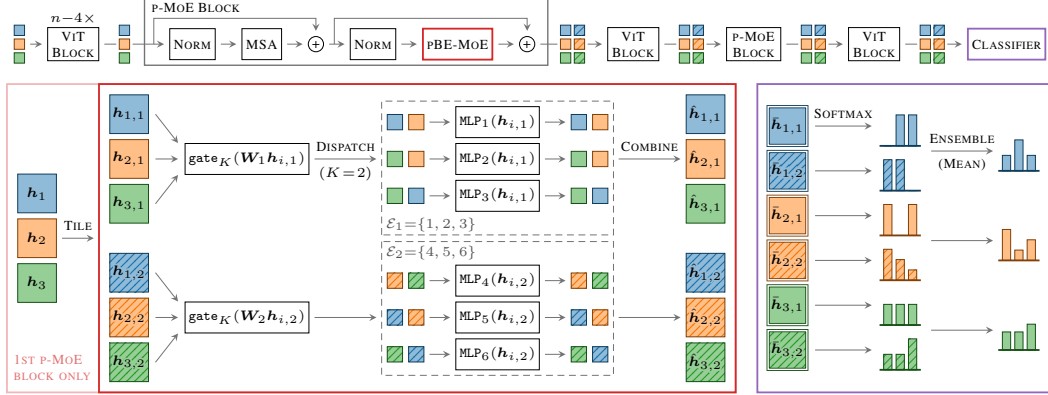

Figure 1: End-to-end overview of pBE with $E=6$ experts, $M=2$ partitions, sparsity of $K=2$, and a "last-2" configuration. **Top**: pBE contains a sequence of ViT blocks, followed by alternating p-MoE and ViT blocks. Images are split into patches whose linear embeddings are processed by each block. Here, we show 1 embedding for each of three images (🟦, 🟧, 🟩). In practice, we have many embeddings including a special `class` embedding, as in Dosovitskiy et al. (2021). **Bottom left**: in a p-MoE block, we replace the ViT block's MLP with parallel partitioned expert MLPs, see (3). Embeddings are tiled (▨) in the first p-MoE block only. The effect of the routing weights is not depicted. **Bottom right**: the classifier uses the `class` embeddings (▣) to make predictions. Ensembling of the predictions for the embeddings and corresponding tiled versions happens only at test time.

**Contribution 2: Partitioned batch ensembles.** We propose partitioned batch ensembles (pBE), see Figure 1, an efficient ensemble approach tailored to sparse MoEs. Specifically:
• pBE improves over sparse MoEs across metrics including few-shot performance, likelihood and calibration error. pBE matches the performance of deep ensembles for 30%-43% fewer FLOPs.
• pBE gracefully scales up to vision Transformers with up to 2.7B parameters.
• pBE is both simple (requiring only minor implementation changes) and convenient because standard sparse-MoE checkpoints can be used directly to initialize pBEs for fine-tuning.

## 2 PRELIMINARIES

We focus on classification tasks where we learn classifiers of the form $f(\boldsymbol{x}; \boldsymbol{\theta})$ based on some training data $\mathcal{D} = \{(\boldsymbol{x}_n, y_n)\}_{n=1}^{N}$. A pair $(\boldsymbol{x}_n, y_n)$ corresponds to an input $\boldsymbol{x}_n \in \mathbb{R}^P$ together with its label $y_n \in \{1, \ldots, C\}$ belonging to one of the $C$ classes. The model $f(\cdot; \boldsymbol{\theta})$ is parametrized by $\boldsymbol{\theta}$ and outputs a $C$-dimensional probability vector. We use $\circ$ to refer to matrix element-wise product.

### 2.1 VISION TRANSFORMERS AND SPARSE MOES

**Vision Transformers.** Throughout the paper, we choose the model $f$ to be a vision Transformer (ViT) (Dosovitskiy et al., 2021). ViT is growing in popularity for vision, especially in transfer-learning settings where it was shown to outperform convolutional networks while requiring fewer pre-training resources. ViT operates at the level of patches. An input image is split into equal-sized patches (e.g., $32 \times 32$, $16 \times 16$, or $14 \times 14$ pixels) whose resulting sequence is (linearly) embedded and processed by a Transformer (Vaswani et al., 2017). The operations in the Transformer then mostly consist of a succession of multiheaded self-attention (MSA) and MLP layers. ViT is defined at different scales (Dosovitskiy et al., 2021): S(mall), B(ase), L(arge) and H(uge); see specifications in Appendix A. For example, ViT-L/16 stands for a large ViT with patch size $16 \times 16$.

**Sparse MoEs and V-MoEs.** The main feature of sparsely-gated mixture-of-experts models (sparse MoEs) lies in the joint use of sparsity and *conditional computation* (Bengio et al., 2013). In those models, we only activate a small subset of the network parameters *for a given input*, which allows the total number of parameters $\boldsymbol{\theta}$ to grow while keeping the overall computational cost constant. The subparts of the network that are activated on a per-input fashion are known as *experts*.

Central to our study, Riquelme et al. (2021) recently extended ViT to sparse MoEs. Their extension, referred to as V-MoE, follows the successful applications of sparse models in NLP (Shazeer et al.,

2017). Riquelme et al. (2021) show that V-MoEs dominate their "dense" ViT counterparts on a variety of tasks for the same computational cost. In the specific case of V-MoEs, the experts are placed in the MLP layers of the Transformer, a design choice reminiscent of Lepikhin et al. (2021) in NLP. Given the input $\boldsymbol{h} \in \mathbb{R}^D$ of such a layer, the output of a single $\texttt{MLP}(\boldsymbol{h})$ is replaced by

$$\texttt{MoE}(\boldsymbol{h}) = \sum_{e=1}^{E} g_e(\boldsymbol{h}) \cdot \texttt{MLP}_e(\boldsymbol{h}) \quad \text{with} \quad \{g_e(\boldsymbol{h})\}_{e=1}^{E} = \texttt{top}_K(\texttt{softmax}(\boldsymbol{W}\boldsymbol{h})), \quad (1)$$

where the *routing* weights $\{g_e(\boldsymbol{h})\}_{e=1}^{E}$ combine the outputs of the $E$ different experts $\{\texttt{MLP}_e\}_{e=1}^{E}$. To sparsely select the experts, $\texttt{top}_K$ sets all but the $K$ largest weights to zero. The router parameters $\boldsymbol{W} \in \mathbb{R}^{E \times D}$ are trained together with the rest of the network parameters. We call the layer defined by (1) an MoE layer. In practice, the weights $\{g_e(\boldsymbol{h})\}_{e=1}^{E}$ are obtained by a noisy version of the routing function $\texttt{top}_K(\texttt{softmax}(\boldsymbol{W}\boldsymbol{h} + \sigma\varepsilon))$ with $\varepsilon \sim \mathcal{N}(\boldsymbol{0}, \boldsymbol{I})$, which mitigates the non-differentiability of $\texttt{top}_K$ when combined with auxiliary losses (Shazeer et al., 2017). We use the shorthand $\texttt{gate}_K(\boldsymbol{z}) = \texttt{top}_K(\texttt{softmax}(\boldsymbol{z} + \sigma\varepsilon))$ and take $\sigma = 1/E$ (Riquelme et al., 2021).

In this paper, we take the "last-$n$" setting of Riquelme et al. (2021) wherein only a few MoE layers are placed at the end of the Transformer ($n = 2$ for the {S, B, L} scale and $n = 5$ for H). This setting retains most of the performance gains of V-MoEs while greatly reducing the training cost.

## 2.2 ENSEMBLES OF NEURAL NETWORKS

**Ensembles.** We build on the idea of ensembles, which is a known scheme to improve the performance of individual models (Hansen & Salamon, 1990; Geman et al., 1992; Krogh & Vedelsby, 1995; Opitz & Maclin, 1999; Dietterich, 2000; Lakshminarayanan et al., 2017). Formally, we assume a set of $M$ model parameters $\Theta = \{\boldsymbol{\theta}_m\}_{m=1}^{M}$. We refer to $M$ as the *ensemble size*. Prediction proceeds by computing $\frac{1}{M}\sum_{\boldsymbol{\theta}\in\Theta} f(\boldsymbol{x}; \boldsymbol{\theta})$, i.e., the average probability vector over the $M$ models. To assess the diversity of the predictions in the ensemble, we will use the KL divergence $D_{\text{KL}}(f(\boldsymbol{x}_t; \boldsymbol{\theta}_m) \| f(\boldsymbol{x}_t; \boldsymbol{\theta}_{m'}))$ between the predictive distributions $f(\boldsymbol{x}_t; \boldsymbol{\theta}_m)$ and $f(\boldsymbol{x}_t; \boldsymbol{\theta}_{m'})$, averaged over the test input $\boldsymbol{x}_t$ and all pairs $(m, m')$ of ensemble members.

**Batch ensembles.** Ensembles differ in the way $\Theta$ is defined. Central to our study, *batch ensembles* (BE) (Wen et al., 2019) build the ensemble as a collection of submodels, with the parameters $\boldsymbol{\theta}_m \in \Theta$ sharing components. This mitigates the computational and memory cost of ensembling, enabling one to improve the performance of the original model at little extra cost. We focus on the example of a single dense layer in $f$ with parameters $\boldsymbol{U} \in \mathbb{R}^{D \times L}$, assuming no bias. BE defines $M$ copies of parameters $\{\boldsymbol{U}_m\}_{m=1}^{M}$ so that $\boldsymbol{U}_m = \boldsymbol{U} \circ (\boldsymbol{r}_m \boldsymbol{s}_m^\top)$, where $\boldsymbol{U}$ are parameters shared across ensemble members, and $\boldsymbol{r}_m$ and $\boldsymbol{s}_m$ are separate $D$- and $L$-dimensional vectors for ensemble member $m$. Given an input, the BE produces $M$ outputs, and the $M$ outputs are averaged after applying all layers. Despite the simple rank-1 parametrization, BE leads to remarkable predictive performance and robustness (Wen et al., 2019). Notably, the efficiency of BE relies on tiling the inputs to simultaneously predict with the $M$ ensemble members, an insight that we also exploit.

## 2.3 UPSTREAM PRE-TRAINING AND DOWNSTREAM FINE-TUNING

Large-scale Transformers pre-trained on *upstream* tasks were shown to have strong performance when fine-tuned on smaller *downstream* tasks, across a variety of domains (Devlin et al., 2018; Dosovitskiy et al., 2021; Radford et al., 2021). We follow this paradigm and focus on the fine-tuning of models pre-trained on JFT-300M (Sun et al., 2017), similar to Riquelme et al. (2021). We will thus assume the availability of already pre-trained ViT and V-MoE model checkpoints. Our assumption relies on the growing popularity of transfer learning, e.g. Kolesnikov et al. (2020), and the increasing accessibility of pre-trained models in repositories such as `www.tensorflow.org/hub` or `www.pytorch.org/hub`. The fine-tuning of all the approaches we study here, including extensions of ViT and V-MoE, will be either directly compatible with those checkpoints or require only mild adjustments, e.g., reshaping or introducing new downstream-specific parameters (see Appendix C). Also, unless otherwise mentioned, the performance we report will always be downstream, e.g., for ImageNet (Deng et al., 2009) or Cifar10/100 (Krizhevsky, 2009). In all our comparisons, we will use the downstream training floating point operations per second (FLOPs), or GFLOPs (i.e., $10^9 \times$FLOPs), to quantify the computational cost of the different methods.

Table 1: Overview of key properties of sparse MoEs and ensembles. `dense` is a base model upon which we add the sparse MoE or ensemble logic, e.g., a ViT model in this paper.

|  | PREDICTIONS | COMBINATIONS | CONDITIONAL COMPUTATION | COST |
|---|---|---|---|---|
| **Sparse MoEs** | Single | At activation level | Yes, adaptively per-input | $\approx$ `dense` |
| **Ensembles** | Multiple | At prediction level | No, static | $>$ `dense` |

## 3 SPARSE MOES MEET ENSEMBLES

As illustrated in Table 1, sparse MoEs and ensembles have different properties. For instance, ensembles typically do not use conditional computation and just statically combine members at the prediction level. This contrasts with sparse MoEs where the different experts are combined at internal activation levels while enjoying per-input adaptivity through the routing logic; see (1). In terms of cost, sparse MoEs are usually designed to match the cost of their dense counterparts whereas ensembles, in their simplest forms, will typically lead to a substantial overhead. In this section, we study the extent to which these properties are complementary and may benefit from each other. In Section 5, we further evaluate this complementarity on tasks where either sparse MoEs or ensembles are known to perform well, e.g., few-shot and out-of-distribution (OOD) evaluations, respectively. More details about the experiments in this section can be found in Appendix B.6.

### 3.1 STATIC VERSUS ADAPTIVE COMBINATION

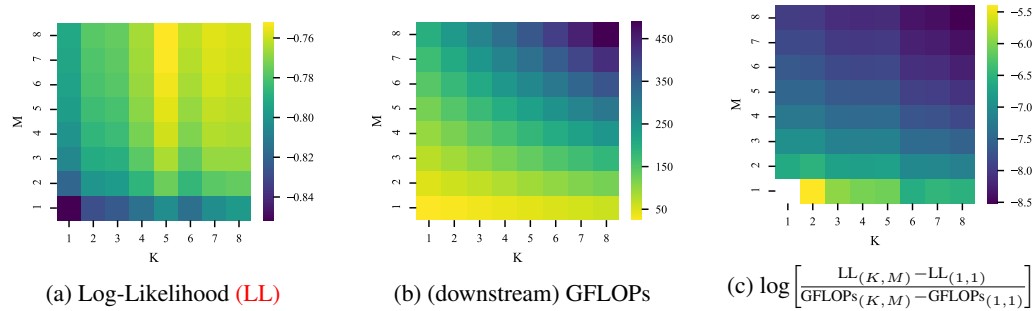

(a) Log-Likelihood (LL)   (b) (downstream) GFLOPs   (c) $\log\left[\frac{\text{LL}_{(K,M)} - \text{LL}_{(1,1)}}{\text{GFLOPs}_{(K,M)} - \text{GFLOPs}_{(1,1)}}\right]$

Figure 2: The effect of increasing static ($M$) and adaptive ($K$) ensembling. ImageNet performance for ViT-S/32 models. **Yellow** indicates better performance; **purple** indicates worse performance.

We first focus on the interplay between the (a) static combination in ensembles and (b) the adaptive combination of experts in sparse MoEs. To this end, we study the performance of *downstream* deep ensembles (i.e., with all ensemble members having the same upstream checkpoint) formed by $M$ independent V-MoEs with $E$ experts per MoE layer and a sparsity level $K$ (the larger $K$, the more selected experts). The parameter $M$ controls the static combination, while $K$ and $E$ impact the adaptive combination of experts in each sparse MoE model. We report in Figure 2 the ImageNet performance and compute cost for ensembles with varying choices of $K$ and $M$, while keeping $E = 32$ fixed. We focus on $K$ rather than $E$ as the axis to explore adaptive computation, as we find that the performance changes for $E$ plateau relatively quickly (see Figure 8 in the Appendix). Also, by fixing $E = 32$, we match more closely the experimental setup of Riquelme et al. (2021). The architecture of the V-MoE is ViT-S/32; see details in Appendix B.6.1. We make the following observations:

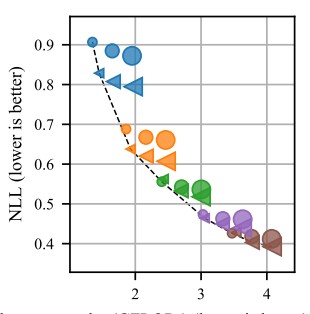

Figure 3: ViT (●) and V-MoE (◄) ensembles of size $M \in \{1, 2, 4\}$ (denoted by markers of increasing size) for S/32 (◆), B/32 (◆), L/32 (◆), L/16 (◆), and H/14 (◆) on ImageNet.

**Cumulative effect.** In the absence of ensembles ($M = 1$), and given a fixed number of experts, Riquelme et al. (2021) already reported an increase in performance as $K$ gets larger. Interestingly, we observe that for each value

Table 2: Feature-level vs. prediction-level ensembling. ImageNet performance of V-MoE and a naive multi-head variant (means $\pm$ standard errors over 5 replications). All models have a ViT-B/32 architecture. For the multi-head variant the last MoE layer is modified as in (2).

| | K | NLL $\downarrow$ | ERROR $\downarrow$ | ECE $\downarrow$ | KL $\uparrow$ |
|---|---|---|---|---|---|
| V-MoE | 2 | $0.638_{\pm 0.001}$ | $\mathbf{16.76}_{\pm 0.05}$ | $0.033_{\pm 0.001}$ | — |
| Naive Multi-head | 2 | $\mathbf{0.633}_{\pm 0.001}$ | $16.85_{\pm 0.01}$ | $\mathbf{0.025}_{\pm 0.000}$ | $0.033_{\pm 0.000}$ |
| V-MoE | 4 | $\mathbf{0.636}_{\pm 0.001}$ | $\mathbf{16.70}_{\pm 0.04}$ | $0.034_{\pm 0.001}$ | — |
| Naive Multi-head | 4 | $0.638_{\pm 0.001}$ | $17.23_{\pm 0.04}$ | $\mathbf{0.020}_{\pm 0.000}$ | $0.012_{\pm 0.000}$ |

of $K$, it is also beneficial to increase the ensemble size $M$. In other words, the static combination of ensembles is beneficial when applied to sparse MoEs. This observation is perhaps surprising since adaptive combination may already encapsulate the effect of static combination. Figure 3, and Appendix H.1, show that the combination of static and adaptive ensembling is beneficial to NLL for a range of ViT families. We also see that the benefits of static ensembling are similar for V-MoE and ViT (which does not have any adaptive ensembling).

**Taking FLOPs into account.** Without any computational constraints, the previous observation would favor approaches with the largest values of $K$ and $M$. However, different values of $(K, M)$ lead to different computational costs, as measured here by FLOPs, with $(K, M) = (1, 1)$ being the cheapest. Figure 2b shows, as expected, that the number of FLOPs grows more quickly along the $M$ axis than along the $K$ axis. To capture the various trade-offs at play, in Figure 2c we report the logarithm of the normalized gains in log likelihood $\frac{\text{LL}_{(K,M)} - \text{LL}_{(1,1)}}{\text{GFLOPs}_{(K,M)} - \text{GFLOPs}_{(1,1)}}$ when going from $(K, M) = (1, 1)$ to other choices of $(K, M)$. Interestingly, it appears more advantageous to first grow $K$, i.e., the adaptive combination, before growing $M$.

## 3.2 Feature-level versus Prediction-level Ensembling

As highlighted in Table 1, an ensemble of size $M$ outputs $M$ predictions for a given input (thereafter, averaged) while sparse MoEs only produce a single prediction. We study the impact of this differentiating property. To this end, we propose a simple variant of sparse MoEs wherein the last MoE layer of the form (1) is replaced by

$$\texttt{multihead-MoE}(\boldsymbol{h}) = \{g_e(\boldsymbol{h}) \cdot \texttt{MLP}_e(\boldsymbol{h})\}_{g_e(\boldsymbol{h}) > 0} \in \mathbb{R}^{K \times Q}, \ \{g_e(\boldsymbol{h})\}_{e=1}^{E} = \texttt{gate}_K(\boldsymbol{Wh}), \ (2)$$

where we have assumed $\texttt{MLP}_e(\boldsymbol{h}) \in \mathbb{R}^Q$. Instead of *summing* the expert outputs like in (1), we *stack* the $K$ selected expert contributions (as a reminder, $\texttt{gate}_K$ zeroes out the $E - K$ smallest weights). Keeping track of those $K$ contributions makes it possible to generate $K$ predictions per input as in the classifier of Figure 1, thus capturing model uncertainty around the true prediction.

Table 2 compares the ImageNet performance—negative log likelihood (NLL), classification error and expected calibration error (ECE) (Guo et al., 2017)—of this naive multi-head method with the standard V-MoE. For $K = 2$, the multi-head method provides small but statistically significant gains in NLL and ECE. However, its classification error is worse. On the other hand, for $K = 4$, both the NLL and classification error for multi-head are worse than V-MoE, despite an even larger improvement in ECE. In fact, the multi-head for $K = 4$ performs *worse* in terms of NLL, classification error, and diversity than for $K = 2$. Note that the KL diversity metric indicates that the multi-head variant is unable to provide diverse predictions (e.g., a downstream ensemble of two V-MoEs with $K = 1$ provides a much higher diversity of 0.073, see Table 10). Following Havasi et al. (2020); Soflaei et al. (2020), a possible fix to this problem would be to consider a multi-head *and multi-input* approach, but we show in Appendix F that this strategy is not effective in our context.

Thus, while *prediction level ensembling can be beneficial for uncertainty calibration* of MoEs, a different strategy is required such that the error is not worse. We propose a better approach next.

## 4 Partitioned Batch Ensemble

Equipped with the insights from Section 3, we describe partitioned batch ensemble (pBE), with the goal of keeping the strengths of both sparse MoEs and ensembles. Conceptually, we can view pBE

as jointly learning an ensemble of smaller sparse MoEs, where all the layers that do not contain experts are shared across the members, e.g., the self-attention layers. As its name indicates, pBE is inspired by batch ensemble in that (a) ensemble members have shared parameters—here, the sharing is tailored to sparse MoEs—and (b) we reuse the idea of tiled representations.

## 4.1 THE ARCHITECTURE

There are two main components in PBE:

**Disjoint subsets of experts as ensemble members.** We change the structure of (1) by *partitioning* the set of $E$ experts into $M$ sets of $E/M$ experts (we assume that $E$ is a multiple of $M$). We denote this partition by $\cup_{m=1}^{M} \mathcal{E}_m$, for example $\mathcal{E}_1 = \{1, 2, 3\}$ and $\mathcal{E}_2 = \{4, 5, 6\}$ for $E = 6$ and $M = 2$. The $M$ sets of $E/M$ experts play the role of the $M$ ensemble members. Intuitively, the ensemble members have separate parameters for independent predictions, while efficiently sharing parameters among all non-expert layers.

Instead of having a single routing function $\texttt{gate}_K(\boldsymbol{W}\cdot)$ like in (1), we apply separate routing functions $\{\texttt{gate}_K(\boldsymbol{W}_m\cdot)\}_{m=1}^{M}$ to each member of the partition. Note that this does not affect the total number of parameters since $\boldsymbol{W}$ has $E$ rows while each $\boldsymbol{W}_m$ has $E/M$ rows. A similar partitioning of the experts was proposed in Yang et al. (2021) but not exploited to create different ensemble members, in particular not in conjunction with tiled representations, which we show to be required to get performance gains (see comparison in Section 4.2.1).

**Tiled representation.** To jointly handle the predictions of the $M$ ensemble members, we tile the inputs by a factor $M$, as proposed in Wen et al. (2019). Tiling naturally fits into the formalism of sparse MoEs, as illustrated by the connection we draw between BE and sparse MoEs (Appendix I). This enables a simple implementation of pBE on top of an existing one of sparse MoEs.

Because of the tiling, a given image patch has $M$ different representations that, when entering an MoE layer, are each routed within their respective parts of the partition $\cup_{m=1}^{M} \mathcal{E}_m$. Formally, consider some tiled inputs $\boldsymbol{H} \in \mathbb{R}^{B \times M \times D}$ where $B$ refers to the batch size and $\boldsymbol{h}_{i,m} \in \mathbb{R}^D$ is the representation of the $i$-th input for the $m$-th member. The routing logic in pBE can be written as

$$\texttt{pBE-MoE}(\boldsymbol{h}_{i,m}) = \sum_{e \in \mathcal{E}_m} g_{e,m}(\boldsymbol{h}_{i,m}) \cdot \texttt{MLP}_e(\boldsymbol{h}_{i,m}), \quad \{g_{e,m}(\boldsymbol{h}_{i,m})\}_{e \in \mathcal{E}_m} = \texttt{gate}_K(\boldsymbol{W}_m \boldsymbol{h}_{i,m}), \quad (3)$$

where the routing weights are now parametrized by $\boldsymbol{W}_m \in \mathbb{R}^{(E/M) \times D}$; see Figure 1. To echo the observations from Section 3, we can first see that pBE brings together the static and adaptive combination of ensembles and sparse MoEs, which we found to be complementary. However, we have seen that static ensembling comes at the cost of a large increase in FLOPs, thus we opt for an efficient ensembling approach. Second, we "split" the MoE layers along the axis of the experts, i.e., from $E$ experts to $M$ times $E/M$ experts. We do so since we observed that the performance of sparse MoEs tends to plateau quickly as the number of experts grows. Finally, pBE retains the important property of ensembles to output multiple predictions per input, which we also saw to be beneficial for uncertainty calibration.

In a generic implementation, we tile a batch of $B$ inputs $\boldsymbol{X} \in \mathbb{R}^{B \times P}$ by a factor $M$ to obtain the tiled inputs $\boldsymbol{X}_{\text{tiled}} = [\boldsymbol{X}; \ldots; \boldsymbol{X}] \in \mathbb{R}^{(M \cdot B) \times P}$ and the model processes $f(\boldsymbol{X}_{\text{tiled}}; \boldsymbol{\theta})$. Since tiling in pBE has an effect only from the first MoE layer onwards, we postpone the tiling operation to that stage, thus saving all prior computations in non MoE-layers that would have been redundant otherwise. For example, for L/16 and $K = M = 2$, we can save about 47% of the FLOPs. We apply the same optimization for all the efficient ensembles methods we compare to in Appendix F. We provide further implementation details of pBE in Appendix D.

## 4.2 ABLATION STUDIES: PARTITIONING AND TILING

Our method introduces two changes to V-MoEs: (a) the partitioning of the experts and (b) the tiling of the representations. In this section, we assess the separate impact of each of those changes and show that it is indeed their combination that explains the performance gains. We summarize the results of the ablation in Table 3 where we show the ImageNet performance of the different variants of V-MoE. All models have a ViT-B/32 base architecture and $K = M = 2$.

Table 3: ImageNet performance (means $\pm$ standard errors over 8 replications) of pBE and two ablations, disabling either the tiling or the expert partitioning. All models have a ViT-B/32 architecture. The level of noise in $\mathtt{gate}_K$ is denoted by $\sigma$.

| | NLL ↓ | ERROR ↓ | ECE ↓ | KL ↑ |
|---|---|---|---|---|
| pBE | **0.612** $_{\pm\,0.001}$ | **16.49** $_{\pm\,0.02}$ | **0.013** $_{\pm\,0.000}$ | **0.198** $_{\pm\,0.003}$ |
| Only tiling ($\sigma \times 1$) | 0.637 $_{\pm\,0.002}$ | 16.74 $_{\pm\,0.06}$ | 0.028 $_{\pm\,0.001}$ | 0.000 $_{\pm\,0.000}$ |
| Only tiling ($\sigma \times 2$) | 0.638 $_{\pm\,0.001}$ | 16.72 $_{\pm\,0.03}$ | 0.033 $_{\pm\,0.001}$ | 0.001 $_{\pm\,0.000}$ |
| Only tiling ($\sigma \times 4$) | 0.638 $_{\pm\,0.001}$ | 16.74 $_{\pm\,0.03}$ | 0.033 $_{\pm\,0.001}$ | 0.002 $_{\pm\,0.000}$ |
| Only partitioning | 0.640 $_{\pm\,0.001}$ | 16.72 $_{\pm\,0.05}$ | 0.034 $_{\pm\,0.001}$ | – |

### 4.2.1 PARTITIONING WITHOUT TILING

We first compare pBE with a variant of V-MoE where we only partition the set of experts ("Only partitioning"). In that variant, each input $\boldsymbol{h}_i \in \mathbb{R}^D$ (note the dropping of the index $m$ due to the absence of tiling) can select $K$ experts in each part of the partition $\cup_{m=1}^M \mathcal{E}_m$, resulting in a total of $K \times M$ selected experts per input. Formally, (3) becomes $\sum_{m=1}^M \sum_{e \in \mathcal{E}_m} g_{e,m}(\boldsymbol{h}_i) \cdot \mathtt{MLP}_e(\boldsymbol{h}_i)$. The *expert prototyping* of Yang et al. (2021) leads to a similar formulation. As shown in Table 3, "Only partitioning" is not competitive with pBE across all metrics. We do not report the KL since without tiling, "Only partitioning" does not output multiple predictions per input.

### 4.2.2 TILING WITHOUT PARTITIONING

We now compare pBE with the variant where only the tiling is enabled ("Only tiling"). In that case, we have tiled inputs $\boldsymbol{H} \in \mathbb{R}^{B \times M \times D}$ applied to the standard formulation of (1). Compared with (3), there is no mechanism to enforce the $M$ representations of the $i$-th input across the ensemble members, i.e., $\{\mathtt{MoE}(\boldsymbol{h}_{i,m})\}_{m=1}^M$, to be different. Indeed, without partitioning, each $\boldsymbol{h}_{i,m}$ could select $K$ identical experts. As a result, we expect "Only tiling" to output $M$ similar predictions across ensemble members. We capture this intuition in Table 3 where we observe that the KL for "Only tiling" is orders of magnitude smaller than for pBE.

To mitigate this effect, we also tried to increase the level of noise $\sigma$ in $\mathtt{gate}_K$ (by a factor $\{2, 4\}$), to cause the expert assignments to differ across $\{\boldsymbol{h}_{i,m}\}_{m=1}^M$. While we do see an increase in KL, "Only tiling" still performs worse than pBE across all metrics. Interestingly, we can interpret "Only tiling" as an approximation, via $M$ samples, of the marginalization $\mathbb{E}_{\varepsilon_1,\dots,\varepsilon_\ell}[f(\boldsymbol{x}; \boldsymbol{\theta})]$ with respect to the noise $\{\varepsilon_l\}_{l=1}^\ell$ in the $\ell$ MoE layers of $f(\cdot; \boldsymbol{\theta})$ (further assuming the capacity constraints of the experts, as described in Riquelme et al. (2021), does not bias the $M$ samples).

## 5 EVALUATION

We now benchmark pBE against V-MoE. As a baseline we also include results for *downstream* ensembles of V-MoE and ViT. These ensembles offer a natural baseline against pBE as they also use a single upstream checkpoint, are easy to implement, and provide consistent improvements upon V-MoE. In Appendix G, we compare with *upstream* ensembles that require multiple upstream checkpoints (Mustafa et al., 2020). In Appendix F, we compare with other efficient ensembling approaches: MIMO (Havasi et al., 2020), BE (Wen et al., 2019), and MC Dropout (Gal & Ghahramani, 2016). All results correspond to the average over 8 (for {S, B, L} single models) or 5 (for H single models and all up/downstream ensembles) replications. In Appendix H we provide standard errors as well as results for additional datasets and metrics. Following Riquelme et al. (2021), we compare the predictive-performance vs. compute cost trade-offs for each method across a range of ViT families. In the results below, pBE uses $(K, M) = (1, 2)$, and V-MoE uses $K = 1$. Experimental details, including about our upstream training, downstream fine-tuning, hyperparameter sweeps and our (linear) few-shot evaluation can be found in Appendix B. Our main findings are as follows:

*(a) V-MoE versus ViT. Predictive performance and robustness.*
• **Ensembles help V-MoE just as much as ViT.** Ensembles were expected to benefit ViT models. However, Figure 3 and Figure 4 suggest that ensembling provides similar gains for V-MoE models in terms of *both* few-shot performance and NLL. We believe this has not been observed before. Moreover, a downstream ensemble with four H/14 V-MoEs leads to a 88.8% accuracy on ImageNet

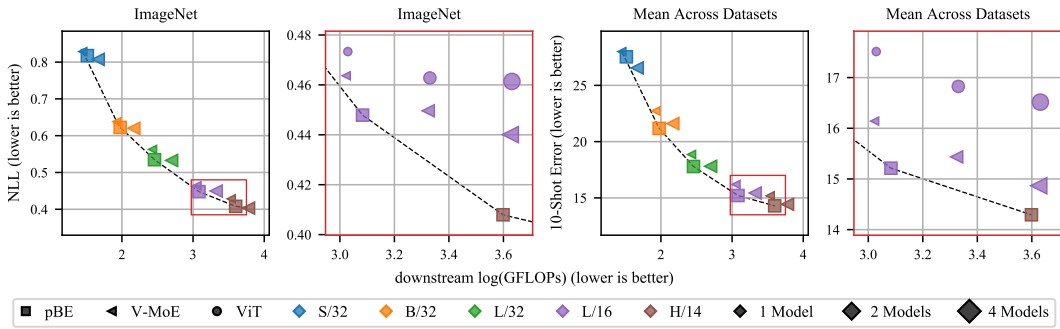

Figure 4: ImageNet NLL (left, center left) and mean 10-shot error across datasets (center right, right). We provide zoomed-in plots of the highlighted areas. The dashed lines show Pareto frontiers.

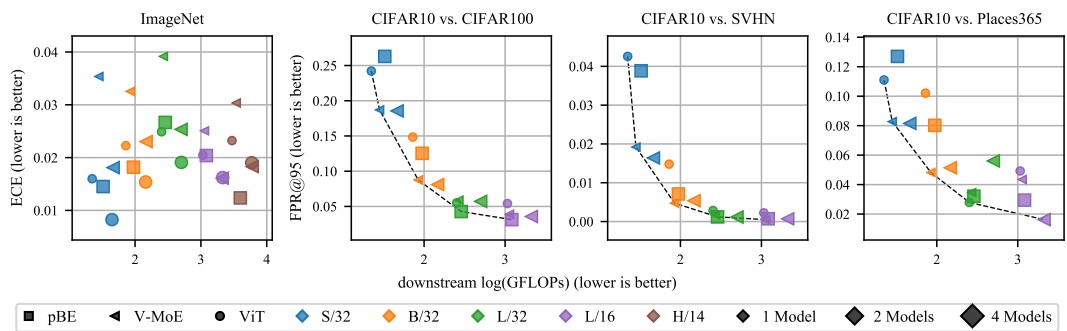

Figure 5: Quality of uncertainty estimates. ImageNet ECE (left), near (center left) and far (center right, right) OOD detection, measured by false positive rate at 95% precision (Fort et al., 2021). These are metrics for which ensembles are known to perform well whereas, to the best of our knowledge, the performance of V-MoE has not been evaluated. The dashed lines represent Pareto frontiers.

(even reaching an impressive 89.3% for an upstream ensemble, see Table 10).

• **ViT consistently provides better ECE than V-MoE.** Surprisingly, despite V-MoE tending to have better NLL than ViT (Figure 3), their ECE is worse (Figure 5).

• **ECE is not consistent for different ViT/V-MoE families.** We see the ECE, unlike other metrics presented in this work, provides no consistent trends as we increase the ViT family size (Figure 5).

• **V-MoE outperforms ViT in OOD detection.** With L/32 being the only exception, V-MoE outperforms ViT on a range of OOD detection tasks (Figure 5).

• **For smaller ViT families, V-MoE outperforms ViT in the presence of distribution shift.** In contrast to the OOD detection results, Figure 6 shows that for smaller ViT families V-MoE improves on the performance of ViT, however, as the ViT family becomes larger, this trend reverses.

*(b) Partitioned Batch Ensembles. Predictive performance and robustness.*

• **pBE improves classification performance.** As shown in Figure 4, pBE is either on or very near to the Pareto frontiers for NLL and 10-shot classification error, despite the fact that these are metrics for which ensembles and V-MoE, respectively, are known to perform well. Furthermore, Figures 14 and 15 show that even starker conclusions hold on Cifar10/100.

• **pBE performs best at the largest scale.** The difference in predictive performance between pBE and V-MoE—or ensembles thereof—increases as the ViT family becomes larger (Figures 4, 5 and 6). See Appendix E for further motivation of this point.

• **pBE tends to be Pareto efficient in the presence of distribution shift**. Figure 6 shows that pBE is more robust to distribution shift for larger ViT families, despite the opposite being true for V-MoE.

• **pBE improves ECE over ViT and V-MoE.** Despite V-MoE providing poor ECE, pBE does not suffer from this limitation (Figure 5). Furthermore, for most ViT families, pBE also provides better ECE than V-MoE ensembles.

• **pBE does not provide consistent OOD detection performance.** Firstly, Figure 5 shows that for small ViT families, pBE performs worse than V-MoE and (even ViT in some cases). Nevertheless, as above, the relative performance improves for larger ViT families such that pBE becomes Pareto efficient for two dataset pairs. Secondly, pBE seems to perform better on the more difficult near

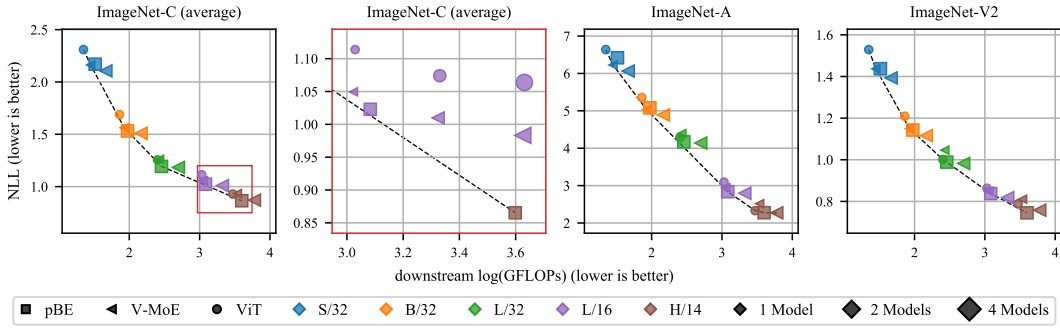

Figure 6: NLL in the presence of distribution shift for models trained on ImageNet. For ImageNet-C, we provide a zoomed-in plot of the highlighted area. The dashed lines represent Pareto frontiers. We provide results for additional distribution shift datasets and metrics in Appendix H.

OOD detection task (Cifar10 vs. Cifar100). These results, although sometimes subtle, are consistent across OOD detection metrics and dataset pairs, as shown in Appendix H.

## 6 RELATED WORK

**Mixture of Experts.** MoEs (Jacobs et al., 1991; Jordan & Jacobs, 1994; Chen et al., 1999; Yuksel et al., 2012; Eigen et al., 2014) combine the outputs of different submodels, or *experts*, in an input-dependent way. Sparse MoEs only select a few experts per input, enabling to greatly scale models while keeping the prediction time constant. Sparse MoEs have been used to build large language models (Shazeer et al., 2017; Lepikhin et al., 2021; Fedus et al., 2021). Recently, sparse MoEs have been also successfully applied to vision problems (Riquelme et al., 2021; Yang et al., 2021; Lou et al., 2021; Xue et al., 2021). Our work builds on the V-MoE architecture proposed by Riquelme et al. (2021), which is based on the vision Transformer (ViT) (Dosovitskiy et al., 2021). While previous work studied ViT's calibration and robustness (Minderer et al., 2021; Fort et al., 2021; Paul & Chen, 2021; Mao et al., 2021), we are the first to study the robustness of V-MoE models.

**Ensembles.** Ensemble methods combine several different models to improve generalization and uncertainty estimation. In their simplest form, they can be inefficient because they consist of multiple models themselves potentially expensive. To reduce test time, Xie et al. (2013) and Hinton et al. (2015) respectively use compression and distillation mechanisms. To reduce training time, ensembles can be constructed with cyclical learning-rate schedules to snapshot models along the training trajectory (Huang et al., 2017; Zhang et al., 2019). Our work builds on batch ensemble (Wen et al., 2019) where a *single* model encapsulates an ensemble of networks, a strategy also explored by Lee et al. (2015); Havasi et al. (2020); Antorán et al. (2020); Dusenberry et al. (2020a); Rame et al. (2021). Wenzel et al. (2020) extended BE to combine models with different hyperparameters.

## 7 CONCLUSIONS AND FUTURE WORK

Our study of the interplay between sparse MoEs and ensembles has shown that these two classes of models are symbiotic. Partitioned batch ensemble exemplifies those mutual benefits—as illustrated by its accuracy, log-likelihood, few-shot learning, robustness, and uncertainty calibration improvements over several challenging baselines in a range of benchmarks. While our study has focused on downstream fine-tuned models, we believe that an extension to the upstream case would also result in a fruitful investigation. Similarly, although we have focused on computer vision, our approach should be readily applicable to the modelling of text, where sparse MoEs have been shown to be remarkably effective. With the growing prevalence of sparse MoEs in NLP (Patterson et al., 2021), the questions of understanding and improving the robustness and reliability of such models become increasingly important. Furthermore, the computational scale at which those models operate make those questions even more challenging to tackle. We believe that our study, and approaches such as pBE, make steps in those directions.

**Ethics Statement.** Our research lies at the intersection of two topics where we hope that our work can make positive contributions.

First, following the conclusions of Patterson et al. (2021), we develop an approach based on sparse MoEs that were shown to reduce the environmental footprint of standard "dense" models. Second, for any decision-making process, it is critical to be able to reliably trust the uncertainty output by ML systems. In particular, this desirable property has a growing importance within a context where those systems are being widely deployed in safety-critical fields such as self-driving cars and medical diagnosis. We think that approaches such as pBE can help make progress in this area.

**Reproducibility Statement.** We are fully aware that (i) relying on the proprietary JFT-300M dataset for our upstream models, as well as (ii) not having already open-sourced code are two obstacles for reproducibility. We are focusing on a lightweight open-sourced version of V-MoE and pBE together with the release of checkpoints pre-trained on ImageNet-21k. We are actively working to release those with the camera-ready version of the paper.

We would like to stress the fact that we have provided as many details as possible about the experimental settings (see Appendix B) and the hyperparameter sweeps. Furthermore, we aimed to provide well-considered and fair experimental setups for all of the baselines used in this work. For instance, to illustrate the fairness of our approach, our experimental design choices tend to make the baselines we compare to more competitive; see Table 6. Moreover, we performed the evaluation of our models using the open-sourced library `robustness_metrics` (Djolonga et al., 2020).

Finally, we have also reported results that could be regarded as "negative" with the hope to inform other researchers about those findings. For instance, in Appendix F.3, we systematically described in detail the failures we encountered while trying to apply MIMO to ViT and V-MoE. Similarly, we have transparently reported—in the core paper—results of pBE for OOD detection where our approach seems to perform worse than the baselines in several settings.

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

|       | HIDDEN DIMENSION | MLP DIMENSION | # LAYERS |
|-------|------------------|---------------|----------|
| Small | 512              | 2048          | 8        |
| Base  | 768              | 3072          | 12       |
| Large | 1024             | 4096          | 24       |
| Huge  | 1280             | 5144          | 32       |

Table 4: Specifications of ViT-S, ViT-B, ViT-L and ViT-H.

## A  ViT Model Specifications

Following Dosovitskiy et al. (2021), we recall the specifications of the ViT models of different scales in Table 4.

## B  Experiment Settings

### B.1  Upstream Setting

For all our upstream experiments, we scrupulously follow the setting described in Riquelme et al. (2021), see their Section B.2 in their appendix. For completeness, we just recall that S/32 models are trained for 5 epochs while B/{16, 32} and L/32 models are trained for 7 epochs. For L/16 models, both 7 and 14 epochs can be considered (Dosovitskiy et al., 2021; Riquelme et al., 2021); we opted for 7 epochs given the breadth of our experiments. Finally, the H/14 model are trained for 14 epochs.

In particular, the models are all trained on JFT-300M (Sun et al., 2017). This dataset contains about 305M training and 50 000 validation images. The labels have a hierarchical structure, with a total of 18 291 classes, leading on average to 1.89 labels per image.

### B.2  Downstream Setting

During fine-tuning, there is a number of *common* design choices we apply. In particular:

- Image resolution: 384.
- Clipping gradient norm at: 10.0.
- Optimizer: SGD with momentum (using half-precision, $\beta = 0.9$).
- Batch size: 512.
- For V-MoE models, we finetune with capacity ratio $C = 1.5$ and evaluate with $C = 8$.

We use the following train/validation split partitions depending on the dataset:

| DATASET           | TRAIN DATASET FRACTION | VALIDATION DATASET FRACTION |
|-------------------|------------------------|-----------------------------|
| ImageNet          | 99%                    | 1%                          |
| CIFAR10           | 98%                    | 2%                          |
| CIFAR100          | 98%                    | 2%                          |
| Oxford-IIIT Pets  | 90%                    | 10%                         |
| Oxford Flowers-102| 90%                    | 10%                         |

All those design choices follow from Riquelme et al. (2021) and Dosovitskiy et al. (2021).

### B.3  Hyperparameter Sweep for Fine-tuning

In all our fine-tuning experiments, we use the sweep of hyperparameters described in Table 5. We use the recommendations from Dosovitskiy et al. (2021) and Riquelme et al. (2021), further considering several factors {0.5, 1.0, 1.5, 2.0} to sweep over different numbers of steps. Riquelme et al. (2021) use a half schedule (with the factor 0.5) while Dosovitskiy et al. (2021) take the factor 1.0.

Table 5: Hyperparameter values for fine-tuning on different datasets. Compared with Dosovitskiy et al. (2021) and Riquelme et al. (2021), we further consider several factors {0.5, 1.0, 1.5, 2.0} to sweep over different numbers of steps.

| DATASET | STEPS | BASE LR | EXPERT DROPOUT |
|---|---|---|---|
| ImageNet | $20\,000 \times$ {0.5, 1.0, 1.5, 2.0} | {0.0024, 0.003, 0.01, 0.03} | 0.1 |
| CIFAR10 | $5\,000 \times$ {0.5, 1.0, 1.5, 2.0} | {0.001, 0.003, 0.01, 0.03} | 0.1 |
| CIFAR100 | $5\,000 \times$ {0.5, 1.0, 1.5, 2.0} | {0.001, 0.003, 0.01, 0.03} | 0.1 |
| Oxford-IIIT Pets | $500 \times$ {0.5, 1.0, 1.5, 2.0} | {0.001, 0.003, 0.01, 0.03} | 0.1 |
| Oxford Flowers-102 | $500 \times$ {0.5, 1.0, 1.5, 2.0} | {0.001, 0.003, 0.01, 0.03} | 0.1 |

Table 6: Impact of using the enlarged sweep of hyperparameters described in Table 5. We typically improve the results reported in Riquelme et al. (2021), therefore strengthening the baselines we compare to. The table displays means and standard errors over 8 replications, except for H/14 that has 4 replications. **L/16\***: For L/16, we consider the setting where the upstream models are trained with 7 epochs, as opposed to 14 epochs in Riquelme et al. (2021), hence the slightly worse accuracy reported in this paper.

| MODEL SIZE | MODEL NAME | ACCURACY (THIS PAPER) | ACCURACY (Riquelme et al., 2021) |
|---|---|---|---|
| S/32 | ViT | $76.31_{\pm 0.05}$ | 73.73 |
| | V-MoE (K=2) | $78.91_{\pm 0.08}$ | 77.10 |
| B/32 | ViT | $81.35_{\pm 0.08}$ | 80.73 |
| | V-MoE (K=2) | $83.24_{\pm 0.05}$ | 82.60 |
| L/32 | ViT | $84.62_{\pm 0.05}$ | 84.37 |
| | V-MoE (K=2) | $84.95_{\pm 0.03}$ | 85.04 |
| B/16 | ViT | $84.30_{\pm 0.06}$ | 84.15 |
| | V-MoE (K=2) | $85.40_{\pm 0.04}$ | 85.39 |
| L/16\* | ViT | $86.63_{\pm 0.08}$ | 87.12 |
| | V-MoE (K=2) | $87.12_{\pm 0.04}$ | 87.54 |
| H/14 | ViT | $88.01_{\pm 0.05}$ | 88.08 |
| | V-MoE (K=2) | $88.11_{\pm 0.13}$ | 88.23 |

We show in Table 6 the impact of this enlarged sweep of hyperparameters in the light of the results reported in Riquelme et al. (2021). We notably tend to improve the performance of ViT and V-MoE (especially for smaller models), which thus makes the baselines we compare to more competitive.

### B.4 DETAILS ABOUT THE (LINEAR) FEW-SHOT EVALUATION

We follow the evaluation methodology proposed by Dosovitskiy et al. (2021); Riquelme et al. (2021) which we recall for completeness. Let us rewrite our model $f$ with parameters $\boldsymbol{\theta} = \{\boldsymbol{Q}, \boldsymbol{\theta}'\}$ as

$$f(\boldsymbol{x}; \boldsymbol{\theta}) = \texttt{softmax}(\boldsymbol{Q}\phi(\boldsymbol{x}; \boldsymbol{\theta}'))$$

where $\boldsymbol{Q} \in \mathbb{R}^{C \times S}$ corresponds to the parameters of the last layer of $f$ with the $S$-dimensional representation $\phi(\boldsymbol{x}; \boldsymbol{\theta}') \in \mathbb{R}^S$.

In linear few-shot evaluation, we construct a linear classifier to predict the target labels (encoded as one-hot vectors) from the $S$-dimensional feature vectors induced by $\phi(\cdot; \boldsymbol{\theta}')$; see Chapter 5 in Hastie et al. (2017) for more background about this type of linear classifiers. This evaluation protocol makes it possible to evaluate the quality of the representations $\phi$ learned by $f$.

While Dosovitskiy et al. (2021); Riquelme et al. (2021) essentially focus on the quality of the representations learned upstream on JFT by computing the (linear) few-shot accuracy on ImageNet, we are interested in the representations after fine-tuning on ImageNet. As a result, we consider a collection of 8 few-shot datasets (that does not contain ImageNet):

- Caltech-UCSD Birds 200 (Wah et al., 2011) with 200 classes,
- Caltech 101 (Bansal et al., 2021) with 101 classes,
- Cars196 (Krause et al., 2013) with 196 classes,

- Cifar100 (Krizhevsky, 2009) with 100 classes,
- Colorectal histology (Kather et al., 2016) with 8 classes,
- Describable Textures Dataset (Cimpoi et al., 2014) with 47 classes,
- Oxford-IIIT pet (Parkhi et al., 2012) with 37 classes and
- UC Merced (Yang & Newsam, 2010) with 21 classes.

In the experiments, we compute the few-shot accuracy for each of the above datasets and we report the averaged accuracy over the datasets, for various number of shots in $\{1, 5, 10, 25\}$. As commonly defined in few-shot learning, we understand by $s$ shots a setting wherein we have access to $s$ training images per class label in each of the dataset.

To account for the different scales of accuracy across the 8 datasets, we also tested to compute a weighted average, normalizing by the accuracy of a reference model (ViT-B/32). This is reminiscent of the normalization carried out in Hendrycks & Dietterich (2019) according to the score of AlexNet. We found the conclusions with the standard average and weighted average to be similar.

### B.4.1 SPECIFIC CONSIDERATIONS IN THE ENSEMBLE CASE

For an ensemble with $M$ members, we have access to $M$ representations $\{\phi(\boldsymbol{x}; \boldsymbol{\theta}'_m)\}_{m=1}^M$ for a given input $\boldsymbol{x}$. We have explored two ways to use those representations:

- **Joint**: We concatenate the $M$ representations $\{\phi(\boldsymbol{x}; \boldsymbol{\theta}'_m)\}_{m=1}^M$ into a single "joint" feature vector in $\mathbb{R}^{M \times S}$, remembering that each $\phi(\boldsymbol{x}; \boldsymbol{\theta}'_m) \in \mathbb{R}^S$. We then train a *single* a linear classifier to predict the target labels from the "joint" feature vectors.
- **Disjoint**: For each of the $M$ representations $\{\phi(\boldsymbol{x}; \boldsymbol{\theta}'_m)\}_{m=1}^M$, we separately train a linear classifier to predict the target labels from the feature vectors induced by $\phi(\boldsymbol{x}; \boldsymbol{\theta}'_m)$. We then average the predictions of the $M$ linear classifiers trained in this fashion.

In Table 7, we report a comparison of those approaches. We aggregate the results over all ensemble models (namely, pBE and upstream ViT/V-MoE ensembles of size 2 and 4) and over 8 replications, for the ViT families S/32, B/32 and L/32.

The results indicate that "joint" and "disjoint" perform similarly. Throughout our experiments, we use the "joint" approach because it eased some implementation considerations.

### B.5 LIST OF DATASETS

For completeness, in addition to the few-shot datasets listed in Appendix B.4, we list the datasets used for downstream training and evaluation in this work.

- ImageNet (ILSVRC2012) (Deng et al., 2009) with 1000 classes and 1281167 training examples.
- ImageNet-C (Hendrycks & Dietterich, 2019), an ImageNet test set constructed by applying 15 different corruptions at 5 levels of intensity to the original ImageNet test set. (We report the mean performance over the different corruptions and intensities.)
- ImageNet-A (Hendrycks et al., 2019), an ImageNet test set constructed by collecting new data and keeping only those images which a ResNet-50 classified incorrectly.
- ImageNet-V2 (Recht et al., 2019), an ImageNet test set independently collected using the same methodology as the original ImageNet dataset.
- Cifar10 (Krizhevsky, 2009) with 10 classes and 50000 training examples.
- Cifar10-C (Hendrycks & Dietterich, 2019), a Cifar10 test set constructed by applying 15 different corruptions at 5 levels of intensity to the original Cifar10 test set. (We report the mean performance over the different corruptions and intensities.)
- Cifar100 (Krizhevsky, 2009) with 100 classes and training 50000 examples.
- Oxford Flowers 102 (Nilsback & Zisserman, 2008) with 102 classes and 1020 training examples.

Table 7: Comparison of two approaches, "joint" and "disjoint", to compute the linear few-shot evaluation in the case of ensembles. For the ViT families S/32, B/32 and L/32, the mean error across datasets is averaged over 8 replications and over all the ensemble models of size 2 and 4.

| MODEL SIZE | METHOD | MEAN ERROR ACROSS DATASETS | | | |
| --- | --- | --- | --- | --- | --- |
| | | 1 SHOT | 5 SHOTS | 10 SHOTS | 25 SHOTS |
| S/32 | disjoint | $51.01_{\pm 0.43}$ | $32.80_{\pm 0.34}$ | $26.33_{\pm 0.26}$ | $20.97_{\pm 0.18}$ |
| | joint | $51.12_{\pm 0.42}$ | $32.81_{\pm 0.30}$ | $26.30_{\pm 0.24}$ | $20.77_{\pm 0.17}$ |
| B/32 | disjoint | $42.43_{\pm 0.41}$ | $25.49_{\pm 0.21}$ | $20.30_{\pm 0.15}$ | $15.98_{\pm 0.11}$ |
| | joint | $42.59_{\pm 0.40}$ | $25.74_{\pm 0.18}$ | $20.54_{\pm 0.13}$ | $16.06_{\pm 0.10}$ |
| L/32 | disjoint | $36.41_{\pm 0.31}$ | $21.49_{\pm 0.15}$ | $17.13_{\pm 0.12}$ | $13.56_{\pm 0.10}$ |
| | joint | $36.48_{\pm 0.30}$ | $21.66_{\pm 0.13}$ | $17.34_{\pm 0.10}$ | $13.56_{\pm 0.08}$ |

- Oxford-IIIT pet (Parkhi et al., 2012) with 37 classes and 3680 training examples.

- SVHN (Netzer et al., 2011) with 10 classes.

- Places365 (Zhou et al., 2017) with 365 classes.

- Describable Textures Dataset (DTD) (Cimpoi et al., 2014) with 47 classes.

## B.6 ABLATION DETAILS

### B.6.1 STATIC VERSUS ADAPTIVE ABLATION DETAILS

The setup for the experiments in Figures 2 and 8 differs slightly the other experiments in this paper. Specifically, while for all other experiments we used upstream V-MoE checkpoints with $(K, E) = (2, 32)$, for these experiments we matched the upstream and downstream checkpoints. We did this to avoid a checkpoint mismatch as a potential confounder in our results.

### B.6.2 FEATURE-LEVEL VERSUS PREDICTION-LEVEL ENSEMBLING ABLATION DETAILS

The "Naive Multi-Head" method presented in Section 3.2 was trained in almost the same manner as the vanilla V-MoE, the only difference being the handling of multiple predictions. This was accomplished by using the average ensemble member cross entropy as described for pBE in Appendix D.

On the other hand, in order to compute the evaluation metrics presented in Table 2, we first averaged predictions of the model and then used the average prediction when calculating each metric.

## C COMPATIBILITY AND ADAPTATION OF THE UPSTREAM CHECKPOINTS

Throughout the paper, we make the assumption that we can start from existing checkpoints of ViT and V-MoE models (trained on JFT-300M; see Appendix B.1). We next describe how we can use those checkpoints for the fine-tuning of the extensions of ViT and V-MoE that we consider in this paper.

In all our experiments that involve V-MoEs, we consider checkpoints with $K = 2$ and $E = 32$, which is the canonical setting advocated by Riquelme et al. (2021).

### C.1 PARTITIONED BATCH ENSEMBLES (PBE)

In the case of pBE, the set of parameters is identical to that of a V-MoE model. In particular, neither the tiled representation nor the partitioning of the experts transforms the set of parameters.

To deal with the fact that the single routing function $\text{gate}_K(\boldsymbol{W}\cdot)$ of a V-MoE becomes separate routing functions $\{\text{gate}_K(\boldsymbol{W}_m\cdot)\}_{m=1}^{M}$, one for each part of the partition, we simply slice row-wise $\boldsymbol{W} \in \mathbb{R}^{E \times D}$ into the $M$ matrices $\boldsymbol{W}_m \in \mathbb{R}^{(E/M) \times D}$.

## C.2 BATCH ENSEMBLES (BE)

We train BE starting from ViT checkpoints, which requires to introduce downstream-specific parameters. Following the design of V-MoEs, we place the batch-ensemble layers in the MLP layers of the Transformer.

Let us consider a dense layer in one of those MLPs, with parameters $\boldsymbol{U} \in \mathbb{R}^{D \times L}$, in absence of bias term. In BE, the parametrization of each ensemble member has the following structure $\boldsymbol{U}_m = \boldsymbol{U} \circ (\boldsymbol{r}_m \boldsymbol{s}_m^\top)$ where $\{\boldsymbol{r}_m\}_{m=1}^M$ and $\{\boldsymbol{s}_m\}_{m=1}^M$ are respectively $D$- and $L$-dimensional vectors.

A standard ViT checkpoint provides pre-trained parameters for $\boldsymbol{U}$. We then introduce $\{\boldsymbol{r}_m\}_{m=1}^M$ and $\{\boldsymbol{s}_m\}_{m=1}^M$ at fine-tuning time, following the random initialization schemes proposed in Wen et al. (2019); see details in the hyperparameter sweep for BE in Appendix F.1.

## C.3 MIMO

We train MIMO models from V-MoE checkpoints. The only required modifications are to the input and output parameters of the checkpoints. The linear input embedding must be modified to be compatible with input images containing $M$ times as more channels, as required by the multiple-input structure of MIMO. Similarly, the final dense layer in the classification head must be modified to have $M$ times more output units, following the multiple-output structure in MIMO.

Concretely, the embedding weight $\boldsymbol{W}_{\text{in}} \in \mathbb{R}^{H \times W \times 3 \times D}$ is replicated in the third (channel) dimension, resulting in $\boldsymbol{W}_{\text{MIMO,in}} \in \mathbb{R}^{H \times W \times 3 \cdot M \times D}$, where $H$ and $W$ are the height and width of the convolution and $D$ is the hidden dimension of the ViT family (specified in Table 4). The output layer weight $\boldsymbol{W}_{\text{out}} \in \mathbb{R}^{D \times C}$ is replicated in the second (output) dimension, resulting in $\boldsymbol{W}_{\text{MIMO,out}} \in \mathbb{R}^{H \times C \cdot M}$, where $C$ is the number of classes. The output layer bias $\boldsymbol{b}_{\text{out}} \in \mathbb{R}^C$ is replicated resulting in $\boldsymbol{b}_{\text{MIMO,out}} \in \mathbb{R}^{C \times M}$. Finally, in order to preserve the magnitude of the activation for these layers, $\boldsymbol{W}_{\text{MIMO,in}}$ and $\boldsymbol{W}_{\text{MIMO,out}}$ are scaled by $1/M$.

## D IMPLEMENTATION DETAILS OF PBE

We provide details about the training loss and the regularizer used by pBE.

### D.1 TRAINING LOSS

Since pBE outputs $M$ predictions $\{f(\boldsymbol{x}; \boldsymbol{\theta}_m)\}_{m=1}^M$ for a given input $\boldsymbol{x}$, we need to adapt the choice of the training loss $\mathcal{L}$ accordingly. Following the literature on efficient ensembles (Wen et al., 2019; Dusenberry et al., 2020a; Wenzel et al., 2020), we choose the average ensemble-member cross entropy

$$\mathcal{L}(y, \boldsymbol{x}; \boldsymbol{\theta}) = \frac{1}{M} \sum_{m=1}^M \text{cross-entropy}(y, f(\boldsymbol{x}; \boldsymbol{\theta}_m))$$

instead of other alternatives such as the ensemble cross-entropy

$$\text{cross-entropy}\Big(y, \frac{1}{M} \sum_{m=1}^M f(\boldsymbol{x}; \boldsymbol{\theta}_m)\Big)$$

that was observed to generalize worse (Dusenberry et al., 2020a).

### D.2 AUXILIARY LOSSES

Inspired by previous applications of sparse MoEs in NLP (Shazeer et al., 2017), Riquelme et al. (2021) employ regularizers, also referred to as *auxiliary losses*, to guarantee a balanced usage of the $E$ experts. Two auxiliary losses—the importance and load losses, see Appendix A in Riquelme et al. (2021) for their formal definitions—are averaged together to form the final regularization term that we denote by $\Omega$.

As a reminder, let us recall the notation of the routing function

$$\boldsymbol{h} \in \mathbb{R}^D \mapsto \texttt{gate}_K(\boldsymbol{W}\boldsymbol{h}) = \texttt{top}_K(\texttt{softmax}(\boldsymbol{W}\boldsymbol{h} + \sigma \varepsilon)) \in \mathbb{R}^E,$$

with $\boldsymbol{W} \in \mathbb{R}^{E \times D}$ and $\varepsilon \sim \mathcal{N}(\boldsymbol{0}, \boldsymbol{I})$. Consider a batch of $B$ inputs $\{\boldsymbol{h}_i\}_{i=1}^B$ that we represent by $\boldsymbol{H} \in \mathbb{R}^{B \times D}$. Finally, let us define

$$\boldsymbol{A} = \boldsymbol{H} \boldsymbol{W}^\top + \sigma \varepsilon_{B \times E} \in \mathbb{R}^{B \times E},$$

where we emphasise that $\varepsilon_{B \times E}$ is a matrix of Gaussian noise entries in $\mathbb{R}^{B \times E}$. The regularization term $\Omega$ used by Riquelme et al. (2021) can be seen as a function that depends on $\boldsymbol{A}$ and $\boldsymbol{H} \boldsymbol{W}^\top$.

In the context of partitioned batch ensemble, the set of $E$ experts is partitioned into $M$ groups of $E/M$ experts, whose partition is denoted by $\cup_{m=1}^M \mathcal{E}_m$; see Section 4.1. With the introduction of the $M$ routing functions $\{\mathtt{gate}_K(\boldsymbol{W}_m \cdot)\}_{m=1}^M$ with each $\boldsymbol{W}_m \in \mathbb{R}^{(E/M) \times D}$, the matrix $\boldsymbol{A}$ becomes accordingly partitioned into $\{\boldsymbol{A}_m\}_{m=1}^M$ where each $\boldsymbol{A}_m \in \mathbb{R}^{B \times (E/M)}$.

Since we want to enforce a balanced usage of the $E/M$ experts in each part $\mathcal{E}_m$ of the partition, we thus redefine the regularization as the average regularization separately applied to each part of the partition

$$\Omega_{\text{partition}}(\boldsymbol{A}, \boldsymbol{H} \boldsymbol{W}^\top) = \frac{1}{M} \sum_{m=1}^M \Omega(\boldsymbol{A}_m, \boldsymbol{H} \boldsymbol{W}_m^\top).$$

We found this option to work better in practice. To guarantee a fair comparison, we also applied $\Omega_{\text{partition}}$ to the "Only partitioning" model in the ablation study of Section 4.2.1.

Following Riquelme et al. (2021), the regularization parameter controlling the strength of $\Omega_{\text{partition}}$ was set to 0.1 throughout the experiments.

# E    PBE AND V-MOE RELATIVE IMPROVEMENTS PER VIT FAMILY

In Section 5 we claim that pBE performs best at the largest scale. In this section we motivate that claim in more detail. Specifically, we consider two metrics of improvement in performance. Firstly, we consider the percentage improvement in NLL for both pBE and V-MoE versus vanilla ViT. Secondly, we consider a normalised version of this improvement. We consider this second metric to take into account the "difficulty" in further improving the NLL of larger ViT family models. Intuitively, the larger the ViT family, the better the corresponding NLL will be, and the more difficult it will be to improve on that NLL.

The normalisation we apply is based on the gradient of the NLL with respect to FLOPs. Indeed, this gradient captures the typical variation of NLL at a particular amount of FLOPs. The ratio of this gradient at the FLOPs values (i.e., the instantaneous change in NLL at those FLOPs values) for two ViT families is a measure of the relative difficulty in increasing the NLL. Thus, we can use this ratio to normalise our results. To be more concrete, let us define the mapping

$$\text{NLL} = \varphi(\text{FLOPs}) \quad \text{and its derivative} \quad \varphi'(\text{FLOPs}) = \frac{d\varphi(\text{FLOPs})}{d\text{FLOPs}}.$$

We estimate $\varphi$ and its gradient by fitting a linear model to the $(\text{NLL}, \text{FLOPs})$ pairs for each ViT family, using the data of the standard ViT models we trained. We use feature expansion $[1.0, \log(\text{FLOPs}), \log(\text{FLOPs})^2, \log(\text{FLOPs})^3]$ and solve for the parameters of the linear model via ordinary least squares. We determine the gradient of this function at each FLOPs value using automatic differentiation in JAX (Bradbury et al., 2018). See Figure 7 for the resulting fit and an indication of the gradients.

The normalised values are calculated as:

$$\text{Normalised improvement}(v) = \text{improvement}(v) \times \frac{\varphi'(\text{FLOPs}_{\text{H/14}})}{\varphi'(\text{FLOPs}_v)}, \tag{4}$$

where $v$ is one of the ViT families, i.e., S/32, B/32, L/32, L/16, or H/14. Note that this normalisation leaves the improvement for H/14 the same. We tried to normalize with respect to other choices of ViT family, different from H/14. Our conclusions are robust, in the sense that both the ordering and the monotonic behavior with respect to scale are preserved. Using the ratio for normalisation also has the advantage that the normalisation is less sensitive to the particular parameterisation of $\varphi$.

Table 8 shows both the difficulty-normalised and original improvements (without normalisation). Looking first at the original improvements, we can see that while both pBE and V-MoE have smaller

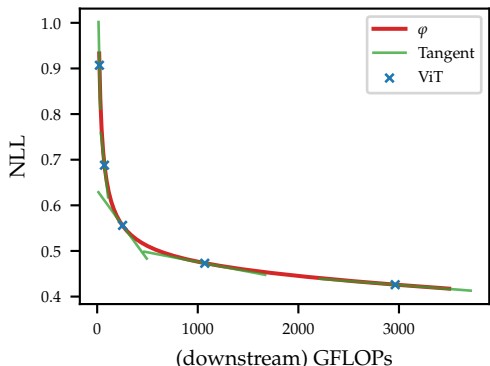

Figure 7: Estimated $\varphi$ compared to the ImageNet NLL values for our ViT models. We also include the tangent at the points corresponding to each ViT model to indicate the gradients at those points.

Table 8: Percentage improvements in NLL for pBE with $(K, M) = (1, 2)$ and V-MoE with $K = 1$ vs. ViT for families of increasing size. The top two rows show normalised improvements, see (4), which take into consideration the increased difficulty of improving NLL for larger ViT families whose performance is beginning to saturate. The bottom two rows are the original percentage improvements without normalisation.

|  |  | S/32 | B/32 | L/32 | L/16 | H/14 |
|---|---|---|---|---|---|---|
| Normalised | pBE vs. ViT | 0.02% | 0.08% | 0.22% | 2.21% | 4.27% |
| | V-MoE vs. ViT | 0.01% | 0.06% | -0.03% | 0.84% | 0.02% |
| Not normalised | pBE vs. ViT | 9.82% | 9.53% | 3.76% | 5.38% | 4.27% |
| | V-MoE vs. ViT | 7.98% | 6.62% | -0.60% | 2.05% | 0.02% |

improvements over ViT for larger families, pBE's improvements decrease more slowly. Furthermore, by comparing the normalised improvements we see that pBE's improvements actually *grow* monotonically when taking difficulty into account. This is not the case for V-MoE.

## F  EFFICIENT ENSEMBLE COMPARISONS

In this section, we compare partitioned batch ensembles (pBE) to several popular efficient ensemble approaches, namely MIMO (Havasi et al., 2020), batch ensemble (BE) (Wen et al., 2019), and MC Dropout (Gal & Ghahramani, 2016).

Table 9 reports the ImageNet performance of those different techniques, when all models are based on a ViT-B/32 architecture. We start by highlighting the most salient conclusions of the experiment and defer to the next subsections the descriptions of the different competing techniques.

We make the following observations:

- BE built upon ViT improves the performance of ViT in terms of NLL, classification error and ECE. However, the resulting increase in FLOPs makes BE a less viable option compared to pBE.

- MC dropout V-MoE is on par with standard V-MoE in terms of NLL and classification error, while it improves the ECE. For all values of $K$, we observe that the performance tends to improve as the number of samples, i.e., $M$, increases. However, already for $M$ in $\{2, 4\}$, the resulting increase in FLOPs makes MC dropout V-MoE a less favorable option compared to pBE.

- Perhaps surprisingly (see detailed investigations in Appendix F.3), MIMO V-MoE does not lead to improvements compared with V-MoE. In fact, for higher ensembles sizes, MIMO V-MoE results in worse performance than standard V-MoE. Moreover, increasing the batch

Table 9: ImageNet performance of different efficient ensemble approaches. The table reports the means $\pm$ standard errors over 8 replications. All models have a ViT-B/32 architecture. $K$ stands for the sparsity in V-MoEs, $M$ denotes the ensemble size while "BR" corresponds to the batch repetition in MIMO (Havasi et al., 2020).

| | K | M | NLL | ERROR | ECE | KL | GFLOPs |
|---|---|---|---|---|---|---|---|
| ViT | – | – | $0.688_{\pm 0.003}$ | $18.65_{\pm 0.08}$ | $0.022_{\pm 0.000}$ | – | 72.5 |
| BE ViT | – | 2 | $0.682_{\pm 0.003}$ | $18.47_{\pm 0.05}$ | $0.021_{\pm 0.000}$ | $0.040_{\pm 0.001}$ | 94.8 |
| | – | 4 | $0.675_{\pm 0.003}$ | $18.40_{\pm 0.09}$ | $0.017_{\pm 0.000}$ | $0.035_{\pm 0.001}$ | 132.1 |
| V-MoE | 1 | – | $0.642_{\pm 0.002}$ | $16.90_{\pm 0.05}$ | $0.029_{\pm 0.001}$ | – | 73.5 |
| | 2 | – | $0.638_{\pm 0.001}$ | $16.76_{\pm 0.05}$ | $0.033_{\pm 0.001}$ | – | 84.7 |
| | 4 | – | $0.636_{\pm 0.001}$ | $16.70_{\pm 0.04}$ | $0.034_{\pm 0.001}$ | – | 107.0 |
| | 8 | – | $0.635_{\pm 0.002}$ | $16.72_{\pm 0.06}$ | $0.028_{\pm 0.001}$ | – | 151.7 |
| MC dropout V-MoE | 1 | 2 | $0.648_{\pm 0.002}$ | $17.10_{\pm 0.05}$ | $0.019_{\pm 0.001}$ | $0.046_{\pm 0.000}$ | 95.9 |
| | 1 | 4 | $0.641_{\pm 0.002}$ | $16.96_{\pm 0.05}$ | $0.017_{\pm 0.001}$ | $0.046_{\pm 0.001}$ | 138.4 |
| | 2 | 2 | $0.642_{\pm 0.002}$ | $16.94_{\pm 0.04}$ | $0.021_{\pm 0.001}$ | $0.046_{\pm 0.001}$ | 119.6 |
| | 2 | 4 | $0.634_{\pm 0.001}$ | $16.80_{\pm 0.03}$ | $0.020_{\pm 0.000}$ | $0.046_{\pm 0.001}$ | 183.5 |
| | 4 | 2 | $0.639_{\pm 0.002}$ | $16.91_{\pm 0.06}$ | $0.022_{\pm 0.001}$ | $0.045_{\pm 0.001}$ | 164.0 |
| MIMO V-MoE (BR=1) | 2 | 2 | $0.636_{\pm 0.002}$ | $16.97_{\pm 0.04}$ | $0.028_{\pm 0.001}$ | $0.000_{\pm 0.000}$ | 90.2 |
| | 2 | 4 | $0.672_{\pm 0.001}$ | $17.72_{\pm 0.04}$ | $0.037_{\pm 0.000}$ | $0.001_{\pm 0.000}$ | 92.8 |
| MIMO V-MoE (BR=2) | 2 | 2 | $0.638_{\pm 0.001}$ | $17.14_{\pm 0.03}$ | $0.031_{\pm 0.000}$ | $0.001_{\pm 0.000}$ | 180.3 |
| | 2 | 4 | $0.665_{\pm 0.002}$ | $17.38_{\pm 0.04}$ | $0.038_{\pm 0.000}$ | $0.000_{\pm 0.000}$ | 185.4 |
| pBE | 1 | 2 | $0.622_{\pm 0.001}$ | $16.70_{\pm 0.03}$ | $0.018_{\pm 0.000}$ | $0.217_{\pm 0.003}$ | 94.5 |
| | 1 | 4 | $0.624_{\pm 0.001}$ | $16.99_{\pm 0.03}$ | $0.013_{\pm 0.000}$ | $0.164_{\pm 0.001}$ | 136.3 |
| | 2 | 2 | $0.612_{\pm 0.001}$ | $16.49_{\pm 0.02}$ | $0.013_{\pm 0.000}$ | $0.198_{\pm 0.003}$ | 116.8 |
| | 2 | 4 | $0.620_{\pm 0.001}$ | $16.86_{\pm 0.02}$ | $0.015_{\pm 0.000}$ | $0.170_{\pm 0.001}$ | 181.0 |
| | 4 | 2 | $0.611_{\pm 0.001}$ | $16.45_{\pm 0.03}$ | $0.014_{\pm 0.000}$ | $0.193_{\pm 0.003}$ | 161.4 |

repetition parameter of MIMO ("BR" in Table 9) further worsens the results. Interestingly, we can see that MIMO does not manage to produce diverse predictions, as illustrated by the small values of KL.

- pBE offers the best performance vs. FLOPs trade-offs, e.g., when looking at $(K, M) = (1, 2)$ and $(K, M) = (2, 2)$. We notably observe that the diversity of the predictions in pBE is orders of magnitude larger than that of the other ensemble approaches.

We briefly recall the optimization explained in Section 4.1 to save redundant computations: In the "last-$n$" setting of Riquelme et al. (2021), it is sufficient to tile the representations only when entering the first MoE layer/dropout layer/batch-ensemble layer for respectively pBE/MC dropout V-MoE/BE. We apply this optimization to all the efficient ensemble methods.

### F.1 BATCH ENSEMBLES

Following the design of V-MoEs, we place the batch-ensemble layers in the MLP layers of the Transformer, following the "last-$n$" setting of Riquelme et al. (2021); see Section 2.1.

The vectors of the rank-1 parametrization introduced at fine-tuning time (see Appendix C) need to be initialized and optimized. Following the recommendation from Wen et al. (2019), we consider the following hyperparameters in addition to the common sweep described in Table 5:

- **Initialization:** Either a random sign vector with entries in $\{-1, 1\}$ independently drawn with probability $\frac{1}{2}$ or a random Gaussian vector with entries independently drawn from $\mathcal{N}(1, 0.5)$.

- **Learning-rate scale factor:** The vectors of the rank-1 parametrization are updated with a learning rate scaled by a factor in $\{0.5, 1, 2\}$.

### F.2 MC Dropout V-MoEs

For MC dropout V-MoE, we take the available fine-tuned V-MoEs and enable dropout at prediction time. Indeed, as described in Table 5, all V-MoE models already have a 0.1 dropout rate in the experts.

### F.3 MIMO V-MoEs

Following Havasi et al. (2020) we consider two MIMO-specific hyperparameters, in addition to the hyperparameters listed in Table 5:

- **Input replication probability:** $\{0.5, 0.625, 0.75\}$
- **Batch repetitions:** $\{1, 2\}$

Our preliminary investigations also considered lower input repetition probabilities and higher batch repetitions. However, lower input repetition probabilities tended to result in poorer performance. While higher batch repetitions did help to some extent, the additional computational cost made it impractical.

Given the surprising result that an ensemble size of $M = 2$ provides no performance improvement over the standard V-MoE and that increasing $M$ further provides worse performance, there seems to be some incompatibility between MIMO and V-MoE. In fact, our investigations revealed that ViT is the source of the problems since applying MIMO to vanilla ViT without experts resulted in the same trends as for V-MoE. Thus we hypothesise that the differences between ViT and ResNet—the architecture to which MIMO was originally applied by Havasi et al. (2020)—are responsible for MIMO's poor performance when applied to ViT.

**Difference 1: Class token.** One of the differences between ViT and ResNet is that ViT makes use of a special learnable class token to classify an image (see Dosovitskiy et al. (2021) for details). ResNet on the other hand makes use of the representation from an entire image for classification. We tried two strategies to mitigate this difference:

1. We applied the global average pooling (GAP) and multi-head attention pooling (MAP) classification strategies introduced in Dosovitskiy et al. (2021) and Zhai et al. (2021), respectively. In short, both of these methods make use of all the tokens from an image for classification. However, neither of these strategies made a significant difference to the relative performance of MIMO and ViT. In fact, the choice of classification method was the least impactful hyperparameter in our sweep.
2. Rather than learning a single class token, we learnt $M$ class tokens. This strategy resulted in MIMO with $M = 2$ outperforming ViT. However, for $M > 2$ the improvement was small enough that ViT still outperformed MIMO.

**Difference 2: Attention.** The other major difference between ViT and ResNet is the building block for each model. While ResNets are primarily composed of convolution operations, ViT makes heavy used of attention. We hypothesised that attention is less suited to separating the information for $M$ input images stored in the channel dimension of a single image. We tried two strategies to mitigate this potential issue:

1. We applied the hybrid architecture, described in Dosovitskiy et al. (2021), in which the input sequence to ViT is formed by CNN feature maps. We used ResNet14 and ResNet50. In both cases, we found that strategy boosted the performance of ViT and MIMO equally.
2. Rather than concatenating images in the channel dimension, we concatenated them in the width dimension, resulting in 3 times as many patches for ViT to process. This strategy was successful in the sense that the MIMO performance for $M > 2$ improved significantly. However, the significant additional computational cost made it an infeasible solution.

Our findings suggest that MIMO and ViT are indeed somewhat incompatible. Unfortunately, none of our proposed solutions to this problem provided high enough predictive performance increases (or indeed low enough computational cost increases in some cases) to warrant immediate further investigation.

# G   UPSTREAM & DOWNSTREAM VERSUS DOWNSTREAM-ONLY ENSEMBLES

In Section 5, and Appendix H include *downstream* deep ensembles (down-DE) of V-MoE, and in some cases ViT, as a baseline. This choice was motivated by the fact that like ViT, V-MoE, and pBE, down-DE requires a only a single upstream checkpoint, which all of the methods more comparable. However, it is clear that using different upstream checkpoints and then further fine-tuning each of these with different random seeds to construct an *upstream* deep ensemble (up-DE) would result in more varied ensemble members and as a result, a better performing ensemble. This idea has recently been explored by Mustafa et al. (2020).

Thus, for completeness, we also investigate the effects of upstream ensembling on V-MoE. Table 10 compares the performance of upstream and downstream V-MoE ($K = 1$) ensembles of sizes $M = 2$ and $M = 4$. Across the range of metrics, for both ImageNet and ImageNet-C, for all ViT families, and for both values of $M$, we see that up-DE outperforms down-DE. In fact, up-DE with $M = 2$ is very often better than or equal to down-DE with $M = 4$. This is especially true for the diversity metrics, which indicates that diversity is indeed the driver for improved performance in up-DE. Not shown in the table is the very large computational cost associated with training upstream ensembles.

Table 10: Comparison of upstream and downstream ensembles of V-MoE with ($K = 1$).

| | | M | IMAGENET NLL ↓ | ERROR ↓ | ECE ↓ | KL ↑ | COS. SIM. ↓ | NORM. DIS. ↑ | IMAGENET-C NLL ↓ | ERROR ↓ | ECE ↓ | KL ↑ | COS. SIM. ↓ | NORM. DIS. ↑ |
|---|---|---|---|---|---|---|---|---|---|---|---|---|---|---|
| H/14 | down-DE | 2 | $0.403_{\pm 0.000}$ | $11.35_{\pm 0.05}$ | $0.018_{\pm 0.001}$ | $0.079_{\pm 0.003}$ | $0.974_{\pm 0.001}$ | $0.488_{\pm 0.006}$ | $0.871_{\pm 0.012}$ | $21.37_{\pm 0.20}$ | $\mathbf{0.021}_{\pm 0.001}$ | $0.218_{\pm 0.002}$ | $0.925_{\pm 0.001}$ | $0.628_{\pm 0.003}$ |
| | up-DE | 2 | $0.391_{\pm 0.000}$ | $11.12_{\pm 0.10}$ | $0.016_{\pm 0.001}$ | $\mathbf{0.126}_{\pm 0.008}$ | $\mathbf{0.963}_{\pm 0.002}$ | $0.625_{\pm 0.012}$ | $0.839_{\pm 0.011}$ | $20.66_{\pm 0.22}$ | $0.022_{\pm 0.000}$ | $\mathbf{0.355}_{\pm 0.007}$ | $\mathbf{0.892}_{\pm 0.002}$ | $0.809_{\pm 0.006}$ |
| | down-DE | 4 | $0.392_{\pm 0.000}$ | $11.20_{\pm 0.000}$ | $0.014_{\pm 0.000}$ | $0.083_{\pm 0.000}$ | $0.973_{\pm 0.000}$ | $0.509_{\pm 0.000}$ | $0.851_{\pm 0.000}$ | $20.97_{\pm 0.000}$ | $\mathbf{0.021}_{\pm 0.000}$ | $0.221_{\pm 0.000}$ | $0.923_{\pm 0.000}$ | $0.650_{\pm 0.000}$ |
| | up-DE | 4 | $\mathbf{0.375}_{\pm 0.000}$ | $\mathbf{10.66}_{\pm 0.000}$ | $\mathbf{0.013}_{\pm 0.000}$ | $\mathbf{0.129}_{\pm 0.000}$ | $0.963_{\pm 0.000}$ | $\mathbf{0.652}_{\pm 0.000}$ | $\mathbf{0.792}_{\pm 0.000}$ | $\mathbf{19.61}_{\pm 0.000}$ | $0.032_{\pm 0.000}$ | $\mathbf{0.361}_{\pm 0.000}$ | $0.892_{\pm 0.000}$ | $\mathbf{0.850}_{\pm 0.000}$ |
| L/16 | down-DE | 2 | $0.450_{\pm 0.002}$ | $12.62_{\pm 0.04}$ | $0.016_{\pm 0.000}$ | $0.061_{\pm 0.001}$ | $0.979_{\pm 0.000}$ | $0.419_{\pm 0.002}$ | $1.010_{\pm 0.006}$ | $24.43_{\pm 0.12}$ | $\mathbf{0.021}_{\pm 0.000}$ | $0.168_{\pm 0.002}$ | $0.936_{\pm 0.001}$ | $0.539_{\pm 0.003}$ |
| | up-DE | 2 | $0.434_{\pm 0.000}$ | $12.23_{\pm 0.04}$ | $\mathbf{0.014}_{\pm 0.000}$ | $\mathbf{0.118}_{\pm 0.001}$ | $\mathbf{0.964}_{\pm 0.000}$ | $0.584_{\pm 0.001}$ | $0.961_{\pm 0.001}$ | $23.46_{\pm 0.03}$ | $0.023_{\pm 0.000}$ | $\mathbf{0.342}_{\pm 0.001}$ | $\mathbf{0.890}_{\pm 0.000}$ | $0.766_{\pm 0.001}$ |
| | down-DE | 4 | $0.440_{\pm 0.002}$ | $12.39_{\pm 0.06}$ | $0.015_{\pm 0.000}$ | $0.061_{\pm 0.001}$ | $0.979_{\pm 0.000}$ | $0.425_{\pm 0.002}$ | $0.983_{\pm 0.006}$ | $23.95_{\pm 0.12}$ | $\mathbf{0.020}_{\pm 0.000}$ | $0.166_{\pm 0.001}$ | $0.937_{\pm 0.001}$ | $0.547_{\pm 0.002}$ |
| | up-DE | 4 | $\mathbf{0.418}_{\pm 0.000}$ | $\mathbf{11.86}_{\pm 0.01}$ | $\mathbf{0.013}_{\pm 0.000}$ | $\mathbf{0.118}_{\pm 0.000}$ | $0.964_{\pm 0.000}$ | $\mathbf{0.603}_{\pm 0.001}$ | $\mathbf{0.916}_{\pm 0.001}$ | $\mathbf{22.45}_{\pm 0.02}$ | $0.034_{\pm 0.000}$ | $\mathbf{0.341}_{\pm 0.000}$ | $0.890_{\pm 0.000}$ | $\mathbf{0.800}_{\pm 0.001}$ |
| L/32 | down-DE | 2 | $0.533_{\pm 0.002}$ | $14.55_{\pm 0.04}$ | $0.025_{\pm 0.001}$ | $0.092_{\pm 0.001}$ | $0.969_{\pm 0.000}$ | $0.479_{\pm 0.004}$ | $1.184_{\pm 0.003}$ | $27.98_{\pm 0.04}$ | $0.029_{\pm 0.000}$ | $0.199_{\pm 0.002}$ | $0.925_{\pm 0.001}$ | $0.556_{\pm 0.002}$ |
| | up-DE | 2 | $0.511_{\pm 0.001}$ | $14.07_{\pm 0.02}$ | $0.019_{\pm 0.000}$ | $\mathbf{0.191}_{\pm 0.001}$ | $\mathbf{0.945}_{\pm 0.000}$ | $0.694_{\pm 0.005}$ | $1.133_{\pm 0.002}$ | $26.97_{\pm 0.04}$ | $\mathbf{0.022}_{\pm 0.000}$ | $\mathbf{0.449}_{\pm 0.005}$ | $\mathbf{0.861}_{\pm 0.001}$ | $0.820_{\pm 0.003}$ |
| | down-DE | 4 | $0.518_{\pm 0.002}$ | $14.29_{\pm 0.03}$ | $0.022_{\pm 0.000}$ | $0.092_{\pm 0.001}$ | $0.969_{\pm 0.000}$ | $0.487_{\pm 0.003}$ | $1.154_{\pm 0.004}$ | $27.47_{\pm 0.05}$ | $0.023_{\pm 0.000}$ | $0.199_{\pm 0.002}$ | $0.925_{\pm 0.001}$ | $0.567_{\pm 0.002}$ |
| | up-DE | 4 | $\mathbf{0.486}_{\pm 0.000}$ | $\mathbf{13.52}_{\pm 0.02}$ | $\mathbf{0.016}_{\pm 0.000}$ | $\mathbf{0.190}_{\pm 0.001}$ | $0.946_{\pm 0.000}$ | $\mathbf{0.722}_{\pm 0.001}$ | $\mathbf{1.073}_{\pm 0.001}$ | $\mathbf{25.74}_{\pm 0.02}$ | $0.030_{\pm 0.000}$ | $\mathbf{0.446}_{\pm 0.001}$ | $0.862_{\pm 0.000}$ | $\mathbf{0.857}_{\pm 0.000}$ |
| B/16 | down-DE | 2 | $0.519_{\pm 0.002}$ | $14.09_{\pm 0.02}$ | $0.021_{\pm 0.001}$ | $0.048_{\pm 0.000}$ | $0.982_{\pm 0.000}$ | $0.351_{\pm 0.002}$ | $1.316_{\pm 0.008}$ | $30.02_{\pm 0.18}$ | $0.030_{\pm 0.000}$ | $0.132_{\pm 0.001}$ | $0.943_{\pm 0.000}$ | $0.448_{\pm 0.002}$ |
| | up-DE | 2 | $0.489_{\pm 0.001}$ | $13.40_{\pm 0.03}$ | $\mathbf{0.015}_{\pm 0.000}$ | $\mathbf{0.169}_{\pm 0.002}$ | $\mathbf{0.951}_{\pm 0.000}$ | $0.668_{\pm 0.004}$ | $1.231_{\pm 0.004}$ | $28.41_{\pm 0.09}$ | $\mathbf{0.023}_{\pm 0.000}$ | $\mathbf{0.481}_{\pm 0.006}$ | $\mathbf{0.845}_{\pm 0.001}$ | $0.838_{\pm 0.003}$ |
| | down-DE | 4 | $0.511_{\pm 0.002}$ | $13.95_{\pm 0.03}$ | $0.019_{\pm 0.000}$ | $0.048_{\pm 0.000}$ | $0.982_{\pm 0.000}$ | $0.354_{\pm 0.002}$ | $1.293_{\pm 0.008}$ | $29.67_{\pm 0.18}$ | $0.026_{\pm 0.000}$ | $0.132_{\pm 0.000}$ | $0.943_{\pm 0.000}$ | $0.453_{\pm 0.002}$ |
| | up-DE | 4 | $\mathbf{0.468}_{\pm 0.000}$ | $\mathbf{12.89}_{\pm 0.03}$ | $0.016_{\pm 0.000}$ | $\mathbf{0.168}_{\pm 0.000}$ | $0.951_{\pm 0.000}$ | $\mathbf{0.690}_{\pm 0.001}$ | $\mathbf{1.166}_{\pm 0.002}$ | $\mathbf{27.08}_{\pm 0.05}$ | $0.037_{\pm 0.000}$ | $\mathbf{0.479}_{\pm 0.001}$ | $0.846_{\pm 0.000}$ | $\mathbf{0.879}_{\pm 0.001}$ |
| B/32 | down-DE | 2 | $0.620_{\pm 0.001}$ | $16.44_{\pm 0.04}$ | $0.023_{\pm 0.000}$ | $0.073_{\pm 0.001}$ | $0.973_{\pm 0.000}$ | $0.414_{\pm 0.002}$ | $1.510_{\pm 0.005}$ | $33.79_{\pm 0.08}$ | $0.032_{\pm 0.000}$ | $0.175_{\pm 0.001}$ | $0.925_{\pm 0.000}$ | $0.498_{\pm 0.002}$ |
| | up-DE | 2 | $0.588_{\pm 0.001}$ | $15.74_{\pm 0.05}$ | $\mathbf{0.017}_{\pm 0.001}$ | $\mathbf{0.214}_{\pm 0.001}$ | $\mathbf{0.937}_{\pm 0.000}$ | $0.709_{\pm 0.001}$ | $1.430_{\pm 0.003}$ | $32.37_{\pm 0.05}$ | $\mathbf{0.022}_{\pm 0.000}$ | $\mathbf{0.537}_{\pm 0.002}$ | $\mathbf{0.824}_{\pm 0.001}$ | $0.844_{\pm 0.002}$ |
| | down-DE | 4 | $0.607_{\pm 0.000}$ | $16.17_{\pm 0.02}$ | $0.021_{\pm 0.001}$ | $0.073_{\pm 0.000}$ | $0.973_{\pm 0.000}$ | $0.418_{\pm 0.005}$ | $1.483_{\pm 0.008}$ | $33.36_{\pm 0.13}$ | $0.027_{\pm 0.000}$ | $0.174_{\pm 0.001}$ | $0.926_{\pm 0.001}$ | $0.504_{\pm 0.002}$ |
| | up-DE | 4 | $\mathbf{0.561}_{\pm 0.001}$ | $\mathbf{15.10}_{\pm 0.03}$ | $0.020_{\pm 0.000}$ | $\mathbf{0.214}_{\pm 0.001}$ | $0.937_{\pm 0.000}$ | $\mathbf{0.739}_{\pm 0.001}$ | $\mathbf{1.357}_{\pm 0.002}$ | $\mathbf{30.92}_{\pm 0.03}$ | $0.036_{\pm 0.000}$ | $\mathbf{0.537}_{\pm 0.001}$ | $0.824_{\pm 0.000}$ | $\mathbf{0.884}_{\pm 0.001}$ |
| S/32 | down-DE | 2 | $0.807_{\pm 0.003}$ | $20.90_{\pm 0.10}$ | $\mathbf{0.018}_{\pm 0.001}$ | $0.102_{\pm 0.001}$ | $0.962_{\pm 0.000}$ | $0.458_{\pm 0.003}$ | $2.106_{\pm 0.010}$ | $44.52_{\pm 0.18}$ | $0.038_{\pm 0.001}$ | $0.223_{\pm 0.003}$ | $0.900_{\pm 0.001}$ | $0.521_{\pm 0.002}$ |
| | up-DE | 2 | $0.763_{\pm 0.001}$ | $19.85_{\pm 0.04}$ | $\mathbf{0.016}_{\pm 0.000}$ | $\mathbf{0.305}_{\pm 0.002}$ | $\mathbf{0.911}_{\pm 0.000}$ | $0.773_{\pm 0.002}$ | $2.004_{\pm 0.004}$ | $42.92_{\pm 0.08}$ | $\mathbf{0.025}_{\pm 0.000}$ | $\mathbf{0.683}_{\pm 0.003}$ | $\mathbf{0.767}_{\pm 0.001}$ | $0.856_{\pm 0.002}$ |
| | down-DE | 4 | $0.795_{\pm 0.003}$ | $20.66_{\pm 0.13}$ | $\mathbf{0.015}_{\pm 0.001}$ | $0.102_{\pm 0.002}$ | $0.962_{\pm 0.001}$ | $0.462_{\pm 0.004}$ | $2.076_{\pm 0.012}$ | $44.16_{\pm 0.21}$ | $0.031_{\pm 0.000}$ | $0.222_{\pm 0.003}$ | $0.900_{\pm 0.001}$ | $0.526_{\pm 0.003}$ |
| | up-DE | 4 | $\mathbf{0.728}_{\pm 0.001}$ | $\mathbf{19.06}_{\pm 0.04}$ | $0.025_{\pm 0.000}$ | $\mathbf{0.304}_{\pm 0.001}$ | $0.911_{\pm 0.000}$ | $\mathbf{0.808}_{\pm 0.002}$ | $\mathbf{1.914}_{\pm 0.003}$ | $\mathbf{41.38}_{\pm 0.05}$ | $0.034_{\pm 0.000}$ | $\mathbf{0.682}_{\pm 0.003}$ | $0.767_{\pm 0.001}$ | $\mathbf{0.891}_{\pm 0.002}$ |

# H ADDITIONAL EXPERIMENTAL RESULTS

In this section we expand on various experiments presented in sections 3 and 5. In experiments considering multiple ViT families we also include B/16 which was excluded from the main text for clarity.

## H.1 STATIC VERSUS ADAPTIVE COMBINATION

Here we continue the investigation into static versus adaptive combination from Section 3.1.

**Individual gains with respect to $E, K$ and $M$.** Figure 8 shows the effect of increasing the the various 'ensemble size' parameters for a deep ensemble of V-MoEs. In particular, we investigate the static combination axis $M$ (the number of ensemble members), as well as the two adaptive axes—$K$ (the number of experts chosen per patch) and $E$ (the total number of experts).

When investigating the effect of $K$, we fix $E = 32$ and average over $M \in \{1, .., 8\}$. Similarly, when investigating $M$, we fix $E = 32$ and average over $K \in \{1, .., 8\}$. When investigating the effect of $E$ we fix $K = 2$ and average over $M \in \{1, .., 8\}$. As a result of this procedure the exact values of the curves are not directly comparable. However, we can still examine the relative trends.

Specifically, we note that while the variation in $K$ and $M$ curves is roughly of the same size, the variation in the $E$ curve is smaller. We also note that there is very little variation beyond $E = 8$ (note the difference in the scales of the axes for the curves). These observations motivate the design of pBE, where we split the sub-models along the $E$ axis, in order to better take advantage of the experts.

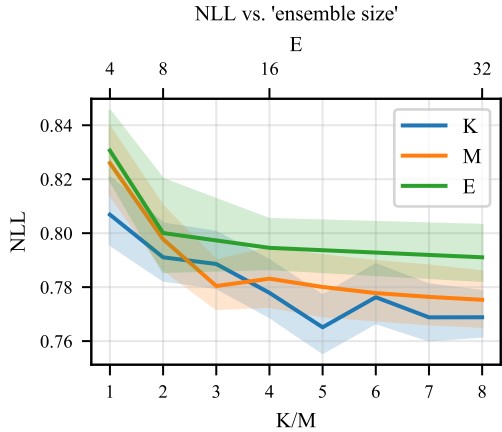

Figure 8: Comparison for the impact on ImageNet NLL of variations in $K$, $E$ and $M$. The underlying model is ViT-S/32.

**Extended Results for the Cumulative Effects of Static and Adaptive Combination.** In Figure 9 we extend the ImageNet NLL results, presented in Figure 3, to a range of other datasets and performance metrics. We see that in most cases, the trends are the same as before. That is, (static) ensembling tends to improve the performance of ViT and V-MoE equally. The two exceptions are ECE and OOD detection for ViT-S/32 where we see that larger ensemble sizes can result in decreased performance. These results indicate that the for small ViT models, larger ensembles can have slightly lower quality uncertainty estimates. The trend for ImageNet-C performance is also not as consistent with ensembling sometimes helping ViT or V-MoE less (as indicated by the changes in ordering on the y-axis).

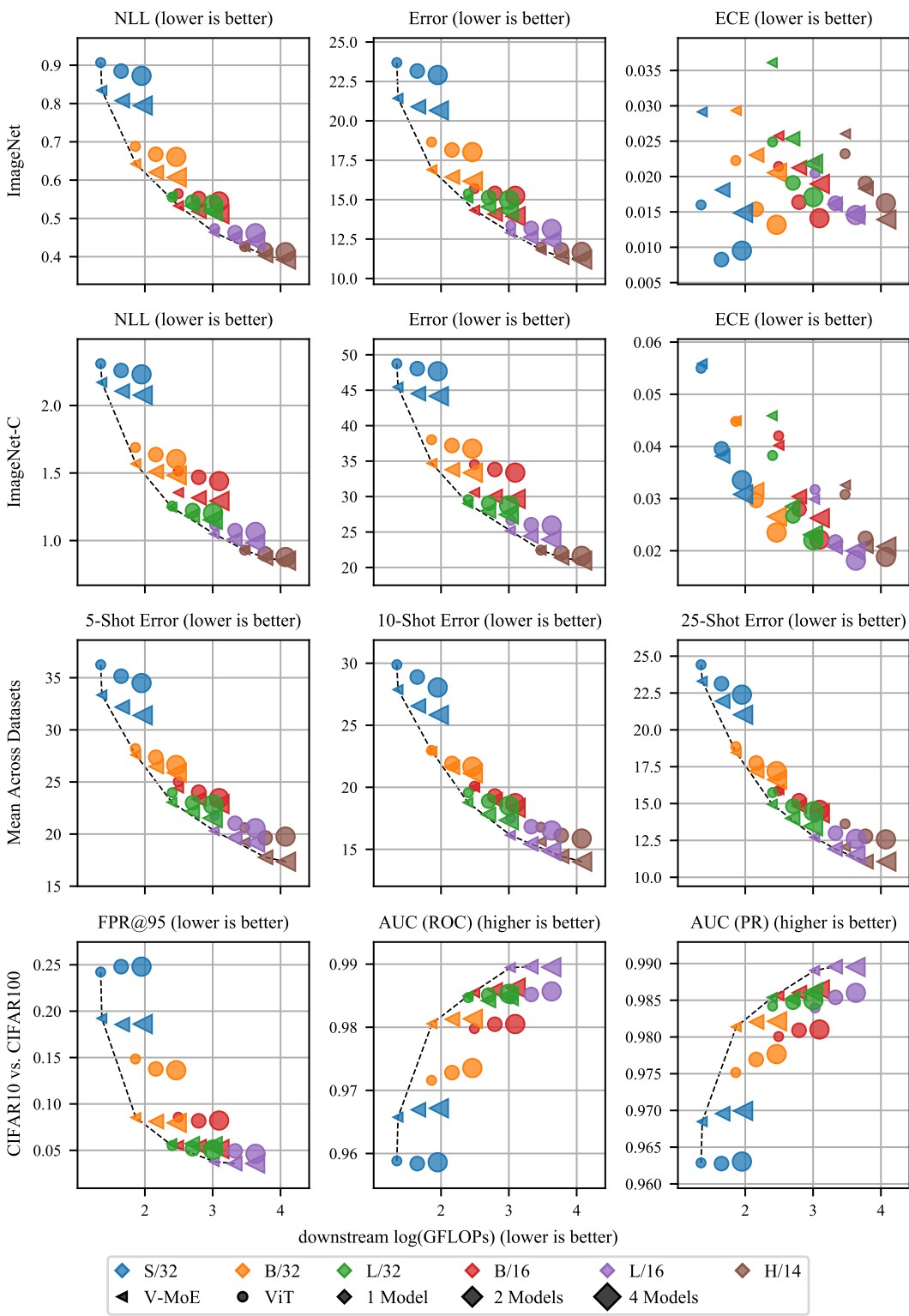

Figure 9: Extended results for Figure 3 to a selection of other tasks and metrics. We see that in most cases, ensembles tend to help ViT and V-MoE equally.

## H.2   EXTENDED RESULTS FOR FEW-SHOT LEARNING

In Figure 10, we extend the few-shot learning results of Figure 4 to also include 1, 5, and 25-shot. Additionally, we show results for the weighted aggregation strategy mentioned in Appendix B.4.

We confirm the result that few-shot performance for pBE gets better, relative to the other baselines, with larger ViT families. Additionally, we see that pBE performance seems to get better, again relative to the other baselines, with more shots. This phenomenon can most easily be noticed by comparing the results for S/32 across the different numbers of shots. Finally, we see that the trends with and without the weighted mean are the same.

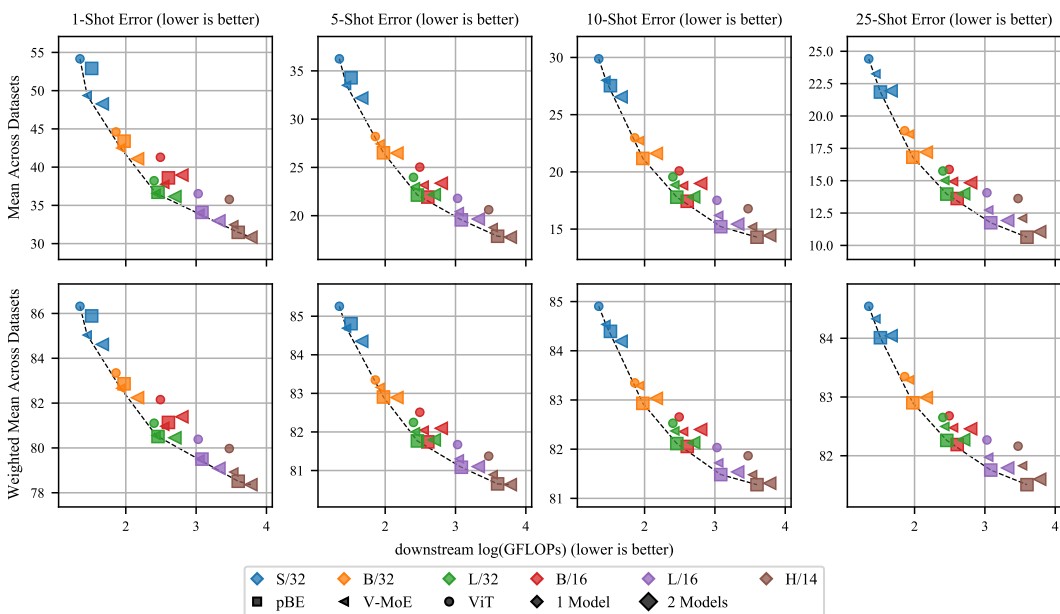

Figure 10: Extended few-shot results from Figure 4 with an additional aggregation method and numbers of shots.

## H.3   EXTENDED RESULTS FOR OOD DETECTION

Here we extended the OOD results of Figure 5. Specifically, we add Cifar100 as an in-distribution dataset and Describable Textures Dataset (DTD) (Cimpoi et al., 2014) as an OOD dataset. We also add area under the receiver operating characteristic (AUC (ROC)) and area under the precision-recall curve (AUC (PR)) as metrics. Figures 11 and 12 contain the results with Cifar10 and Cifar100 as the in-distribution datasets, respectively.

As in Figure 5, we see that pBE performs better (relative to the other baselines) for larger ViT families. Furthermore, pBE seems to perform better in near OOD detection (i.e. Cifar10 versus Cifar100, and vice versa) than far OOD detection. Finally, we see that these trends are consistent across ODD metrics.

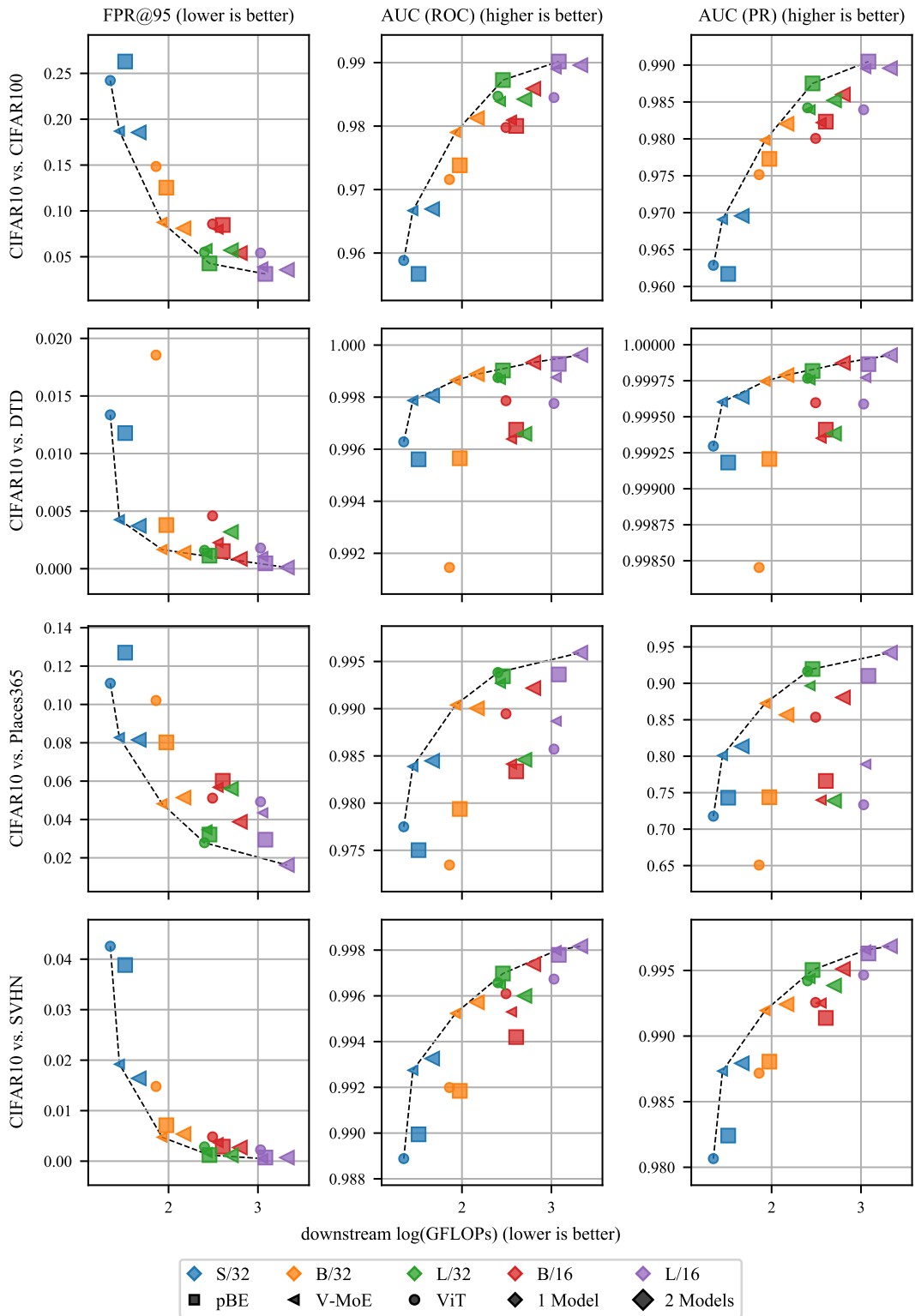

Figure 11: Extended OOD detection the results from Figure 5 with an additional OOD dataset and more metrics.

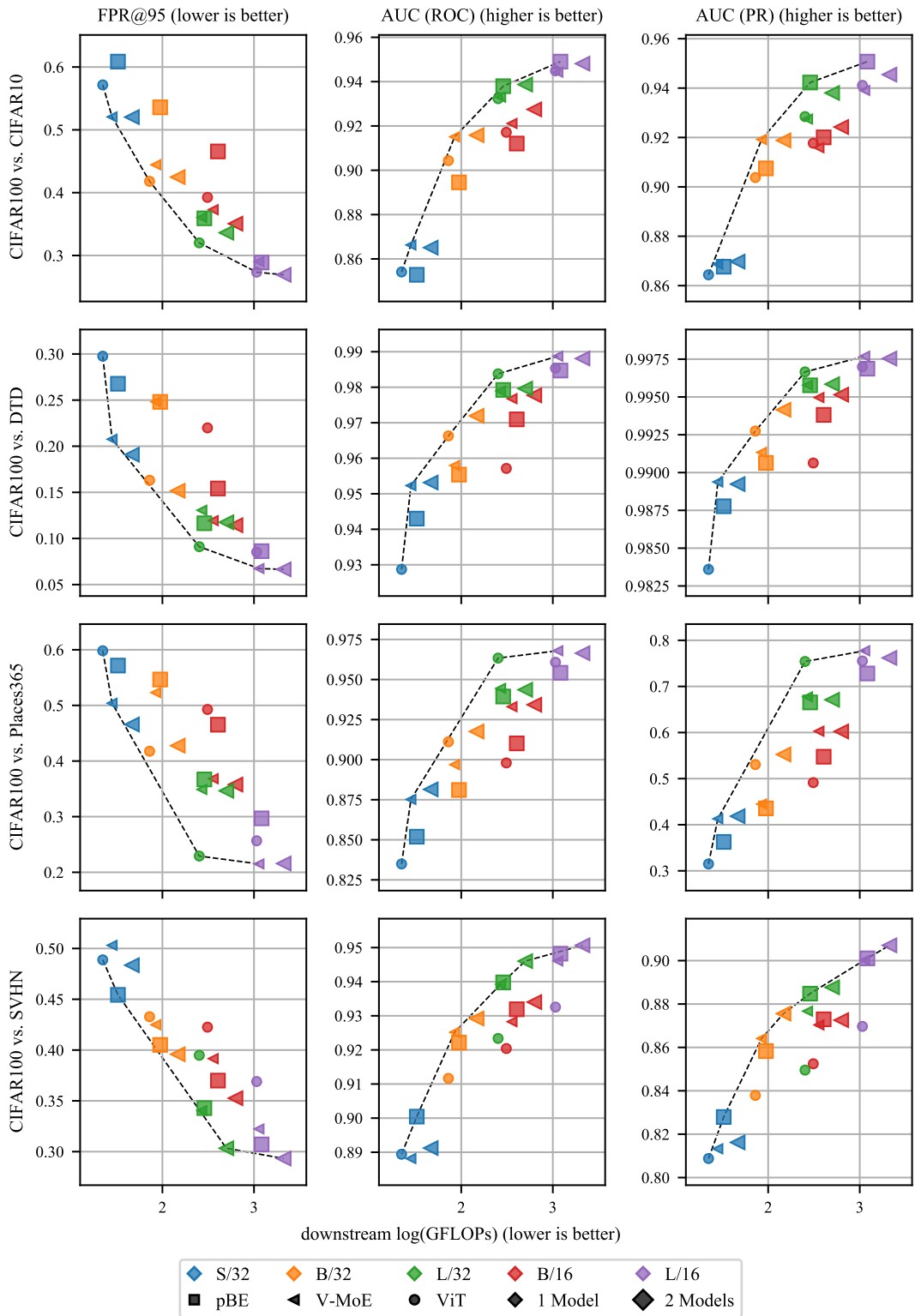

Figure 12: Extended OOD detection the results from Figure 5 with Cifar100 as the in-distribution dataset, an additional OOD dataset, and more metrics.

## H.4 Extended Results for ImageNet

In this section we extend the results for ImageNet and the corrupted variants presented in Figures 4, 5 and 6. In addition to NLL (and ECE for standard ImageNet), Figure 13 provides classification error and ECE for all ImageNet variants.

Most of the trends observed in Section 5 remain true:

- pBE tends to be Pareto efficient in the presence of distribution shift.
- For smaller ViT families, V-MoE outperforms ViT in the presence of distribution shift.
- pBE improves ECE over ViT and V-MoE.
- pBE improves classification performance.
- ViT consistently provides better ECE than V-MoE.

However, there are some exceptions:

- **ImageNet-A classification error.** All models (including pBE) under-perform relative to ViT-S/32 and ViT-H/14.
- **ECE for ImageNet-C, ImageNet-A, and ImageNet-V2.** Interestingly for the nonstandard ImageNet variants, and in particular for ImageNet-A, there is a strong correlation between lower ECE and larger ViT families.

## H.5 Additional Cifar10, Cifar100, Flowers, and Pets Results

Here we extend the results for ImageNet and the corrupted variants presented in Figures 4, 5 and 6 to 4 additional datasets. Figures 14, 15, 16 and 17 provide results for Cifar10, Cifar100, Oxford Flowers 102, and Oxford IIIT Pet, respectively. As in Appendix H.4, we find that the results are similar to those in Section 5.

Compared to ImageNet, for Cifar10, Cifar10-C, and Cifar100, pBE seems to perform even better relative to the other baselines. Note, for example, that pBE is Pareto efficient (even for S/32) in the cases of Cifar10-C and Cifar100 NLL. As in Appendix H.4, we see that the ECE has a stronger downward trend with respect to increased ViT family size for shifted test data.

For Flowers and Pets, where we only have results for smaller ViT families, pBE seems to under perform. However, the performance for L/32 is better than for S/32 and B/32 which suggests that the results for these datasets are consistent with the other datasets presented in this work and, therefore, that we should expect pBE's predictive performance to keep improving with larger models.

## H.6 pBE and V-MoE with larger values of $K$ and $M$

Figure 18 and Figure 19 show the effect of varying $K$ on pBE and V-MoE, and the effect of varying $M$ on pBE, respectively. We make the following observations:

- In almost all cases, increasing $K$ or $M$ does not result in Pareto efficient models.
- For V-MoE, increasing $K$ seems to help in most cases, except for ECE performance where it usually hurts.
- For pBE, going from $K = 1$ to $K = 2$ seems to help in most cases but going from $K = 2$ to $K = 4$ usually hurts. Going from $K = 1$ to $K = 4$ still helps but to a lesser extent than from $K = 1$ to $K = 2$.
- For pBE, increasing $M$ either doesn't make a consistent and significant difference or hurts (e.g. in OOD detection).

These conclusions should, however, be considered with caution. Recall that the upstream checkpoints used for fine-tuning all V-MoE and pBE models in this work are V-MoE models with $K = 2$. Thus, the results in this experiment are confounded by upstream and downstream checkpoint mismatch for all pBE models and all V-MoE models with $K \neq 2$. For example, we hypothesise that it is more difficult to train downstream pBE models with larger values of $M$ from upstream V-MoE models because in each partition some common expert specialisations will need to be duplicated.

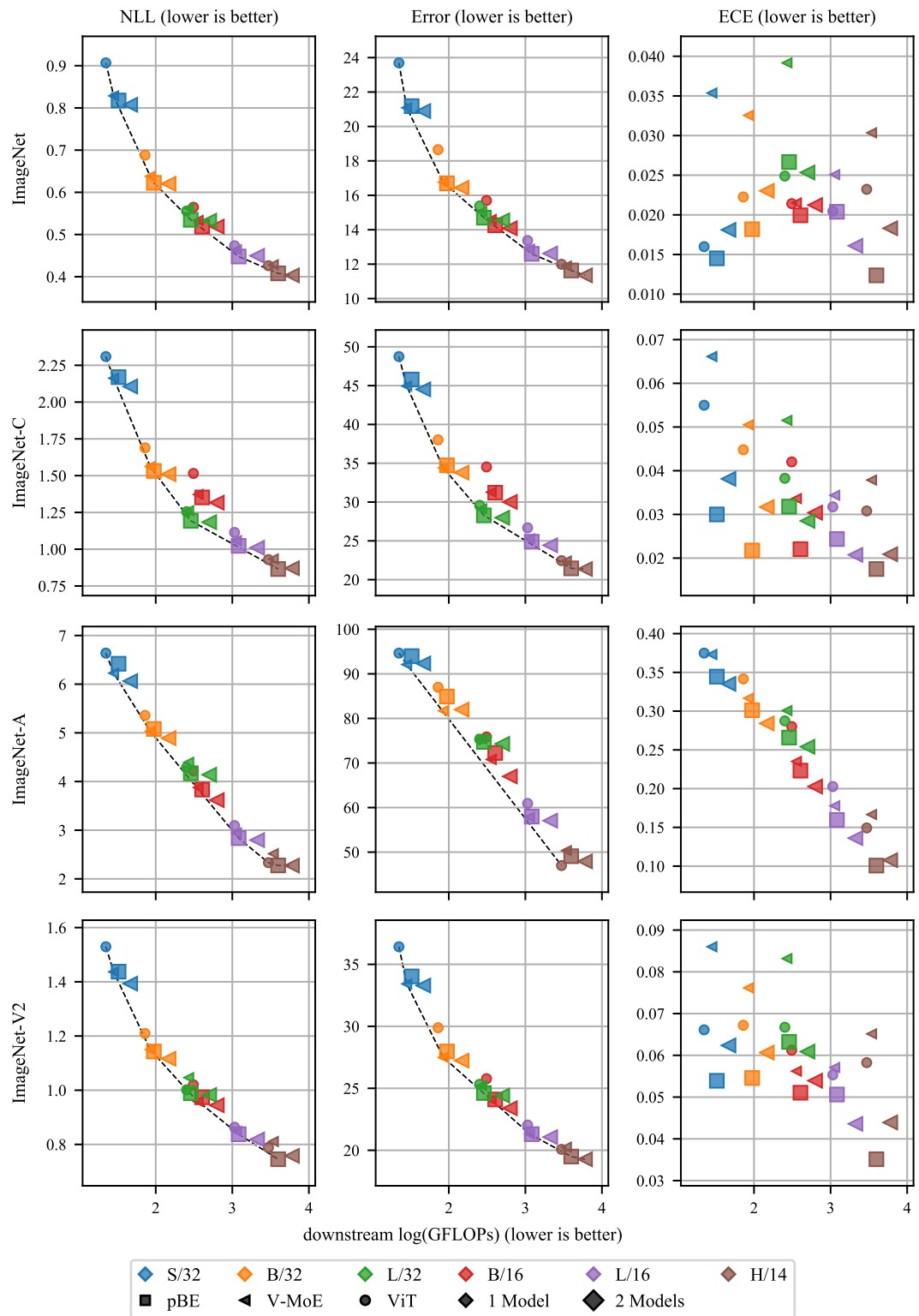

Figure 13: Extended results from Figures 4, 5 and 6 with additional metrics.

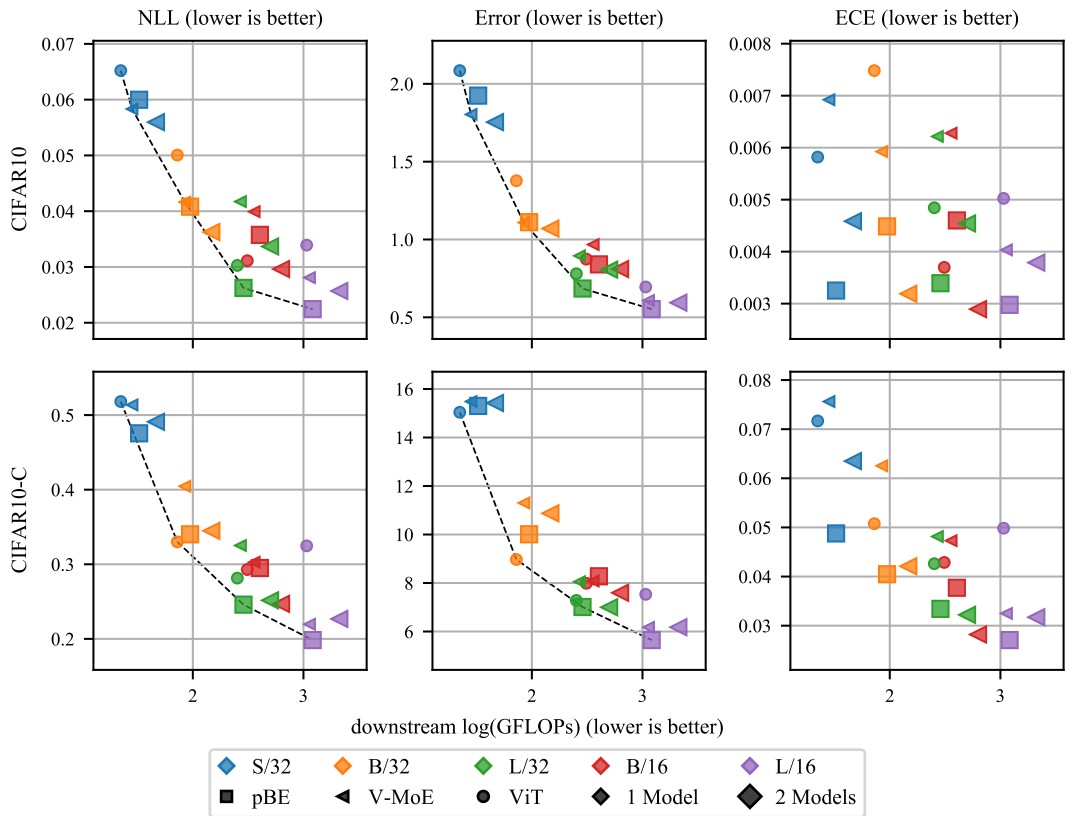

Figure 14: Results for Cifar10 and Cifar10-C.

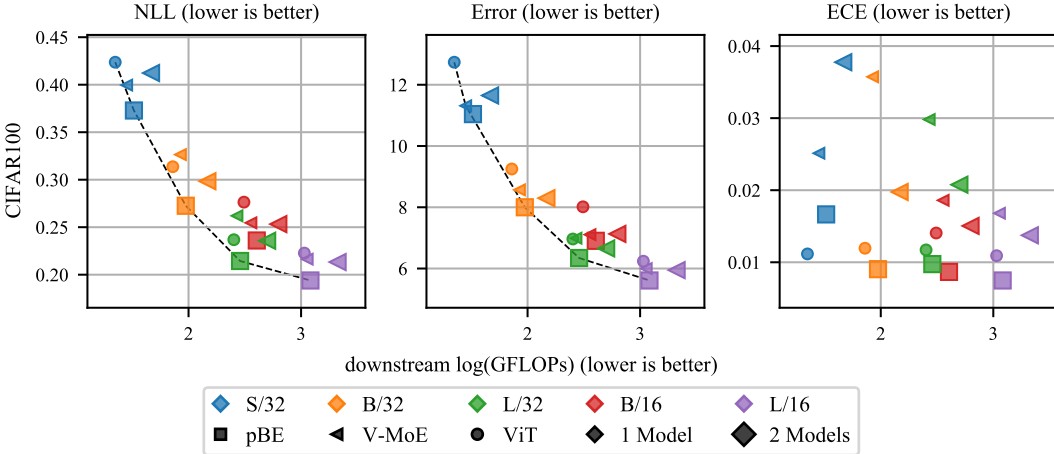

Figure 15: Results for Cifar100.

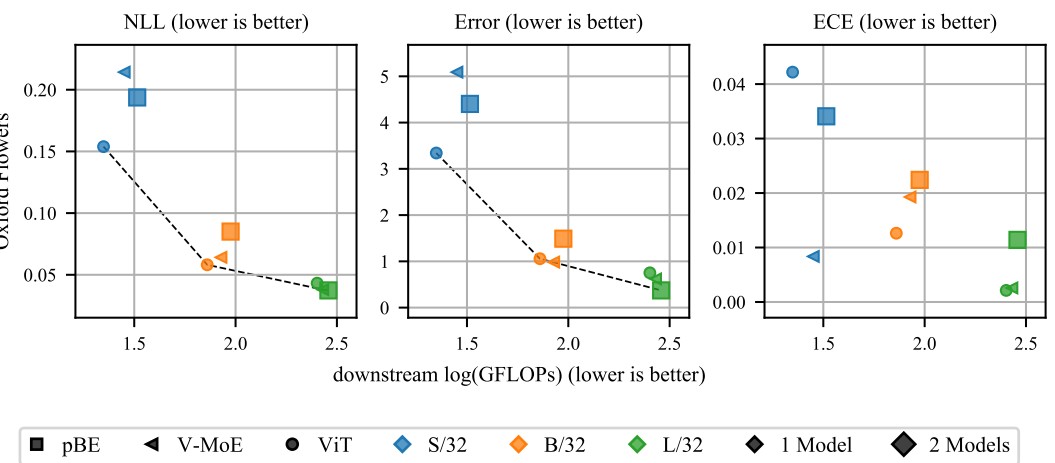

Figure 16: Results for Oxford Flowers 102.

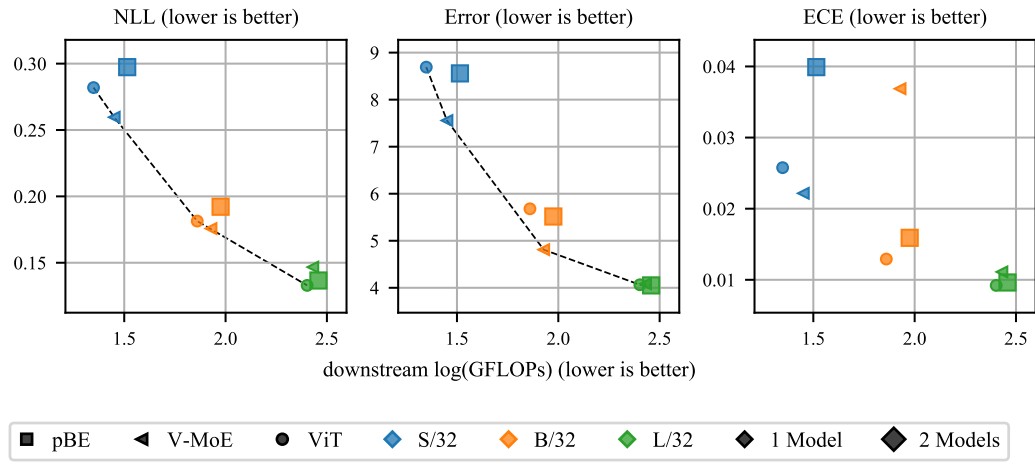

Figure 17: Results for Oxford IIIT Pet.

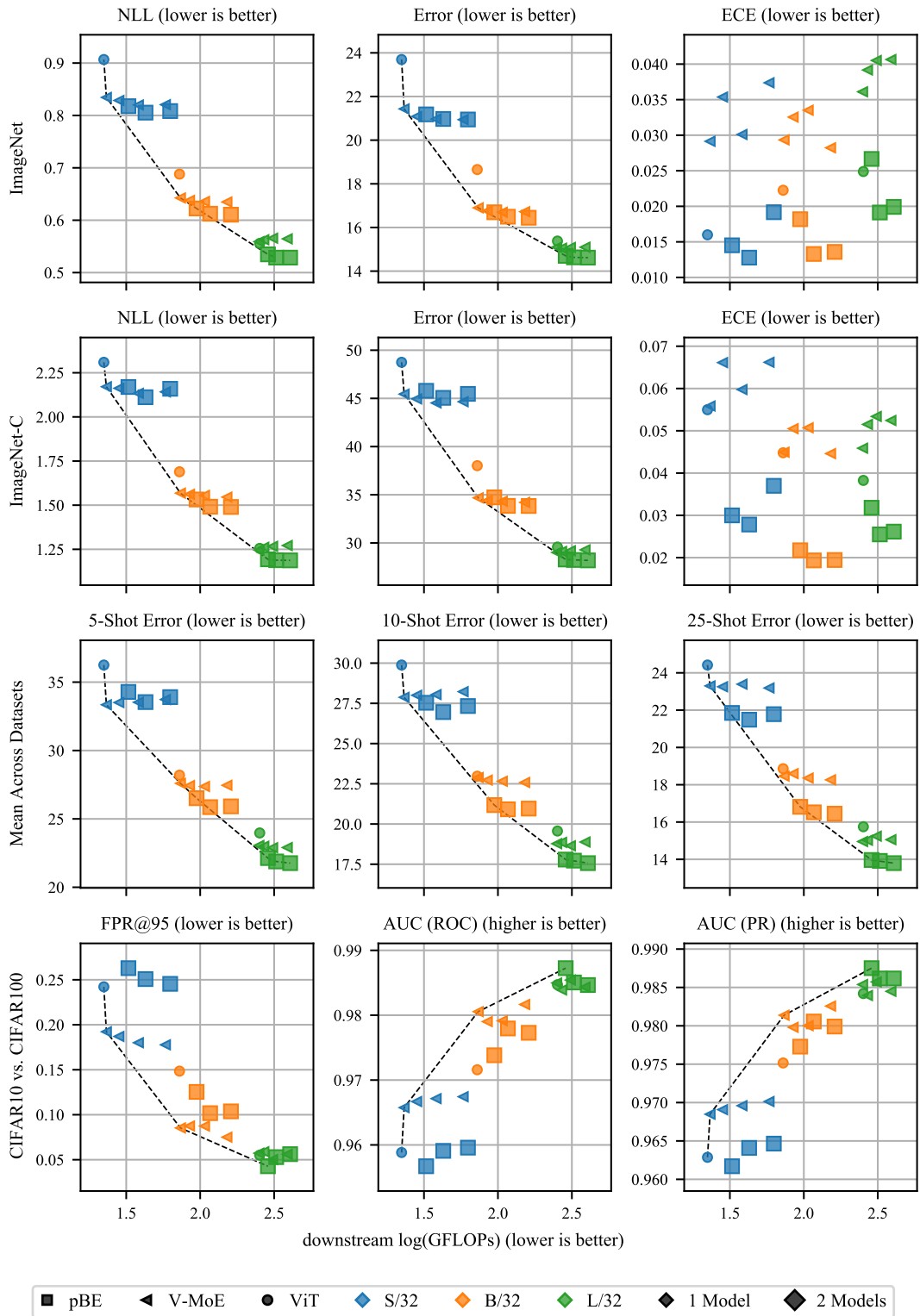

Figure 18: Results for V-MoE with $K \in \{1, 2, 4, 8\}$ and pBE with $K \in \{1, 2, 4\}$. Models with larger values of $K$ have larger FLOPs.

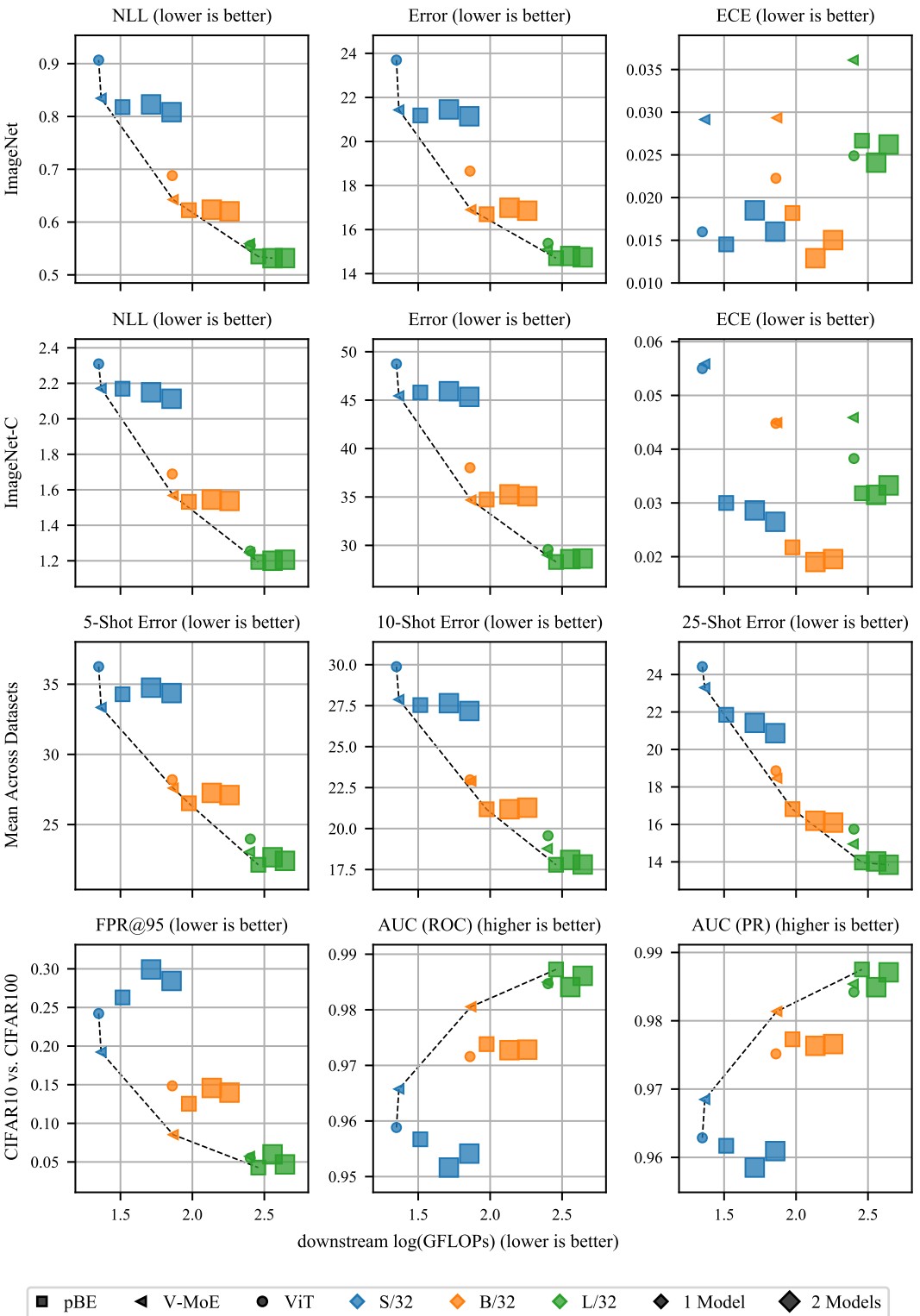

Figure 19: Results for pBE with $M = 4$ and $K \in \{1, 2\}$.

## H.7 FLOPs Numbers

Table 11 provides the downstream training FLOPs for various pBE, V-MoE, and ViT configurations. These numbers correspond to the x-values of the points in the figures presented in Section 5 and Appendix H. Table 12 provides the percentage difference in FLOPs between the pBE, V-MoE and down-DE models most commonly used in this work. Note that the percentage differences for H/14 do not follow the trend of the other sizes, e.g. that the percentage difference between pBE and V-MoE gets smaller for larger sizes, due to the fact that for H/15 we use a last-5 configuration rather than the last-2 configuration used for the other ViT families.

Table 11: Downstream training GFLOPs for the various pBE, V-MoE, and ViT baselines used in this work.

|  | K | M | S/32 | B/32 | B/16 | L/32 | L/16 | H/14 |
|---|---|---|---|---|---|---|---|---|
| pBE | 1 | 2 | 32.71 | 94.51 | 403.92 | 287.14 | 1210.91 | 3967.77 |
| pBE | 1 | 4 | 51.65 | 136.35 | — | 360.62 | — | — |
| pBE | 2 | 2 | 42.69 | 116.79 | 492.62 | 326.41 | 1367.99 | — |
| pBE | 2 | 4 | 71.65 | 181.00 | — | 439.89 | — | — |
| pBE | 4 | 2 | 62.68 | 161.44 | — | 405.67 | — | — |
| ViT | - | 1 | 22.32 | 72.46 | 310.67 | 252.61 | 1069.53 | 2962.57 |
| ViT | - | 2 | 44.63 | 144.93 | 621.34 | 505.22 | 2139.06 | 5925.13 |
| ViT | - | 4 | 89.26 | 289.86 | 1242.68 | 1010.45 | 4278.13 | 11850.27 |
| V-MoE | 1 | 1 | 23.22 | 73.55 | 313.95 | 249.70 | 1055.14 | 3008.02 |
| V-MoE | 1 | 2 | 46.44 | 147.10 | 627.89 | 499.39 | 2110.29 | 6016.04 |
| V-MoE | 1 | 4 | 92.87 | 294.19 | 1255.79 | 998.78 | 4220.58 | 12032.08 |
| V-MoE | 2 | 1 | 28.23 | 84.74 | 358.70 | 269.51 | 1134.41 | 3436.74 |
| V-MoE | 4 | 1 | 38.20 | 107.01 | 447.39 | 308.96 | 1291.49 | — |
| V-MoE | 8 | 1 | 58.20 | 151.67 | — | 388.05 | — | — |

Table 12: Percentage difference in downstream training FLOPs for pBE with $(K, M) = (1, 2)$ compared with V-MoE with $K = 1$ and an ensemble of two such V-MoE members.

|  | S/32 | B/32 | B/16 | L/32 | L/16 | H/14 |
|---|---|---|---|---|---|---|
| pBE vs. V-MoE | 40.88 | 28.51 | 28.66 | 14.99 | 14.76 | 31.91 |
| pBE vs. down-DE | -29.56 | -35.75 | -35.67 | -42.50 | -42.62 | -34.05 |

## H.8 Auxiliary Tables with Standard Errors.

The tables in this section provide the mean values and corresponding standard errors for many of the results depicted in figures throughout Section 5 and Appendix H.

Table 13: ImageNet comparison of V-MoE, downstream ensembles there-of, and pBE with 2 experts per input in each case.

| | | K | M | NLL ↓ (ImageNet) | Error ↓ (ImageNet) | ECE ↓ (ImageNet) | NLL ↓ (ImageNet-C) | Error ↓ (ImageNet-C) | ECE ↓ (ImageNet-C) | NLL ↓ (ImageNet-A) | Error ↓ (ImageNet-A) | ECE ↓ (ImageNet-A) | NLL ↓ (ImageNet-V2) | Error ↓ (ImageNet-V2) | ECE ↓ (ImageNet-V2) | Downstream ΔFLOPs (%) ↓ |
|---|---|---|---|---|---|---|---|---|---|---|---|---|---|---|---|---|
| H/14 | pBE | 1 | 2 | 0.408 ±0.001 | 11.63 ±0.05 | **0.012** ±0.000 | **0.865** ±0.010 | **21.46** ±0.16 | **0.018** ±0.000 | **2.276** ±0.042 | 49.09 ±0.78 | **0.101** ±0.003 | **0.745** ±0.001 | 19.50 ±0.08 | **0.035** ±0.001 | **15.45** |
| | down-DE | 1 | 2 | **0.403** ±0.000 | **11.35** ±0.05 | 0.018 ±0.001 | 0.871 ±0.012 | 21.37 ±0.20 | 0.021 ±0.001 | 2.273 ±0.049 | **47.93** ±0.93 | 0.108 ±0.001 | 0.758 ±0.003 | **19.28** ±0.17 | 0.044 ±0.001 | 75.05 |
| | V-MoE | 2 | 1 | 0.428 ±0.003 | 11.89 ±0.13 | 0.030 ±0.001 | 0.934 ±0.013 | 22.41 ±0.20 | 0.038 ±0.001 | 2.517 ±0.063 | 50.34 ±1.12 | 0.167 ±0.003 | 0.811 ±0.005 | 20.25 ±0.13 | 0.065 ±0.001 | — |
| L/16 | pBE | 1 | 2 | **0.448** ±0.001 | **12.60** ±0.03 | 0.020 ±0.000 | **1.023** ±0.005 | 24.89 ±0.11 | 0.024 ±0.000 | **2.836** ±0.015 | 57.97 ±0.21 | 0.160 ±0.001 | 0.838 ±0.002 | 21.30 ±0.07 | 0.051 ±0.001 | **6.74** |
| | down-DE | 1 | 2 | 0.450 ±0.002 | 12.62 ±0.04 | **0.016** ±0.000 | 1.010 ±0.006 | 24.43 ±0.12 | **0.021** ±0.000 | 2.796 ±0.016 | 57.06 ±0.32 | **0.136** ±0.000 | **0.818** ±0.003 | 21.06 ±0.11 | **0.044** ±0.001 | 86.03 |
| | V-MoE | 2 | 1 | 0.464 ±0.001 | 12.88 ±0.04 | 0.025 ±0.000 | 1.058 ±0.004 | 25.27 ±0.08 | 0.034 ±0.000 | 2.945 ±0.012 | 57.85 ±0.21 | 0.178 ±0.001 | 0.848 ±0.002 | 21.33 ±0.03 | 0.057 ±0.001 | — |
| L/32 | pBE | 1 | 2 | **0.535** ±0.001 | 14.70 ±0.03 | **0.027** ±0.000 | 1.193 ±0.003 | 28.28 ±0.05 | 0.032 ±0.000 | 4.170 ±0.010 | 74.73 ±0.09 | 0.266 ±0.002 | 0.989 ±0.002 | 24.62 ±0.08 | **0.063** ±0.001 | **6.54** |
| | down-DE | 1 | 2 | 0.533 ±0.002 | **14.55** ±0.04 | 0.025 ±0.001 | **1.184** ±0.003 | **27.98** ±0.04 | **0.029** ±0.000 | **4.139** ±0.017 | **74.29** ±0.21 | **0.254** ±0.003 | **0.982** ±0.002 | **24.42** ±0.08 | **0.061** ±0.002 | 85.29 |
| | V-MoE | 2 | 1 | 0.563 ±0.001 | 15.05 ±0.03 | 0.039 ±0.000 | 1.261 ±0.005 | 29.12 ±0.07 | 0.052 ±0.000 | 4.394 ±0.014 | 75.39 ±0.17 | 0.301 ±0.002 | 1.046 ±0.002 | 25.22 ±0.06 | 0.083 ±0.001 | — |
| B/16 | pBE | 1 | 2 | **0.519** ±0.001 | 14.26 ±0.03 | **0.020** ±0.000 | 1.352 ±0.005 | 31.20 ±0.09 | **0.022** ±0.000 | 3.835 ±0.019 | 72.25 ±0.22 | 0.223 ±0.002 | 0.974 ±0.002 | 24.10 ±0.10 | **0.051** ±0.001 | **12.61** |
| | down-DE | 1 | 2 | **0.519** ±0.002 | **14.09** ±0.02 | 0.021 ±0.001 | **1.316** ±0.008 | **30.02** ±0.18 | 0.030 ±0.000 | **3.618** ±0.013 | **66.99** ±0.28 | **0.203** ±0.002 | **0.945** ±0.003 | **23.39** ±0.09 | 0.054 ±0.001 | 75.05 |
| | V-MoE | 2 | 1 | 0.533 ±0.001 | 14.60 ±0.04 | 0.022 ±0.000 | 1.372 ±0.005 | 31.27 ±0.11 | 0.034 ±0.000 | 3.875 ±0.016 | 70.79 ±0.30 | 0.235 ±0.002 | 0.959 ±0.003 | 24.09 ±0.11 | 0.056 ±0.001 | — |
| B/32 | pBE | 1 | 2 | 0.622 ±0.001 | 16.70 ±0.03 | **0.018** ±0.000 | 1.532 ±0.005 | 34.73 ±0.08 | **0.022** ±0.000 | 5.080 ±0.013 | 84.91 ±0.09 | 0.301 ±0.001 | 1.143 ±0.002 | 27.97 ±0.10 | **0.055** ±0.001 | **11.54** |
| | down-DE | 1 | 2 | **0.620** ±0.001 | **16.44** ±0.04 | 0.023 ±0.000 | **1.510** ±0.005 | **33.79** ±0.08 | 0.032 ±0.000 | **4.891** ±0.018 | **81.98** ±0.18 | **0.284** ±0.001 | **1.116** ±0.002 | **27.24** ±0.08 | 0.061 ±0.000 | 73.59 |
| | V-MoE | 2 | 1 | 0.638 ±0.001 | 16.76 ±0.05 | 0.033 ±0.001 | 1.562 ±0.004 | 34.40 ±0.08 | 0.050 ±0.001 | 5.032 ±0.014 | 81.71 ±0.11 | 0.317 ±0.002 | 1.150 ±0.002 | 27.50 ±0.08 | 0.076 ±0.001 | — |
| S/32 | pBE | 1 | 2 | **0.818** ±0.002 | 21.18 ±0.04 | **0.015** ±0.000 | 2.169 ±0.008 | 45.79 ±0.10 | **0.030** ±0.000 | 6.419 ±0.011 | 93.98 ±0.09 | 0.345 ±0.001 | 1.437 ±0.002 | 34.03 ±0.05 | **0.054** ±0.001 | **15.88** |
| | down-DE | 1 | 2 | 0.807 ±0.003 | 20.90 ±0.10 | 0.018 ±0.001 | **2.106** ±0.010 | **44.52** ±0.18 | 0.038 ±0.001 | **6.063** ±0.007 | **92.33** ±0.07 | **0.335** ±0.001 | **1.393** ±0.003 | 33.29 ±0.10 | 0.062 ±0.002 | 64.50 |
| | V-MoE | 2 | 1 | 0.829 ±0.002 | **21.09** ±0.08 | 0.035 ±0.001 | 2.162 ±0.007 | 44.95 ±0.12 | 0.066 ±0.001 | 6.227 ±0.015 | 92.09 ±0.12 | 0.373 ±0.001 | 1.437 ±0.002 | 33.42 ±0.08 | 0.086 ±0.001 | — |

Table 14: ImageNet comparison of V-MoE, downstream ensembles there-of, and pBE with 4 experts per input in each case.

| | | K | M | NLL ↓ (ImageNet) | Error ↓ (ImageNet) | ECE ↓ (ImageNet) | NLL ↓ (ImageNet-C) | Error ↓ (ImageNet-C) | ECE ↓ (ImageNet-C) | NLL ↓ (ImageNet-A) | Error ↓ (ImageNet-A) | ECE ↓ (ImageNet-A) | NLL ↓ (ImageNet-V2) | Error ↓ (ImageNet-V2) | ECE ↓ (ImageNet-V2) | Downstream ΔFLOPs (%) ↓ |
|---|---|---|---|---|---|---|---|---|---|---|---|---|---|---|---|---|
| L/16 | pBE | 2 | 2 | 0.451 ±0.001 | 12.61 ±0.04 | 0.023 ±0.000 | 1.028 ±0.003 | 24.90 ±0.07 | 0.028 ±0.000 | 2.886 ±0.016 | 58.26 ±0.24 | 0.171 ±0.002 | 0.849 ±0.002 | 21.40 ±0.11 | 0.055 ±0.001 | **5.92** |
| | down-DE | 1 | 4 | **0.440** ±0.002 | **12.39** ±0.06 | **0.015** ±0.000 | **0.983** ±0.006 | **23.95** ±0.12 | **0.020** ±0.000 | **2.712** ±0.015 | **56.36** ±0.36 | **0.120** ±0.002 | **0.803** ±0.002 | **20.83** ±0.14 | **0.039** ±0.001 | 226.80 |
| | V-MoE | 4 | 1 | 0.465 ±0.002 | 12.86 ±0.05 | 0.026 ±0.000 | 1.060 ±0.006 | 25.28 ±0.09 | 0.035 ±0.000 | 2.976 ±0.022 | 58.26 ±0.19 | 0.186 ±0.002 | 0.856 ±0.002 | 21.53 ±0.07 | 0.060 ±0.001 | — |
| L/32 | pBE | 2 | 2 | 0.529 ±0.001 | 14.63 ±0.02 | **0.019** ±0.000 | 1.188 ±0.004 | 28.23 ±0.05 | 0.025 ±0.000 | 4.095 ±0.015 | 74.38 ±0.14 | 0.253 ±0.002 | **0.962** ±0.001 | 24.41 ±0.04 | 0.055 ±0.001 | **5.65** |
| | down-DE | 1 | 4 | **0.518** ±0.002 | **14.29** ±0.03 | 0.022 ±0.000 | **1.154** ±0.004 | **27.47** ±0.05 | **0.023** ±0.000 | **4.033** ±0.016 | **73.79** ±0.13 | **0.237** ±0.002 | 0.959 ±0.002 | **24.03** ±0.04 | **0.054** ±0.001 | 223.27 |
| | V-MoE | 4 | 1 | 0.566 ±0.002 | 15.08 ±0.04 | 0.041 ±0.000 | 1.267 ±0.003 | 29.19 ±0.05 | 0.053 ±0.000 | 4.428 ±0.017 | 75.52 ±0.06 | 0.306 ±0.001 | 1.050 ±0.003 | 25.21 ±0.05 | 0.084 ±0.000 | — |
| B/16 | pBE | 2 | 2 | **0.510** ±0.001 | 14.13 ±0.02 | **0.016** ±0.000 | 1.328 ±0.006 | 30.81 ±0.11 | **0.020** ±0.000 | 3.743 ±0.011 | 71.12 ±0.16 | 0.211 ±0.002 | 0.945 ±0.002 | 23.75 ±0.03 | **0.044** ±0.001 | **10.11** |
| | down-DE | 1 | 4 | 0.511 ±0.002 | **13.95** ±0.01 | 0.019 ±0.001 | **1.293** ±0.008 | **29.67** ±0.18 | 0.026 ±0.000 | **3.544** ±0.015 | **66.49** ±0.27 | **0.190** ±0.002 | **0.930** ±0.003 | **23.17** ±0.06 | 0.050 ±0.001 | 180.69 |
| | V-MoE | 4 | 1 | 0.532 ±0.002 | 14.21 ±0.04 | 0.029 ±0.001 | 1.350 ±0.005 | 30.44 ±0.11 | 0.046 ±0.001 | 3.726 ±0.014 | 66.88 ±0.15 | 0.238 ±0.002 | 0.973 ±0.003 | 23.69 ±0.10 | 0.069 ±0.001 | — |
| B/32 | pBE | 2 | 2 | 0.612 ±0.001 | 16.49 ±0.02 | **0.013** ±0.000 | **1.491** ±0.003 | 33.85 ±0.05 | **0.019** ±0.000 | 4.872 ±0.007 | 82.80 ±0.08 | **0.275** ±0.001 | 1.099 ±0.003 | 27.36 ±0.10 | **0.045** ±0.001 | **9.14** |
| | down-DE | 1 | 4 | **0.607** ±0.001 | **16.17** ±0.02 | 0.021 ±0.001 | 1.483 ±0.003 | **33.36** ±0.13 | 0.027 ±0.000 | **4.787** ±0.011 | 82.11 ±0.07 | 0.276 ±0.001 | 1.099 ±0.003 | 26.98 ±0.10 | 0.055 ±0.002 | 174.91 |
| | V-MoE | 4 | 1 | 0.636 ±0.001 | 16.70 ±0.04 | 0.034 ±0.001 | 1.555 ±0.003 | 34.33 ±0.08 | 0.051 ±0.001 | 5.031 ±0.013 | **81.62** ±0.13 | 0.322 ±0.002 | 1.150 ±0.003 | 27.49 ±0.10 | 0.079 ±0.002 | — |
| S/32 | pBE | 2 | 2 | 0.805 ±0.002 | **20.97** ±0.05 | **0.013** ±0.000 | **2.112** ±0.007 | 45.05 ±0.09 | **0.028** ±0.000 | 6.283 ±0.013 | 93.68 ±0.09 | 0.342 ±0.001 | 1.408 ±0.003 | 33.71 ±0.10 | **0.051** ±0.001 | **11.73** |
| | down-DE | 1 | 4 | **0.795** ±0.003 | **20.66** ±0.13 | 0.015 ±0.001 | 2.076 ±0.012 | **44.16** ±0.21 | 0.031 ±0.000 | **5.990** ±0.011 | 92.25 ±0.03 | **0.324** ±0.001 | **1.372** ±0.002 | 32.83 ±0.13 | 0.054 ±0.002 | 143.10 |
| | V-MoE | 4 | 1 | 0.820 ±0.002 | **20.97** ±0.07 | 0.030 ±0.001 | 2.133 ±0.006 | 44.53 ±0.12 | 0.060 ±0.001 | 6.142 ±0.014 | **91.62** ±0.12 | 0.365 ±0.002 | 1.417 ±0.003 | 33.30 ±0.12 | 0.081 ±0.001 | — |

Table 15: ImageNet comparison of V-MoE and ViT.

| | | K | M | IMAGENET NLL↓ | IMAGENET ERROR↓ | ECE↓ | IMAGENET-C NLL↓ | IMAGENET-C ERROR↓ | ECE↓ | IMAGENET-A NLL↓ | IMAGENET-A ERROR↓ | ECE↓ | IMAGENET-V2 NLL↓ | IMAGENET-V2 ERROR↓ | ECE↓ |
|---|---|---|---|---|---|---|---|---|---|---|---|---|---|---|---|
| H/14 | V-MoE | 1 | 1 | **0.426** $_{\pm 0.002}$ | **11.81** $_{\pm 0.08}$ | 0.026 $_{\pm 0.000}$ | **0.932** $_{\pm 0.009}$ | **22.43** $_{\pm 0.16}$ | 0.033 $_{\pm 0.000}$ | 2.494 $_{\pm 0.057}$ | 50.22 $_{\pm 1.02}$ | **0.153** $_{\pm 0.004}$ | **0.806** $_{\pm 0.005}$ | **20.08** $_{\pm 0.13}$ | **0.059** $_{\pm 0.001}$ |
| | ViT | - | 1 | **0.426** $_{\pm 0.001}$ | 11.99 $_{\pm 0.05}$ | **0.023** $_{\pm 0.000}$ | **0.929** $_{\pm 0.004}$ | **22.45** $_{\pm 0.06}$ | **0.031** $_{\pm 0.000}$ | **2.332** $_{\pm 0.018}$ | **46.99** $_{\pm 0.21}$ | **0.149** $_{\pm 0.002}$ | **0.788** $_{\pm 0.004}$ | **20.08** $_{\pm 0.03}$ | **0.058** $_{\pm 0.001}$ |
| L/16 | V-MoE | 1 | 1 | **0.464** $_{\pm 0.001}$ | **12.91** $_{\pm 0.04}$ | 0.022 $_{\pm 0.000}$ | **1.050** $_{\pm 0.004}$ | **25.19** $_{\pm 0.08}$ | **0.030** $_{\pm 0.000}$ | **2.914** $_{\pm 0.016}$ | **58.02** $_{\pm 0.33}$ | **0.167** $_{\pm 0.001}$ | **0.844** $_{\pm 0.003}$ | **21.42** $_{\pm 0.06}$ | **0.053** $_{\pm 0.001}$ |
| | ViT | - | 1 | 0.473 $_{\pm 0.004}$ | 13.37 $_{\pm 0.08}$ | **0.020** $_{\pm 0.000}$ | 1.114 $_{\pm 0.009}$ | 26.68 $_{\pm 0.17}$ | 0.032 $_{\pm 0.001}$ | 3.094 $_{\pm 0.042}$ | 60.92 $_{\pm 0.56}$ | 0.203 $_{\pm 0.004}$ | 0.864 $_{\pm 0.007}$ | 22.04 $_{\pm 0.17}$ | 0.055 $_{\pm 0.001}$ |
| L/32 | V-MoE | 1 | 1 | **0.559** $_{\pm 0.001}$ | **15.10** $_{\pm 0.03}$ | 0.036 $_{\pm 0.001}$ | **1.246** $_{\pm 0.004}$ | **29.02** $_{\pm 0.07}$ | 0.046 $_{\pm 0.000}$ | 4.320 $_{\pm 0.013}$ | 75.25 $_{\pm 0.11}$ | **0.288** $_{\pm 0.002}$ | 1.032 $_{\pm 0.003}$ | 25.13 $_{\pm 0.09}$ | 0.077 $_{\pm 0.001}$ |
| | ViT | - | 1 | **0.556** $_{\pm 0.002}$ | 15.38 $_{\pm 0.05}$ | **0.025** $_{\pm 0.000}$ | 1.255 $_{\pm 0.009}$ | 29.57 $_{\pm 0.16}$ | **0.038** $_{\pm 0.000}$ | 4.286 $_{\pm 0.021}$ | 75.32 $_{\pm 0.30}$ | **0.287** $_{\pm 0.002}$ | **1.002** $_{\pm 0.004}$ | 25.32 $_{\pm 0.09}$ | **0.067** $_{\pm 0.001}$ |
| B/16 | V-MoE | 1 | 1 | **0.533** $_{\pm 0.001}$ | **14.33** $_{\pm 0.04}$ | 0.026 $_{\pm 0.000}$ | **1.355** $_{\pm 0.005}$ | **30.59** $_{\pm 0.10}$ | **0.040** $_{\pm 0.000}$ | **3.727** $_{\pm 0.014}$ | **67.72** $_{\pm 0.16}$ | **0.222** $_{\pm 0.001}$ | **0.965** $_{\pm 0.003}$ | **23.63** $_{\pm 0.11}$ | **0.060** $_{\pm 0.001}$ |
| | ViT | - | 1 | 0.565 $_{\pm 0.003}$ | 15.70 $_{\pm 0.06}$ | **0.021** $_{\pm 0.001}$ | 1.515 $_{\pm 0.006}$ | 34.52 $_{\pm 0.10}$ | 0.042 $_{\pm 0.001}$ | 4.219 $_{\pm 0.032}$ | 75.85 $_{\pm 0.29}$ | 0.280 $_{\pm 0.003}$ | 1.020 $_{\pm 0.006}$ | 25.77 $_{\pm 0.12}$ | **0.061** $_{\pm 0.001}$ |
| B/32 | V-MoE | 1 | 1 | **0.642** $_{\pm 0.002}$ | **16.90** $_{\pm 0.05}$ | 0.029 $_{\pm 0.001}$ | **1.568** $_{\pm 0.003}$ | **34.68** $_{\pm 0.05}$ | **0.045** $_{\pm 0.001}$ | **5.039** $_{\pm 0.009}$ | **82.49** $_{\pm 0.13}$ | **0.307** $_{\pm 0.002}$ | **1.151** $_{\pm 0.003}$ | **27.83** $_{\pm 0.10}$ | **0.071** $_{\pm 0.001}$ |
| | ViT | - | 1 | 0.688 $_{\pm 0.003}$ | 18.65 $_{\pm 0.08}$ | **0.022** $_{\pm 0.000}$ | 1.689 $_{\pm 0.005}$ | 38.02 $_{\pm 0.09}$ | **0.045** $_{\pm 0.000}$ | 5.358 $_{\pm 0.014}$ | 87.00 $_{\pm 0.12}$ | 0.342 $_{\pm 0.001}$ | 1.209 $_{\pm 0.005}$ | 29.89 $_{\pm 0.10}$ | **0.067** $_{\pm 0.001}$ |
| S/32 | V-MoE | 1 | 1 | **0.834** $_{\pm 0.002}$ | **21.43** $_{\pm 0.08}$ | 0.029 $_{\pm 0.001}$ | **2.171** $_{\pm 0.005}$ | **45.44** $_{\pm 0.09}$ | **0.056** $_{\pm 0.001}$ | **6.199** $_{\pm 0.012}$ | **92.47** $_{\pm 0.11}$ | **0.355** $_{\pm 0.001}$ | **1.433** $_{\pm 0.002}$ | **33.84** $_{\pm 0.02}$ | 0.078 $_{\pm 0.001}$ |
| | ViT | - | 1 | 0.907 $_{\pm 0.003}$ | 23.69 $_{\pm 0.05}$ | **0.016** $_{\pm 0.000}$ | 2.309 $_{\pm 0.010}$ | 48.75 $_{\pm 0.13}$ | **0.055** $_{\pm 0.001}$ | 6.639 $_{\pm 0.014}$ | 94.66 $_{\pm 0.07}$ | 0.375 $_{\pm 0.001}$ | 1.529 $_{\pm 0.003}$ | 36.42 $_{\pm 0.08}$ | **0.066** $_{\pm 0.001}$ |

Table 16: Cifar10 comparison of V-MoE, downstream ensembles there-of, and pBE with 2 experts per input in each case.

| | | K | M | NLL ↓ | Cifar10 Error ↓ | ECE ↓ | NLL ↓ | Cifar10-C Error ↓ | ECE ↓ | Downstream ΔFLOPs (%) ↓ |
|---|---|---|---|---|---|---|---|---|---|---|
| L/16 | pBE | 1 | 2 | **0.022** ± 0.000 | **0.55** ± 0.02 | **0.003** ± 0.000 | **0.198** ± 0.012 | **5.65** ± 0.20 | **0.027** ± 0.002 | **6.74** |
| | down-DE | 1 | 2 | 0.026 ± 0.001 | **0.59** ± 0.04 | 0.004 ± 0.000 | 0.227 ± 0.011 | 6.18 ± 0.21 | 0.032 ± 0.003 | 86.02 |
| | V-MoE | 2 | 1 | 0.028 ± 0.000 | 0.61 ± 0.01 | 0.004 ± 0.000 | 0.220 ± 0.012 | 6.17 ± 0.23 | 0.032 ± 0.003 | — |
| L/32 | pBE | 1 | 2 | **0.026** ± 0.001 | **0.69** ± 0.03 | **0.003** ± 0.000 | 0.246 ± 0.008 | 7.01 ± 0.15 | 0.033 ± 0.002 | **6.54** |
| | down-DE | 1 | 2 | 0.034 ± 0.001 | 0.81 ± 0.02 | 0.005 ± 0.000 | 0.252 ± 0.013 | 7.00 ± 0.26 | 0.032 ± 0.004 | 85.29 |
| | V-MoE | 2 | 1 | 0.042 ± 0.001 | 0.90 ± 0.03 | 0.006 ± 0.000 | 0.325 ± 0.012 | 8.05 ± 0.17 | 0.048 ± 0.002 | — |
| B/16 | pBE | 1 | 2 | 0.036 ± 0.001 | **0.84** ± 0.02 | 0.005 ± 0.000 | 0.295 ± 0.008 | 8.28 ± 0.15 | 0.038 ± 0.002 | **12.60** |
| | down-DE | 1 | 2 | **0.030** ± 0.001 | 0.81 ± 0.04 | **0.003** ± 0.000 | **0.247** ± 0.005 | **7.60** ± 0.16 | **0.028** ± 0.001 | 75.05 |
| | V-MoE | 2 | 1 | 0.040 ± 0.001 | 0.97 ± 0.02 | 0.006 ± 0.000 | 0.303 ± 0.008 | **8.08** ± 0.09 | 0.047 ± 0.002 | — |
| B/32 | pBE | 1 | 2 | 0.041 ± 0.001 | **1.11** ± 0.03 | 0.004 ± 0.000 | **0.340** ± 0.010 | **10.01** ± 0.18 | **0.040** ± 0.003 | **11.53** |
| | down-DE | 1 | 2 | **0.036** ± 0.000 | **1.07** ± 0.03 | **0.003** ± 0.000 | 0.345 ± 0.006 | 10.87 ± 0.13 | 0.042 ± 0.002 | 73.59 |
| | V-MoE | 2 | 1 | 0.042 ± 0.001 | **1.11** ± 0.03 | 0.006 ± 0.000 | 0.405 ± 0.015 | 11.30 ± 0.24 | 0.063 ± 0.004 | — |
| S/32 | pBE | 1 | 2 | 0.060 ± 0.001 | 1.92 ± 0.03 | **0.003** ± 0.000 | 0.476 ± 0.007 | **15.30** ± 0.17 | **0.049** ± 0.002 | **15.87** |
| | down-DE | 1 | 2 | **0.056** ± 0.001 | **1.75** ± 0.03 | 0.005 ± 0.000 | 0.491 ± 0.004 | 15.42 ± 0.15 | 0.064 ± 0.001 | 64.49 |
| | V-MoE | 2 | 1 | **0.058** ± 0.001 | **1.80** ± 0.03 | 0.007 ± 0.000 | 0.514 ± 0.006 | 15.48 ± 0.21 | 0.076 ± 0.002 | — |

Table 17: Cifar10 comparison of V-MoE, downstream ensembles there-of, and pBE with 4 experts per input in each case.

| | | K | M | NLL ↓ | Cifar10 Error ↓ | ECE ↓ | NLL ↓ | Cifar10-C Error ↓ | ECE ↓ | Downstream ΔFLOPs (%) ↓ |
|---|---|---|---|---|---|---|---|---|---|---|
| L/16 | pBE | 2 | 2 | **0.026** ± 0.001 | 0.58 ± 0.02 | **0.003** ± 0.000 | 0.247 ± 0.016 | 6.56 ± 0.26 | 0.036 ± 0.003 | **5.92** |
| | down-DE | 1 | 4 | **0.025** ± 0.001 | 0.57 ± 0.03 | 0.004 ± 0.000 | **0.219** ± 0.010 | **6.12** ± 0.19 | **0.030** ± 0.003 | 226.80 |
| | V-MoE | 4 | 1 | 0.034 ± 0.001 | 0.75 ± 0.02 | 0.005 ± 0.000 | 0.278 ± 0.010 | 7.71 ± 0.19 | 0.043 ± 0.002 | — |
| L/32 | pBE | 2 | 2 | **0.033** ± 0.001 | 0.80 ± 0.02 | 0.004 ± 0.000 | 0.242 ± 0.006 | 6.65 ± 0.12 | 0.031 ± 0.002 | **5.64** |
| | down-DE | 1 | 4 | **0.030** ± 0.001 | 0.78 ± 0.02 | 0.004 ± 0.000 | **0.232** ± 0.012 | 6.74 ± 0.27 | **0.028** ± 0.003 | 223.27 |
| | V-MoE | 4 | 1 | 0.035 ± 0.001 | **0.81** ± 0.02 | 0.005 ± 0.000 | 0.295 ± 0.009 | 7.63 ± 0.14 | 0.044 ± 0.002 | — |
| B/16 | pBE | 2 | 2 | 0.034 ± 0.000 | **0.81** ± 0.02 | 0.004 ± 0.000 | 0.283 ± 0.006 | 8.06 ± 0.12 | 0.037 ± 0.002 | **10.11** |
| | down-DE | 1 | 4 | **0.029** ± 0.000 | 0.80 ± 0.03 | **0.003** ± 0.000 | **0.242** ± 0.006 | 7.54 ± 0.16 | **0.026** ± 0.001 | 180.69 |
| | V-MoE | 4 | 1 | **0.032** ± 0.001 | **0.79** ± 0.02 | 0.005 ± 0.000 | 0.266 ± 0.004 | **7.55** ± 0.10 | 0.040 ± 0.001 | — |
| B/32 | pBE | 2 | 2 | **0.034** ± 0.001 | **1.00** ± 0.02 | **0.003** ± 0.000 | **0.306** ± 0.009 | **9.44** ± 0.23 | 0.037 ± 0.002 | **9.13** |
| | down-DE | 1 | 4 | 0.035 ± 0.001 | 1.06 ± 0.04 | **0.003** ± 0.000 | 0.338 ± 0.005 | 10.77 ± 0.12 | 0.040 ± 0.002 | 174.91 |
| | V-MoE | 4 | 1 | 0.042 ± 0.001 | 1.12 ± 0.02 | 0.006 ± 0.000 | 0.410 ± 0.017 | 11.29 ± 0.26 | 0.066 ± 0.004 | — |
| S/32 | pBE | 2 | 2 | 0.058 ± 0.001 | 1.82 ± 0.01 | 0.004 ± 0.000 | 0.472 ± 0.007 | **15.01** ± 0.17 | 0.051 ± 0.002 | **11.73** |
| | down-DE | 1 | 4 | **0.055** ± 0.001 | **1.76** ± 0.04 | 0.004 ± 0.000 | 0.486 ± 0.003 | 15.36 ± 0.13 | 0.061 ± 0.001 | 143.08 |
| | V-MoE | 4 | 1 | 0.059 ± 0.001 | **1.80** ± 0.05 | 0.007 ± 0.000 | 0.536 ± 0.012 | 15.64 ± 0.27 | 0.084 ± 0.004 | — |

Table 18: Cifar10 comparison of V-MoE and ViT.

| | | K | M | NLL ↓ | Cifar10 Error ↓ | ECE ↓ | NLL ↓ | Cifar10-C Error ↓ | ECE ↓ |
|---|---|---|---|---|---|---|---|---|---|
| L/16 | V-MoE | 1 | 1 | **0.029** ± 0.001 | **0.59** ± 0.02 | **0.004** ± 0.000 | **0.236** ± 0.011 | **6.25** ± 0.20 | **0.034** ± 0.002 |
| | ViT | - | 1 | 0.034 ± 0.001 | **0.69** ± 0.03 | **0.005** ± 0.000 | 0.325 ± 0.020 | 7.53 ± 0.40 | 0.050 ± 0.004 |
| L/32 | V-MoE | 1 | 1 | 0.040 ± 0.001 | 0.86 ± 0.02 | 0.006 ± 0.000 | 0.290 ± 0.013 | 7.35 ± 0.22 | **0.043** ± 0.003 |
| | ViT | - | 1 | **0.030** ± 0.001 | **0.78** ± 0.02 | **0.005** ± 0.000 | **0.281** ± 0.010 | **7.28** ± 0.13 | **0.043** ± 0.002 |
| B/16 | V-MoE | 1 | 1 | **0.030** ± 0.001 | **0.85** ± 0.02 | **0.003** ± 0.000 | **0.249** ± 0.004 | **7.53** ± 0.12 | **0.030** ± 0.001 |
| | ViT | - | 1 | **0.031** ± 0.001 | **0.87** ± 0.03 | 0.004 ± 0.000 | 0.293 ± 0.009 | 7.99 ± 0.15 | 0.043 ± 0.002 |
| B/32 | V-MoE | 1 | 1 | **0.038** ± 0.000 | 1.14 ± 0.03 | **0.004** ± 0.000 | 0.359 ± 0.008 | 11.09 ± 0.17 | **0.046** ± 0.002 |
| | ViT | - | 1 | 0.050 ± 0.002 | 1.38 ± 0.04 | 0.007 ± 0.000 | **0.330** ± 0.010 | **8.97** ± 0.19 | 0.051 ± 0.002 |
| S/32 | V-MoE | 1 | 1 | **0.059** ± 0.001 | **1.84** ± 0.02 | **0.006** ± 0.000 | **0.497** ± 0.007 | 15.53 ± 0.20 | **0.066** ± 0.002 |
| | ViT | - | 1 | 0.065 ± 0.001 | 2.08 ± 0.02 | **0.006** ± 0.000 | 0.518 ± 0.008 | **15.04** ± 0.24 | **0.072** ± 0.002 |

Table 19: Cifar10 OOD comparison of V-MoE, downstream ensembles there-of, and pBE with 2 experts per input in each case.

| | | K | M | CIFAR10 vs. CIFAR100 AUC (PR) ↑ | AUC (ROC) ↑ | FPR@95 ↓ | CIFAR10 vs. DTD AUC (PR) ↑ | AUC (ROC) ↑ | FPR@95 ↓ | CIFAR10 vs. PLACES365 AUC (PR) ↑ | AUC (ROC) ↑ | FPR@95 ↓ | CIFAR10 vs. SVHN AUC (PR) ↑ | AUC (ROC) ↑ | FPR@95 ↓ |
|---|---|---|---|---|---|---|---|---|---|---|---|---|---|---|---|
| L/16 | pBE | 1 | 2 | **0.9905** ±0.0003 | **0.9902** ±0.0003 | **0.0313** ±0.0010 | 0.9999 ±0.0000 | 0.9993 ±0.0000 | 0.0005 ±0.0001 | 0.9103 ±0.0068 | 0.9936 ±0.0003 | 0.0294 ±0.0013 | **0.9963** ±0.0002 | **0.9978** ±0.0001 | **0.0007** ±0.0002 |
| | down-DE | 1 | 2 | **0.9896** ±0.0005 | **0.9896** ±0.0005 | **0.0357** ±0.0029 | **0.9999** ±0.0000 | **0.9996** ±0.0000 | **0.0001** ±0.0001 | **0.9421** ±0.0063 | **0.9959** ±0.0003 | **0.0162** ±0.0010 | **0.9968** ±0.0002 | **0.9982** ±0.0002 | **0.0007** ±0.0002 |
| | V-MoE | 2 | 1 | **0.9895** ±0.0004 | **0.9890** ±0.0003 | 0.0389 ±0.0018 | 0.9998 ±0.0000 | 0.9988 ±0.0001 | 0.0011 ±0.0001 | 0.7891 ±0.0120 | 0.9887 ±0.0004 | 0.0435 ±0.0005 | **0.9966** ±0.0001 | **0.9980** ±0.0001 | **0.0005** ±0.0001 |
| L/32 | pBE | 1 | 2 | **0.9875** ±0.0003 | **0.9873** ±0.0002 | **0.0427** ±0.0011 | **0.9998** ±0.0000 | **0.9990** ±0.0000 | **0.0011** ±0.0002 | **0.9196** ±0.0028 | **0.9934** ±0.0002 | **0.0321** ±0.0013 | **0.9950** ±0.0002 | **0.9970** ±0.0002 | **0.0012** ±0.0001 |
| | down-DE | 1 | 2 | 0.9852 ±0.0005 | 0.9842 ±0.0005 | 0.0571 ±0.0016 | 0.9994 ±0.0001 | 0.9966 ±0.0004 | 0.0032 ±0.0002 | 0.7388 ±0.0165 | 0.9846 ±0.0007 | 0.0561 ±0.0019 | 0.9939 ±0.0004 | 0.9960 ±0.0003 | 0.0012 ±0.0002 |
| | V-MoE | 2 | 1 | 0.9839 ±0.0007 | 0.9839 ±0.0005 | 0.0589 ±0.0020 | 0.9998 ±0.0000 | 0.9987 ±0.0001 | **0.0013** ±0.0003 | 0.8967 ±0.0056 | **0.9927** ±0.0003 | **0.0346** ±0.0012 | **0.9944** ±0.0003 | 0.9965 ±0.0002 | 0.0017 ±0.0003 |
| B/16 | pBE | 1 | 2 | 0.9823 ±0.0003 | 0.9800 ±0.0003 | 0.0846 ±0.0026 | 0.9994 ±0.0000 | 0.9967 ±0.0002 | **0.0015** ±0.0002 | 0.7661 ±0.0121 | 0.9834 ±0.0004 | 0.0602 ±0.0010 | 0.9914 ±0.0002 | 0.9942 ±0.0002 | **0.0029** ±0.0005 |
| | down-DE | 1 | 2 | **0.9860** ±0.0005 | **0.9859** ±0.0004 | **0.0540** ±0.0026 | **0.9999** ±0.0000 | **0.9993** ±0.0001 | **0.0009** ±0.0002 | **0.8807** ±0.0148 | **0.9922** ±0.0005 | **0.0388** ±0.0013 | **0.9951** ±0.0003 | **0.9974** ±0.0002 | **0.0027** ±0.0005 |
| | V-MoE | 2 | 1 | 0.9822 ±0.0005 | 0.9809 ±0.0005 | 0.0800 ±0.0030 | 0.9994 ±0.0000 | 0.9964 ±0.0002 | 0.0023 ±0.0003 | 0.7399 ±0.0141 | 0.9841 ±0.0006 | 0.0568 ±0.0014 | 0.9925 ±0.0004 | 0.9953 ±0.0002 | 0.0037 ±0.0009 |
| B/32 | pBE | 1 | 2 | 0.9773 ±0.0003 | 0.9738 ±0.0004 | 0.1254 ±0.0020 | 0.9992 ±0.0000 | 0.9957 ±0.0003 | 0.0038 ±0.0006 | 0.7438 ±0.0085 | 0.9794 ±0.0006 | 0.0802 ±0.0023 | 0.9881 ±0.0003 | 0.9918 ±0.0002 | 0.0071 ±0.0004 |
| | down-DE | 1 | 2 | **0.9820** ±0.0003 | **0.9813** ±0.0002 | **0.0810** ±0.0033 | **0.9998** ±0.0000 | **0.9989** ±0.0001 | **0.0014** ±0.0004 | 0.8567 ±0.0099 | 0.9900 ±0.0005 | 0.0514 ±0.0017 | **0.9924** ±0.0004 | **0.9957** ±0.0003 | **0.0054** ±0.0004 |
| | V-MoE | 2 | 1 | 0.9798 ±0.0004 | 0.9790 ±0.0004 | 0.0875 ±0.0020 | **0.9997** ±0.0000 | 0.9986 ±0.0001 | 0.0017 ±0.0003 | **0.8725** ±0.0074 | **0.9904** ±0.0006 | **0.0482** ±0.0019 | 0.9919 ±0.0002 | 0.9952 ±0.0002 | 0.0047 ±0.0004 |
| S/32 | pBE | 1 | 2 | 0.9617 ±0.0003 | 0.9567 ±0.0003 | 0.2629 ±0.0040 | 0.9992 ±0.0000 | 0.9956 ±0.0002 | 0.0118 ±0.0011 | 0.7431 ±0.0070 | 0.9750 ±0.0006 | 0.1270 ±0.0036 | 0.9824 ±0.0004 | 0.9899 ±0.0003 | 0.0388 ±0.0028 |
| | down-DE | 1 | 2 | **0.9696** ±0.0003 | **0.9669** ±0.0005 | **0.1856** ±0.0046 | **0.9996** ±0.0000 | **0.9981** ±0.0001 | **0.0037** ±0.0002 | **0.8135** ±0.0146 | **0.9845** ±0.0007 | **0.0815** ±0.0036 | **0.9879** ±0.0004 | **0.9933** ±0.0002 | **0.0164** ±0.0019 |
| | V-MoE | 2 | 1 | 0.9691 ±0.0008 | 0.9667 ±0.0009 | 0.1870 ±0.0065 | 0.9996 ±0.0000 | 0.9979 ±0.0001 | 0.0043 ±0.0006 | 0.8010 ±0.0087 | 0.9839 ±0.0007 | 0.0827 ±0.0032 | 0.9873 ±0.0005 | 0.9927 ±0.0004 | 0.0192 ±0.0016 |

Table 20: Cifar10 OOD comparison of ViT and V-MoE

| | | K | M | CIFAR10 vs. CIFAR100 AUC (PR) ↑ | AUC (ROC) ↑ | FPR@95 ↓ | CIFAR10 vs. DTD AUC (PR) ↑ | AUC (ROC) ↑ | FPR@95 ↓ | CIFAR10 vs. PLACES365 AUC (PR) ↑ | AUC (ROC) ↑ | FPR@95 ↓ | CIFAR10 vs. SVHN AUC (PR) ↑ | AUC (ROC) ↑ | FPR@95 ↓ |
|---|---|---|---|---|---|---|---|---|---|---|---|---|---|---|---|
| L/16 | V-MoE | 1 | 1 | **0.9891** ±0.0004 | **0.9895** ±0.0004 | **0.0379** ±0.0022 | **0.9999** ±0.0000 | **0.9997** ±0.0000 | **0.0001** ±0.0001 | **0.9400** ±0.0057 | **0.9960** ±0.0002 | **0.0162** ±0.0013 | **0.9972** ±0.0001 | **0.9984** ±0.0001 | **0.0007** ±0.0001 |
| | ViT | - | 1 | 0.9839 ±0.0008 | 0.9845 ±0.0007 | 0.0541 ±0.0026 | 0.9996 ±0.0000 | 0.9978 ±0.0002 | 0.0018 ±0.0004 | 0.7334 ±0.0227 | 0.9857 ±0.0010 | 0.0492 ±0.0024 | 0.9947 ±0.0003 | 0.9967 ±0.0002 | 0.0022 ±0.0003 |
| L/32 | V-MoE | 1 | 1 | **0.9854** ±0.0003 | **0.9850** ±0.0003 | **0.0573** ±0.0018 | 0.9996 ±0.0000 | 0.9976 ±0.0002 | 0.0030 ±0.0003 | 0.7489 ±0.0049 | 0.9862 ±0.0003 | 0.0520 ±0.0011 | **0.9946** ±0.0003 | **0.9967** ±0.0002 | **0.0015** ±0.0003 |
| | ViT | - | 1 | **0.9842** ±0.0005 | **0.9847** ±0.0004 | **0.0551** ±0.0018 | **0.9998** ±0.0000 | **0.9987** ±0.0001 | **0.0016** ±0.0003 | **0.9164** ±0.0046 | **0.9938** ±0.0003 | **0.0279** ±0.0015 | 0.9942 ±0.0002 | 0.9966 ±0.0001 | 0.0028 ±0.0003 |
| B/16 | V-MoE | 1 | 1 | **0.9856** ±0.0004 | **0.9855** ±0.0004 | **0.0554** ±0.0025 | **0.9998** ±0.0000 | **0.9992** ±0.0001 | **0.0015** ±0.0005 | **0.8598** ±0.0191 | **0.9912** ±0.0008 | **0.0413** ±0.0028 | **0.9949** ±0.0003 | **0.9972** ±0.0002 | **0.0027** ±0.0003 |
| | ViT | - | 1 | 0.9801 ±0.0004 | 0.9798 ±0.0004 | 0.0857 ±0.0027 | 0.9996 ±0.0000 | 0.9979 ±0.0001 | 0.0046 ±0.0004 | 0.8536 ±0.0057 | 0.9895 ±0.0004 | 0.0511 ±0.0012 | 0.9926 ±0.0002 | 0.9961 ±0.0002 | 0.0048 ±0.0004 |
| B/32 | V-MoE | 1 | 1 | **0.9814** ±0.0003 | **0.9806** ±0.0003 | **0.0853** ±0.0027 | **0.9998** ±0.0000 | **0.9989** ±0.0001 | **0.0017** ±0.0005 | **0.8590** ±0.0097 | **0.9902** ±0.0005 | **0.0506** ±0.0021 | **0.9923** ±0.0003 | **0.9958** ±0.0003 | **0.0061** ±0.0005 |
| | ViT | - | 1 | 0.9752 ±0.0005 | 0.9716 ±0.0006 | 0.1485 ±0.0040 | 0.9985 ±0.0001 | 0.9915 ±0.0004 | 0.0186 ±0.0010 | 0.6507 ±0.0056 | 0.9734 ±0.0006 | 0.1021 ±0.0037 | 0.9872 ±0.0004 | 0.9920 ±0.0003 | 0.0148 ±0.0012 |
| S/32 | V-MoE | 1 | 1 | **0.9685** ±0.0008 | **0.9658** ±0.0010 | **0.1922** ±0.0056 | **0.9996** ±0.0000 | **0.9977** ±0.0002 | **0.0045** ±0.0007 | **0.8092** ±0.0105 | **0.9841** ±0.0008 | **0.0821** ±0.0032 | **0.9874** ±0.0006 | **0.9929** ±0.0004 | **0.0185** ±0.0019 |
| | ViT | - | 1 | 0.9629 ±0.0004 | 0.9588 ±0.0004 | 0.2422 ±0.0027 | 0.9993 ±0.0000 | 0.9963 ±0.0002 | 0.0134 ±0.0013 | 0.7177 ±0.0048 | 0.9775 ±0.0003 | 0.1110 ±0.0015 | 0.9807 ±0.0003 | 0.9889 ±0.0002 | 0.0426 ±0.0013 |

Table 21: Cifar100 OOD comparison of V-MoE, downstream ensembles there-of, and pBE with 2 experts per input in each case.

| | | K | M | CIFAR100 vs. CIFAR10 AUC (PR)↑ | AUC (ROC)↑ | FPR@95↓ | CIFAR100 vs. DTD AUC (PR)↑ | AUC (ROC)↑ | FPR@95↓ | CIFAR100 vs. PLACES365 AUC (PR)↑ | AUC (ROC)↑ | FPR@95↓ | CIFAR100 vs. SVHN AUC (PR)↑ | AUC (ROC)↑ | FPR@95↓ |
|---|---|---|---|---|---|---|---|---|---|---|---|---|---|---|---|
| L/16 | pBE | 1 | 2 | **0.9507** $_{\pm 0.0012}$ | **0.9490** $_{\pm 0.0011}$ | **0.2889** $_{\pm 0.0058}$ | 0.9969 $_{\pm 0.0001}$ | 0.9847 $_{\pm 0.0002}$ | 0.0862 $_{\pm 0.0010}$ | 0.7281 $_{\pm 0.0023}$ | 0.9542 $_{\pm 0.0007}$ | 0.2969 $_{\pm 0.0036}$ | **0.9011** $_{\pm 0.0057}$ | **0.9482** $_{\pm 0.0024}$ | **0.3071** $_{\pm 0.0093}$ |
| | down-DE | 1 | 2 | 0.9455 $_{\pm 0.0013}$ | **0.9481** $_{\pm 0.0019}$ | **0.2692** $_{\pm 0.0093}$ | **0.9975** $_{\pm 0.0000}$ | **0.9881** $_{\pm 0.0002}$ | **0.0665** $_{\pm 0.0035}$ | **0.7619** $_{\pm 0.0065}$ | **0.9664** $_{\pm 0.0012}$ | **0.2158** $_{\pm 0.0071}$ | **0.9071** $_{\pm 0.0027}$ | **0.9507** $_{\pm 0.0016}$ | **0.2932** $_{\pm 0.0144}$ |
| | V-MoE | 2 | 1 | 0.9391 $_{\pm 0.0014}$ | **0.9443** $_{\pm 0.0015}$ | 0.2901 $_{\pm 0.0076}$ | **0.9977** $_{\pm 0.0000}$ | **0.9887** $_{\pm 0.0002}$ | 0.0674 $_{\pm 0.0017}$ | **0.7773** $_{\pm 0.0054}$ | **0.9680** $_{\pm 0.0009}$ | **0.2150** $_{\pm 0.0041}$ | 0.9002 $_{\pm 0.0045}$ | 0.9461 $_{\pm 0.0024}$ | 0.3223 $_{\pm 0.0124}$ |
| L/32 | pBE | 1 | 2 | **0.9423** $_{\pm 0.0003}$ | **0.9379** $_{\pm 0.0006}$ | 0.3591 $_{\pm 0.0058}$ | **0.9958** $_{\pm 0.0001}$ | **0.9792** $_{\pm 0.0004}$ | **0.1165** $_{\pm 0.0028}$ | **0.6653** $_{\pm 0.0037}$ | **0.9394** $_{\pm 0.0011}$ | **0.3670** $_{\pm 0.0051}$ | **0.8848** $_{\pm 0.0061}$ | **0.9398** $_{\pm 0.0026}$ | 0.3428 $_{\pm 0.0080}$ |
| | down-DE | 1 | 2 | **0.9380** $_{\pm 0.0029}$ | **0.9387** $_{\pm 0.0015}$ | 0.3362 $_{\pm 0.0042}$ | **0.9958** $_{\pm 0.0001}$ | 0.9796 $_{\pm 0.0004}$ | 0.1176 $_{\pm 0.0015}$ | 0.6710 $_{\pm 0.0088}$ | 0.9436 $_{\pm 0.0019}$ | 0.3465 $_{\pm 0.0093}$ | **0.8877** $_{\pm 0.0081}$ | 0.9460 $_{\pm 0.0027}$ | **0.3033** $_{\pm 0.0077}$ |
| | V-MoE | 2 | 1 | 0.9275 $_{\pm 0.0041}$ | **0.9330** $_{\pm 0.0020}$ | 0.3604 $_{\pm 0.0052}$ | **0.9958** $_{\pm 0.0001}$ | 0.9790 $_{\pm 0.0005}$ | 0.1306 $_{\pm 0.0029}$ | 0.6773 $_{\pm 0.0075}$ | 0.9445 $_{\pm 0.0017}$ | 0.3489 $_{\pm 0.0089}$ | 0.8767 $_{\pm 0.0075}$ | 0.9393 $_{\pm 0.0020}$ | 0.3404 $_{\pm 0.0069}$ |
| B/16 | pBE | 1 | 2 | **0.9200** $_{\pm 0.0008}$ | 0.9121 $_{\pm 0.0010}$ | 0.4656 $_{\pm 0.0071}$ | 0.9938 $_{\pm 0.0002}$ | 0.9710 $_{\pm 0.0011}$ | 0.1542 $_{\pm 0.0063}$ | 0.5475 $_{\pm 0.0076}$ | 0.9102 $_{\pm 0.0014}$ | 0.4653 $_{\pm 0.0052}$ | **0.8730** $_{\pm 0.0056}$ | 0.9319 $_{\pm 0.0025}$ | **0.3699** $_{\pm 0.0090}$ |
| | down-DE | 1 | 2 | **0.9242** $_{\pm 0.0016}$ | **0.9275** $_{\pm 0.0013}$ | **0.3506** $_{\pm 0.0070}$ | **0.9952** $_{\pm 0.0003}$ | **0.9777** $_{\pm 0.0011}$ | **0.1144** $_{\pm 0.0052}$ | **0.6023** $_{\pm 0.0101}$ | **0.9343** $_{\pm 0.0015}$ | **0.3577** $_{\pm 0.0049}$ | 0.8726 $_{\pm 0.0049}$ | **0.9340** $_{\pm 0.0027}$ | 0.3526 $_{\pm 0.0124}$ |
| | V-MoE | 2 | 1 | 0.9159 $_{\pm 0.0019}$ | 0.9211 $_{\pm 0.0015}$ | 0.3729 $_{\pm 0.0053}$ | 0.9950 $_{\pm 0.0002}$ | 0.9768 $_{\pm 0.0008}$ | 0.1193 $_{\pm 0.0038}$ | 0.6029 $_{\pm 0.0066}$ | 0.9331 $_{\pm 0.0008}$ | 0.3682 $_{\pm 0.0029}$ | 0.8704 $_{\pm 0.0038}$ | 0.9283 $_{\pm 0.0020}$ | 0.3915 $_{\pm 0.0076}$ |
| B/32 | pBE | 1 | 2 | 0.9075 $_{\pm 0.0012}$ | 0.8945 $_{\pm 0.0015}$ | 0.5358 $_{\pm 0.0063}$ | 0.9906 $_{\pm 0.0003}$ | 0.9554 $_{\pm 0.0019}$ | 0.2481 $_{\pm 0.0101}$ | 0.4355 $_{\pm 0.0071}$ | 0.8811 $_{\pm 0.0014}$ | 0.5465 $_{\pm 0.0043}$ | **0.8583** $_{\pm 0.0040}$ | 0.9221 $_{\pm 0.0024}$ | **0.4051** $_{\pm 0.0100}$ |
| | down-DE | 1 | 2 | **0.9188** $_{\pm 0.0021}$ | **0.9158** $_{\pm 0.0014}$ | **0.4248** $_{\pm 0.0077}$ | **0.9942** $_{\pm 0.0003}$ | **0.9719** $_{\pm 0.0014}$ | **0.1517** $_{\pm 0.0090}$ | **0.5525** $_{\pm 0.0096}$ | **0.9176** $_{\pm 0.0022}$ | **0.4276** $_{\pm 0.0085}$ | 0.8756 $_{\pm 0.0083}$ | **0.9292** $_{\pm 0.0033}$ | **0.3959** $_{\pm 0.0083}$ |
| | V-MoE | 2 | 1 | **0.9192** $_{\pm 0.0014}$ | **0.9151** $_{\pm 0.0012}$ | 0.4444 $_{\pm 0.0060}$ | 0.9913 $_{\pm 0.0002}$ | 0.9580 $_{\pm 0.0008}$ | 0.2481 $_{\pm 0.0064}$ | 0.4449 $_{\pm 0.0100}$ | 0.8969 $_{\pm 0.0009}$ | 0.5230 $_{\pm 0.0027}$ | 0.8641 $_{\pm 0.0073}$ | 0.9252 $_{\pm 0.0028}$ | 0.4249 $_{\pm 0.0080}$ |
| S/32 | pBE | 1 | 2 | **0.8677** $_{\pm 0.0016}$ | 0.8528 $_{\pm 0.0017}$ | 0.6086 $_{\pm 0.0049}$ | **0.9878** $_{\pm 0.0007}$ | 0.9430 $_{\pm 0.0026}$ | 0.2677 $_{\pm 0.0084}$ | 0.3628 $_{\pm 0.0072}$ | 0.8519 $_{\pm 0.0026}$ | 0.5716 $_{\pm 0.0074}$ | **0.8279** $_{\pm 0.0037}$ | **0.9004** $_{\pm 0.0024}$ | **0.4543** $_{\pm 0.0098}$ |
| | down-DE | 1 | 2 | **0.8697** $_{\pm 0.0019}$ | **0.8651** $_{\pm 0.0024}$ | **0.5203** $_{\pm 0.0095}$ | **0.9892** $_{\pm 0.0005}$ | **0.9532** $_{\pm 0.0023}$ | **0.1909** $_{\pm 0.0086}$ | **0.4185** $_{\pm 0.0063}$ | **0.8814** $_{\pm 0.0020}$ | **0.4657** $_{\pm 0.0078}$ | 0.8162 $_{\pm 0.0045}$ | 0.8912 $_{\pm 0.0026}$ | 0.4835 $_{\pm 0.0066}$ |
| | V-MoE | 2 | 1 | 0.8687 $_{\pm 0.0011}$ | **0.8663** $_{\pm 0.0012}$ | 0.5206 $_{\pm 0.0061}$ | **0.9894** $_{\pm 0.0004}$ | 0.9523 $_{\pm 0.0018}$ | 0.2076 $_{\pm 0.0075}$ | 0.4129 $_{\pm 0.0082}$ | 0.8752 $_{\pm 0.0026}$ | 0.5039 $_{\pm 0.0072}$ | 0.8133 $_{\pm 0.0029}$ | 0.8882 $_{\pm 0.0020}$ | 0.5031 $_{\pm 0.0058}$ |

Table 22: Cifar100 OOD comparison of V-MoE and ViT

| | | K | M | CIFAR100 vs. CIFAR10 AUC (PR)↑ | AUC (ROC)↑ | FPR@95↓ | CIFAR100 vs. DTD AUC (PR)↑ | AUC (ROC)↑ | FPR@95↓ | CIFAR100 vs. PLACES365 AUC (PR)↑ | AUC (ROC)↑ | FPR@95↓ | CIFAR100 vs. SVHN AUC (PR)↑ | AUC (ROC)↑ | FPR@95↓ |
|---|---|---|---|---|---|---|---|---|---|---|---|---|---|---|---|
| L/16 | V-MoE | 1 | 1 | **0.9454** $_{\pm 0.0013}$ | **0.9481** $_{\pm 0.0015}$ | **0.2682** $_{\pm 0.0073}$ | **0.9976** $_{\pm 0.0000}$ | **0.9882** $_{\pm 0.0002}$ | **0.0658** $_{\pm 0.0024}$ | **0.7631** $_{\pm 0.0043}$ | **0.9667** $_{\pm 0.0007}$ | **0.2141** $_{\pm 0.0040}$ | **0.9115** $_{\pm 0.0035}$ | **0.9533** $_{\pm 0.0020}$ | **0.2794** $_{\pm 0.0123}$ |
| | ViT | - | 1 | **0.9411** $_{\pm 0.0019}$ | **0.9449** $_{\pm 0.0012}$ | **0.2734** $_{\pm 0.0062}$ | 0.9970 $_{\pm 0.0001}$ | 0.9854 $_{\pm 0.0006}$ | 0.0853 $_{\pm 0.0039}$ | 0.7552 $_{\pm 0.0118}$ | 0.9608 $_{\pm 0.0017}$ | 0.2566 $_{\pm 0.0066}$ | 0.8697 $_{\pm 0.0033}$ | 0.9326 $_{\pm 0.0016}$ | 0.3690 $_{\pm 0.0058}$ |
| L/32 | V-MoE | 1 | 1 | **0.9358** $_{\pm 0.0028}$ | **0.9368** $_{\pm 0.0018}$ | 0.3431 $_{\pm 0.0052}$ | 0.9961 $_{\pm 0.0001}$ | 0.9806 $_{\pm 0.0004}$ | 0.1148 $_{\pm 0.0022}$ | 0.6757 $_{\pm 0.0080}$ | 0.9461 $_{\pm 0.0016}$ | 0.3308 $_{\pm 0.0081}$ | **0.8863** $_{\pm 0.0093}$ | **0.9459** $_{\pm 0.0025}$ | **0.3063** $_{\pm 0.0123}$ |
| | ViT | - | 1 | **0.9285** $_{\pm 0.0020}$ | **0.9323** $_{\pm 0.0016}$ | **0.3201** $_{\pm 0.0068}$ | **0.9967** $_{\pm 0.0001}$ | **0.9838** $_{\pm 0.0006}$ | **0.0911** $_{\pm 0.0036}$ | **0.7541** $_{\pm 0.0066}$ | **0.9634** $_{\pm 0.0014}$ | **0.2292** $_{\pm 0.0068}$ | 0.8495 $_{\pm 0.0068}$ | 0.9234 $_{\pm 0.0035}$ | 0.3949 $_{\pm 0.0137}$ |
| B/16 | V-MoE | 1 | 1 | **0.9206** $_{\pm 0.0015}$ | **0.9241** $_{\pm 0.0014}$ | **0.3572** $_{\pm 0.0061}$ | **0.9951** $_{\pm 0.0002}$ | **0.9776** $_{\pm 0.0007}$ | **0.1094** $_{\pm 0.0034}$ | **0.5924** $_{\pm 0.0076}$ | **0.9334** $_{\pm 0.0012}$ | **0.3532** $_{\pm 0.0035}$ | **0.8720** $_{\pm 0.0048}$ | **0.9309** $_{\pm 0.0023}$ | **0.3705** $_{\pm 0.0122}$ |
| | ViT | - | 1 | **0.9177** $_{\pm 0.0013}$ | 0.9171 $_{\pm 0.0014}$ | 0.3925 $_{\pm 0.0069}$ | 0.9906 $_{\pm 0.0003}$ | 0.9571 $_{\pm 0.0012}$ | 0.2199 $_{\pm 0.0059}$ | 0.4913 $_{\pm 0.0088}$ | 0.8980 $_{\pm 0.0024}$ | 0.4927 $_{\pm 0.0079}$ | 0.8525 $_{\pm 0.0044}$ | 0.9204 $_{\pm 0.0022}$ | 0.4226 $_{\pm 0.0065}$ |
| B/32 | V-MoE | 1 | 1 | **0.9166** $_{\pm 0.0017}$ | **0.9145** $_{\pm 0.0012}$ | 0.4211 $_{\pm 0.0048}$ | **0.9940** $_{\pm 0.0002}$ | **0.9714** $_{\pm 0.0009}$ | **0.1562** $_{\pm 0.0068}$ | **0.5433** $_{\pm 0.0089}$ | **0.9178** $_{\pm 0.0016}$ | **0.4258** $_{\pm 0.0049}$ | **0.8730** $_{\pm 0.0071}$ | **0.9276** $_{\pm 0.0033}$ | **0.4026** $_{\pm 0.0100}$ |
| | ViT | - | 1 | 0.9038 $_{\pm 0.0022}$ | 0.9044 $_{\pm 0.0018}$ | **0.4180** $_{\pm 0.0061}$ | 0.9927 $_{\pm 0.0002}$ | 0.9663 $_{\pm 0.0008}$ | **0.1631** $_{\pm 0.0055}$ | 0.5306 $_{\pm 0.0046}$ | 0.9112 $_{\pm 0.0015}$ | **0.4176** $_{\pm 0.0062}$ | 0.8379 $_{\pm 0.0029}$ | 0.9116 $_{\pm 0.0014}$ | 0.4328 $_{\pm 0.0062}$ |
| S/32 | V-MoE | 1 | 1 | **0.8678** $_{\pm 0.0012}$ | **0.8631** $_{\pm 0.0015}$ | **0.5281** $_{\pm 0.0050}$ | **0.9893** $_{\pm 0.0007}$ | **0.9524** $_{\pm 0.0031}$ | **0.1978** $_{\pm 0.0129}$ | **0.4024** $_{\pm 0.0091}$ | **0.8778** $_{\pm 0.0029}$ | **0.4763** $_{\pm 0.0085}$ | **0.8224** $_{\pm 0.0058}$ | **0.8945** $_{\pm 0.0034}$ | **0.4729** $_{\pm 0.0098}$ |
| | ViT | - | 1 | **0.8644** $_{\pm 0.0018}$ | 0.8541 $_{\pm 0.0020}$ | 0.5716 $_{\pm 0.0061}$ | 0.9836 $_{\pm 0.0007}$ | 0.9287 $_{\pm 0.0026}$ | 0.2976 $_{\pm 0.0082}$ | 0.3149 $_{\pm 0.0031}$ | 0.8349 $_{\pm 0.0024}$ | 0.5982 $_{\pm 0.0061}$ | 0.8088 $_{\pm 0.0051}$ | **0.8894** $_{\pm 0.0029}$ | **0.4888** $_{\pm 0.0084}$ |

Table 23: Few-shot comparison of pBE, V-MoE and ensembles thereof.

| | | K | M | Mean Across Datasets 1-Shot Error ↓ | 5-Shot Error ↓ | 10-Shot Error ↓ | 25-Shot Error ↓ | Weighted Mean Across Datasets 1-Shot Error ↓ | 5-Shot Error ↓ | 10-Shot Error ↓ | 25-Shot Error ↓ | Downstream ΔFLOPs (%) ↓ |
|---|---|---|---|---|---|---|---|---|---|---|---|---|
| H/14 | pBE | 1 | 2 | **31.47** ±0.72 | **17.87** ±0.32 | **14.29** ±0.22 | **10.64** ±0.25 | **78.51** ±0.24 | **80.65** ±0.08 | **81.28** ±0.05 | **81.51** ±0.06 | **15.45** |
| | down-DE | 1 | 2 | **30.84** ±0.42 | **17.77** ±0.51 | **14.43** ±0.03 | **11.06** ±0.40 | **78.36** ±0.20 | **80.63** ±0.14 | **81.31** ±0.02 | **81.60** ±0.09 | 75.05 |
| | V-MoE | 2 | 1 | 32.47 ±0.55 | 18.77 ±0.41 | 15.19 ±0.27 | 12.09 ±0.38 | 78.93 ±0.20 | 80.89 ±0.11 | 81.48 ±0.07 | 81.83 ±0.09 | — |
| L/16 | pBE | 1 | 2 | 34.08 ±0.21 | **19.57** ±0.16 | **15.21** ±0.14 | **11.75** ±0.11 | 79.50 ±0.07 | **81.09** ±0.04 | **81.48** ±0.03 | **81.76** ±0.02 | **6.74** |
| | down-DE | 1 | 2 | **32.98** ±0.42 | 19.65 ±0.16 | 15.44 ±0.16 | 11.92 ±0.14 | **79.09** ±0.13 | 81.10 ±0.04 | 81.54 ±0.04 | 81.79 ±0.03 | 86.03 |
| | V-MoE | 2 | 1 | **33.98** ±0.28 | 20.42 ±0.12 | 16.20 ±0.08 | 12.73 ±0.13 | **79.50** ±0.10 | 81.30 ±0.03 | 81.72 ±0.02 | 81.97 ±0.03 | — |
| L/32 | pBE | 1 | 2 | 36.71 ±0.20 | **22.14** ±0.16 | **17.79** ±0.05 | **13.97** ±0.09 | 80.51 ±0.07 | **81.77** ±0.04 | **82.11** ±0.01 | **82.26** ±0.02 | **6.54** |
| | down-DE | 1 | 2 | **36.15** ±0.28 | 22.18 ±0.15 | 17.81 ±0.14 | 14.00 ±0.12 | **80.45** ±0.14 | 81.79 ±0.04 | 82.13 ±0.03 | 82.27 ±0.03 | 85.29 |
| | V-MoE | 2 | 1 | **36.56** ±0.34 | 23.00 ±0.12 | 18.86 ±0.07 | 15.02 ±0.16 | **80.56** ±0.13 | 82.00 ±0.03 | 82.37 ±0.02 | 82.50 ±0.03 | — |
| B/16 | pBE | 1 | 2 | **38.60** ±0.38 | **21.93** ±0.17 | **17.43** ±0.13 | **13.62** ±0.08 | 81.14 ±0.11 | **81.74** ±0.04 | **82.05** ±0.03 | **82.19** ±0.02 | **12.61** |
| | down-DE | 1 | 2 | 38.96 ±0.36 | 23.36 ±0.23 | 18.99 ±0.12 | 14.84 ±0.11 | 81.39 ±0.10 | 82.09 ±0.06 | 82.40 ±0.03 | 82.46 ±0.02 | 75.05 |
| | V-MoE | 2 | 1 | **37.76** ±0.15 | 23.16 ±0.09 | 18.79 ±0.14 | 14.94 ±0.16 | **80.97** ±0.05 | 82.04 ±0.02 | 82.36 ±0.03 | 82.48 ±0.04 | — |
| B/32 | pBE | 1 | 2 | 43.39 ±0.33 | **26.51** ±0.11 | **21.18** ±0.17 | **16.82** ±0.09 | 82.86 ±0.11 | **82.91** ±0.03 | **82.93** ±0.04 | **82.90** ±0.02 | **11.54** |
| | down-DE | 1 | 2 | **41.09** ±0.31 | 26.48 ±0.24 | 21.61 ±0.22 | 17.20 ±0.13 | **82.24** ±0.13 | 82.89 ±0.06 | 83.03 ±0.05 | 82.99 ±0.02 | 73.59 |
| | V-MoE | 2 | 1 | 42.50 ±0.28 | 27.44 ±0.19 | 22.73 ±0.19 | 18.60 ±0.14 | 82.66 ±0.08 | 83.16 ±0.03 | 83.29 ±0.05 | 83.29 ±0.04 | — |
| S/32 | pBE | 1 | 2 | 52.91 ±0.27 | 34.28 ±0.19 | 27.54 ±0.18 | 21.85 ±0.11 | 85.89 ±0.08 | 84.81 ±0.04 | 84.39 ±0.04 | 84.01 ±0.02 | **15.88** |
| | down-DE | 1 | 2 | **48.27** ±0.21 | **32.19** ±0.22 | **26.55** ±0.10 | **21.95** ±0.20 | **84.62** ±0.06 | **84.35** ±0.05 | **84.19** ±0.03 | **84.04** ±0.05 | 64.50 |
| | V-MoE | 2 | 1 | 49.37 ±0.19 | 33.51 ±0.17 | 28.00 ±0.14 | 23.25 ±0.15 | 85.04 ±0.06 | 84.69 ±0.04 | 84.54 ±0.03 | 84.33 ±0.03 | — |

Table 24: Few-shot comparison of V-MoE and ViT.

| | | K | M | Mean Across Datasets 1-Shot Error ↓ | 5-Shot Error ↓ | 10-Shot Error ↓ | 25-Shot Error ↓ | Weighted Mean Across Datasets 1-Shot Error ↓ | 5-Shot Error ↓ | 10-Shot Error ↓ | 25-Shot Error ↓ |
|---|---|---|---|---|---|---|---|---|---|---|---|
| H/14 | V-MoE | 1 | 1 | **33.04** ±0.32 | **19.23** ±0.35 | **15.63** ±0.25 | **12.06** ±0.36 | **79.05** ±0.12 | **81.00** ±0.09 | **81.59** ±0.06 | **81.82** ±0.08 |
| | ViT | - | 1 | 35.78 ±0.41 | 20.61 ±0.15 | 16.78 ±0.12 | 13.62 ±0.24 | 79.97 ±0.16 | 81.37 ±0.04 | 81.87 ±0.03 | 82.16 ±0.05 |
| L/16 | V-MoE | 1 | 1 | **34.15** ±0.27 | **20.32** ±0.12 | **16.14** ±0.12 | **12.71** ±0.15 | **79.54** ±0.08 | **81.27** ±0.03 | **81.71** ±0.03 | **81.97** ±0.03 |
| | ViT | - | 1 | 36.52 ±0.20 | 21.79 ±0.12 | 17.51 ±0.08 | 14.07 ±0.14 | 80.38 ±0.05 | 81.67 ±0.03 | 82.03 ±0.02 | 82.27 ±0.03 |
| L/32 | V-MoE | 1 | 1 | **36.83** ±0.27 | **23.05** ±0.08 | **18.78** ±0.11 | **14.95** ±0.07 | **80.67** ±0.10 | **82.01** ±0.02 | **82.35** ±0.02 | **82.48** ±0.02 |
| | ViT | - | 1 | 38.22 ±0.31 | 23.97 ±0.14 | 19.56 ±0.13 | 15.75 ±0.14 | 81.10 ±0.10 | 82.25 ±0.04 | 82.53 ±0.03 | 82.65 ±0.03 |
| B/16 | V-MoE | 1 | 1 | **39.22** ±0.27 | **24.42** ±0.13 | **20.01** ±0.11 | 15.94 ±0.18 | **81.47** ±0.08 | **82.36** ±0.04 | **82.64** ±0.03 | 82.70 ±0.04 |
| | ViT | - | 1 | 41.29 ±0.14 | 25.03 ±0.10 | **20.08** ±0.10 | 15.87 ±0.17 | 82.16 ±0.04 | 82.51 ±0.03 | **82.65** ±0.02 | 82.68 ±0.04 |
| B/32 | V-MoE | 1 | 1 | **42.37** ±0.31 | **27.60** ±0.20 | **22.89** ±0.19 | **18.46** ±0.10 | **82.64** ±0.11 | **83.19** ±0.05 | **83.33** ±0.05 | **83.26** ±0.02 |
| | ViT | - | 1 | 44.60 ±0.22 | 28.20 ±0.17 | 22.97 ±0.12 | 18.86 ±0.15 | 83.35 ±0.08 | 83.35 ±0.04 | 83.35 ±0.03 | 83.35 ±0.03 |
| S/32 | V-MoE | 1 | 1 | **49.60** ±0.28 | **33.34** ±0.13 | **27.88** ±0.11 | **23.30** ±0.18 | **85.09** ±0.08 | **84.64** ±0.03 | **84.50** ±0.03 | **84.33** ±0.04 |
| | ViT | - | 1 | 54.16 ±0.16 | 36.25 ±0.18 | 29.88 ±0.17 | 24.42 ±0.13 | 86.32 ±0.05 | 85.26 ±0.04 | 84.90 ±0.04 | 84.54 ±0.03 |

## I    FROM BATCH ENSEMBLES TO SPARSE MOES

Wen et al. (2019) have shown that, given a batch of $B$ inputs $\boldsymbol{X} \in \mathbb{R}^{B \times P}$, a single forward pass can efficiently compute the predictions of all the ensemble members $\{f(\boldsymbol{X}; \boldsymbol{\theta}_m)\}_{m=1}^{M}$. By appropriately tiling the inputs of the network $\boldsymbol{X}_{\text{tiled}} \in \mathbb{R}^{(M \cdot B) \times P}$ by a factor $M$, each internal operation per ensemble member can then be vectorized.

We take the previous example of a dense layer with parameters $\boldsymbol{U} \in \mathbb{R}^{D \times L}$ and we assume the layer receives the tiled inputs $\{\boldsymbol{H}_m\}_{m=1}^{M}$ where $\boldsymbol{H}_m \in \mathbb{R}^{B \times D}$. We need to compute for each ensemble member $\boldsymbol{H}_m \boldsymbol{U}_m = \boldsymbol{H}_m[\boldsymbol{U} \circ (\boldsymbol{r}_m \boldsymbol{s}_m^\top)]$. Denoting by $\boldsymbol{h}_{i,m} \in \mathbb{R}^D$ the $i$-th input in $\boldsymbol{H}_m$, we have

$$\boldsymbol{h}_{i,m}^\top \boldsymbol{U}_m = \sum_{e=1}^{E} g_e(\boldsymbol{h}_{i,m}) \cdot \text{expert}_e(\boldsymbol{h}_{i,m}) \quad \text{with } M = E, \quad \begin{cases} g_e(\boldsymbol{h}_{i,m}) = 1 \text{ if } e = m, \\ g_e(\boldsymbol{h}_{i,m}) = 0 \text{ otherwise} \end{cases} \tag{5}$$

and $\text{expert}_e(\boldsymbol{z}) = \boldsymbol{z}^\top[\boldsymbol{U} \circ (\boldsymbol{r}_e \boldsymbol{s}_e^\top)]$. Although (5) may appear as a convoluted way of writing the operations in batch ensembles, it unveils a connection with (1). Indeed, operations in batch ensembles can be seen as a specific sparse MoE, e.g., with binary routing weights depending only on the position in the tiled inputs. While Wen et al. (2019) primarily tiled the inputs for the sake of efficiency, it also induces some form of conditional computation, an insight that we exploit here.

## J    NEW RESULTS FOR THE REBUTTAL

### J.1    NEW RESULTS FOR THE FEATURE-LEVEL VERSUS PREDICTION-LEVEL ABLATION

Table 25 shows our updated and expanded results for the feature-level versus prediction-level ablation, described in Section 3.2. The original conclusion—i.e., ensembling at the prediction level is helpful for calibration but that the naive multi-head approach provides worse error, NLL, and diversity—seems to hold for the $K = 8$ case.

Note that the results for $K = 2$ and $K = 4$ differ slightly from the original version in Table 2 due to a change in experimental setup. In the original version, we matched $K$ for the up- and downstream models in the multi-head case (i.e., the pretrained and fine-tuned models both use either $K = 2$ or $K = 4$). We did this to be as fair as possible to the multi-head model. While it is known, from Riquelme et al. (2021), that V-MoE models are fairly robust to the choice of upstream model (e.g., $K_{\text{UPSTREAM}}$ can be set to 2 and we can just change $K_{\text{DOWNSTREAM}}$, which is the approach we take throughout our paper–see Appendix C), this was not known for the multi-head model. Unfortunately, due to time constraints we are unable to train an upstream model with $K = 8$. Thus, for the updated results we have opted for a single upstream model with $K = 2$ in all cases. As a side observation, we can notably observe that the multi-head variant is more sensitive than V-MoE to the choice of the upstream model, e.g., the performance worsens from $(K_{\text{UPSTREAM}}, K_{\text{DOWNSTREAM}}) = (4, 4)$ to $(K_{\text{UPSTREAM}}, K_{\text{DOWNSTREAM}}) = (2, 4)$.

Table 25: New feature-level vs. prediction-level ensembling ablation results. ImageNet performance of V-MoE and a naive multi-head variant (means $\pm$ standard errors over 8 replications). All models have a ViT-B/32 architecture. For the multi-head variant the last MoE layer is modified as in (2)

|  | K | NLL ↓ | ERROR ↓ | ECE ↓ | KL ↑ |
|---|---|---|---|---|---|
| V-MoE | 2 | **0.638** $\pm 0.001$ | **16.76** $\pm 0.05$ | 0.033 $\pm 0.001$ | — |
| Naive Multi-head | 2 | **0.636** $\pm 0.001$ | 17.16 $\pm 0.02$ | **0.024** $\pm 0.000$ | 0.032 $\pm 0.001$ |
| V-MoE | 4 | **0.636** $\pm 0.001$ | **16.70** $\pm 0.04$ | 0.034 $\pm 0.001$ | — |
| Naive Multi-head | 4 | 0.645 $\pm 0.001$ | 17.39 $\pm 0.04$ | **0.021** $\pm 0.000$ | 0.011 $\pm 0.001$ |
| V-MoE | 8 | **0.635** $\pm 0.002$ | **16.72** $\pm 0.06$ | 0.028 $\pm 0.001$ | — |
| Naive Multi-head | 8 | 0.650 $\pm 0.001$ | 17.50 $\pm 0.03$ | **0.021** $\pm 0.000$ | 0.005 $\pm 0.000$ |

### J.2    NEW RESULTS FOR THE STATIC VERSUS ADAPTIVE ABLATION

Figure 20 repeats the experiment of Figure 2a with two new random seeds. The new results show a smoother improvement in LL as $K$ increases. In particular, we see that there is nothing special

about $K = 5$ or $K = 6$. This indicates that the anomalous drop at $K = 6$ in the original results was simply due to noise in the training of the upstream model. We note that, like Riquelme et al. (2021), we have observed that training noise is more pronounced in the smallest (i.e., S/32) models.

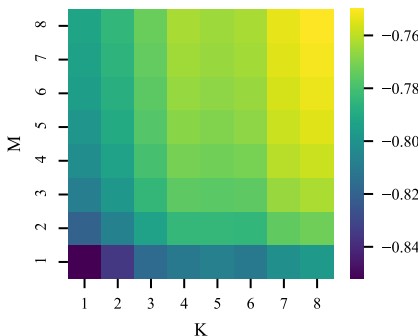

Figure 20: Replication of Figure 2a, averaged over two new random seeds, showing the effect on LL of increasing static ($M$) and adaptive ($K$) ensembling. ImageNet performance for ViT-S/32 models. **Yellow** indicates better performance; **purple** indicates worse performance.

### J.3    NEW LOAD BALANCING ABLATION

Figure 21 shows a new experiment exploring the effect of the load balancing loss on diversity and predictive performance. We see that for both V-MoE and pBE the predictive performance, as measured by NLL, Error, and ECE are largely insensitive to the strength of the load balancing loss. Similarly, the diversity of the predictions of pBE—as measured by the KL—mildly depends on that regularization. Only when the load balancing loss strength becomes excessively large (4 orders of magnitude larger than standard values) do all the performance metrics plummet. This hyperparameter does not allow us to *increase* the diversity and thereby the predictive performance of our models.

### J.4    NEW EXPERT INITIALIZATION ABLATION

Figure 22 shows a new experiment exploring the influence of the initialization of the experts. In particular, we add Gaussian noise with varying standard deviations to the initial weights of the expert MLPs. We notably show that more diverse initializations (larger standard deviations) do not translate to any clear performance gain. Note that this new experiment takes place *upstream*, since downstream the experts are already initialized (we just fine-tune them from the upstream checkpoint).

### J.5    NEW PARAMETER COUNTS TABLE

Table 26 compares the parameter counts for ViT and V-MoE/pBE models in each ViT family.

|  | S/32 | B/32 | B/16 | L/32 | L16 | H/14 |
|---|---|---|---|---|---|---|
| ViT | 36.5M | 102.1M | 100.5M | 325.3M | 323.1M | 655.8M |
| V-MoE/pBE | 166.7M | 395.0M | 393.3M | 845.8M | 843.6M | 2688.6M |

Table 26: Parameter counts for ViT vs V-MoE and pBE.

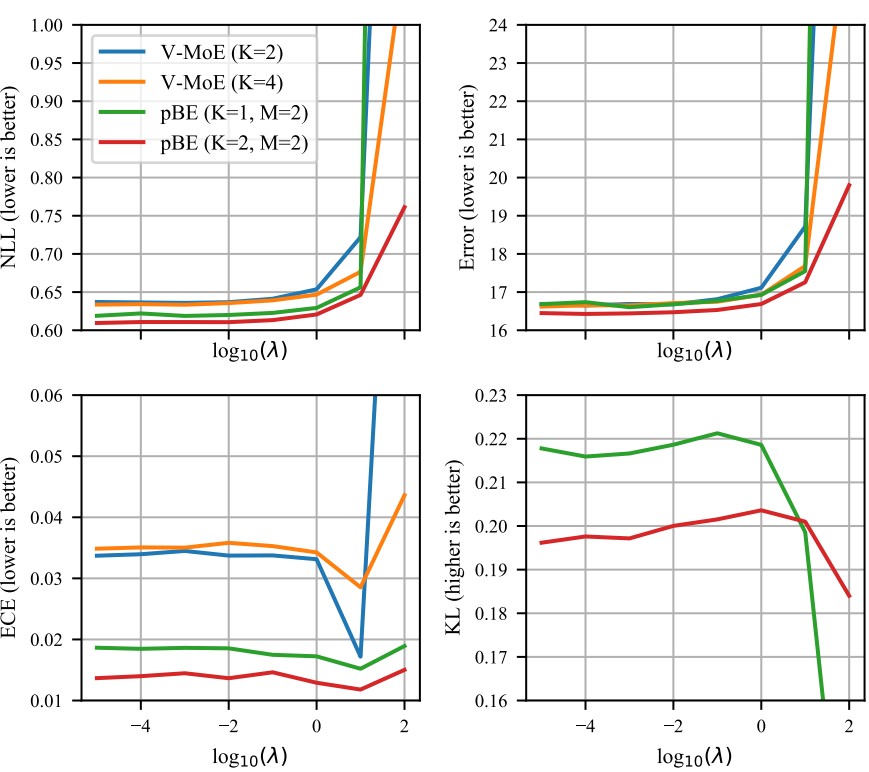

Figure 21: The effect of $\lambda$, which controls the strength of the load balancing loss as described in appendix A of Riquelme et al. (2021), on NLL, Error, ECE and diversity, for pBE and V-MoE. The resuls are averaged over three random seeds. All models have a ViT-B/32 architecture.

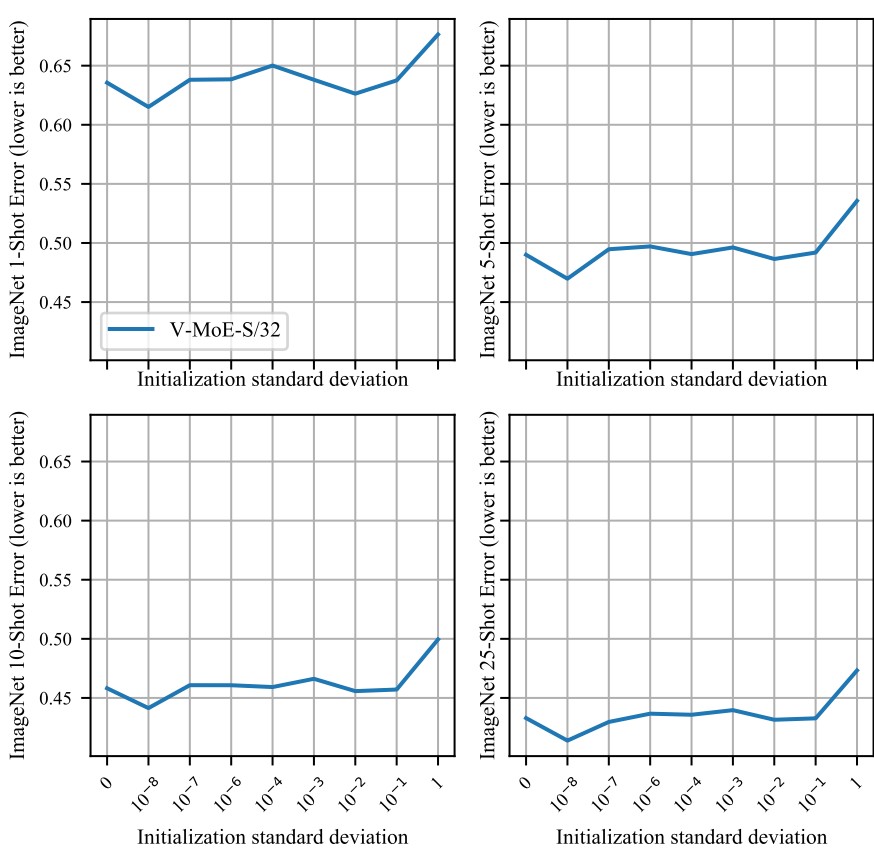

Figure 22: The influence of the noise standard deviation of the expert MLPs' initial random weights. The models are trained on JFT-300M and we measure the ImageNet few-shot Error as in Riquelme et al. (2021). Results are for a single random seed.

