# OpenReview forum: "Sparse MoEs meet Efficient Ensembles"
_ICLR.cc/2022/Conference — ICLR 2022 Submitted_

### Official Review · Reviewer_5srE · 2021-10-19

**Correctness:** 4
**Technical Novelty And Significance:** 2
**Empirical Novelty And Significance:** 1
**Recommendation:** 5
**Confidence:** 5

**Main Review:**

The paper is well written and the experiments are comprehensive. The main concerns about this paper are as follows.
1. The paper lacks novelty. It just combines the MoE with Ensemble in a trivial manner. The tiling technique is quite simple and it is just the classical way of doing ensemble across models.
2. The improvement is quite limited. In Table 3, I find although the proposed method achieves gains across different metrics e.g. NLL, Error, and ECE, the improvement in terms of accuracy is less than 0.5% which is quite small and cannot justify the effectiveness of the method. Furthermore, when compared to ViT and V-MoE in Figure 4,5,6, I can only find the curve for NLL instead of Error, which I think is more critical.
3. The number of parameters should be reported and compared to ViT, since MoE is a parameter-consuming method and we cannot ignore this factor.
4. The FLOPs cost by tiling and partitioning is another concern. More details about how to compute FLOPs for the proposed method and V-MoE should be clarified. For V-MoE, it is possible to first compute the coefficient for each expert and then synthesize the experts before extracting features. Therefore, it can save a large scale of computational costs. But due to tiling and partitioning, each branch should cause its own computation, which I think will cost a lot of extra computation and let me doubt the efficiency of the proposed method. I wonder whether the author computes the FLOPs with the optimal way.

**Summary Of The Paper:**

This paper considers the combination of MoE and Ensembles for image recognition tasks to obtain both improvement of classification accuracy as well as stability of prediction. The proposed method is evaluated with extensive expereiments from large-scale image recognition to OOD and the results indicate better performance for the proposed method.

**Summary Of The Review:**

The paper still has a big space for further improvement. I recommend rejecting it this time.

---

> ### Author Response · Authors · 2021-11-15
> **Response to Reviewer 5srE**
>
> Thank you for your comments and questions! Please let us know if anything is unclear.
>
> > The paper lacks novelty. It just combines the MoE with Ensemble in a trivial manner. The tiling technique is quite simple and it is just the classical way of doing ensemble across models.
>
> Our method is not a trivial combination. Firstly, we note that tiling alone is not enough to provide good performance, it must be combined with partitioning, as shown in section 4.1. Secondly, we note that the idea of using partitioning as an alternative to the fast weights of a BE is a novel idea which is specialized to V-MoEs. Finally, we note that efficient ensembles and sparse MoEs are in general a non-trivial combination. Further, in Appendix F we show that other complex combinations of efficient ensemble and V-MoEs perform poorly compared to pBE. For instance, we show the surprising result that the MIMO + V-MoE performs worse in many cases than vanilla V-MoE.
>
> > The improvement is quite limited. In Table 3, I find although the proposed method achieves gains across different metrics e.g. NLL, Error, and ECE, the improvement in terms of accuracy is less than 0.5% which is quite small and cannot justify the effectiveness of the method.
>
> While a 0.5% increase in accuracy is modest (although non-trivial), when coupled with improvements in all other metrics, represents a substantial overall practical advance. In particular, for safety critical applications such as self-driving cars and medical diagnosis, better performance in uncertainty estimation (e.g. NLL, ECE, OOD detection) is crucial. Furthermore, we note that Table 3 is only a narrow ablation study on a single dataset with a single ViT family. We also note that looking at a raw percentage improvement without context is not necessarily illustrative. The results for our full evaluation in Section 5 and Appendix H.5 show cases where pBE improves over the error of V-MoE more significantly. For example, the ImageNet (and ImageNet variants) results in Table 13 indicate that pBE with larger ViT families improves over the error of V-MoE as much as deep ensembles. Similarly, Tables 16 and 17, show that pBE is on par with deep ensembles for both large and small ViT families (while being computationally much cheaper).
>
> > Furthermore, when compared to ViT and V-MoE in Figure 4,5,6, I can only find the curve for NLL instead of Error, which I think is more critical.
>
> Please see appendix H, Figures 9 and 13-17, as well as tables 13-18 for results with error.
>
> > The number of parameters should be reported and compared to ViT, since MoE is a parameter-consuming method and we cannot ignore this factor.
>
> We note that pBE requires no additional parameters compared to V-MoE and that this comparison for ViT and V-MoE can be found in Riquelme et al. (2021). However, we agree that this is an important factor, and have added the information in Table 26 (Appendix J.5) of our updated manuscript. Thanks!
>
> > The FLOPs cost by tiling and partitioning is another concern. More details about how to compute FLOPs for the proposed method and V-MoE should be clarified. For V-MoE, it is possible to first compute the coefficient for each expert and then synthesize the experts before extracting features. Therefore, it can save a large scale of computational costs. But due to tiling and partitioning, each branch should cause its own computation, which I think will cost a lot of extra computation and let me doubt the efficiency of the proposed method. I wonder whether the author computes the FLOPs with the optimal way.
>
> Neither tiling nor partitioning prevents the routing weights from being computed beforehand and then activating only the appropriate experts in a sparse manner. In both cases, the router first decides which K < E experts need to be applied to each token, and then only those experts are applied to the token. The partitioning simply limits the choice of experts for each input token. The tiling, on the other hand, increases the total number of tokens. We further note that the tiling happens only before the first p-MoE block (which is typically near the very end of the network) which results in a minimal increase in flops over V-MoE compared with deep ensembles due to an increase in tokens (see Tables 12 and 13).
>
> To calculate the FLOPs we used a TPU XLA profiler (see [here](https://cloud.google.com/tpu/docs/cloud-tpu-tools#profile_tab) and [here](https://github.com/tensorflow/tensorflow/blob/master/tensorflow/compiler/xla/service/hlo_cost_analysis.cc)). ​​The XLA program is analyzed when compiled and all floating point operations are considered. In other words, the FLOPs count is automated. This means that our FLOPs include things like data conversion (e.g. float64 to float32) and even "wasted" computation on padded data (XLA will pad arrays to have columns/rows multiple of 8/128).

---

> > ### Comment · Reviewer_5srE · 2021-11-27
> > **Response**
> >
> > Thanks for your response. However, I'm not quite convinced, especially the computation of FLOPs. In Figure 2 bottom, the features are first tiled. For instance, $h_{11}$ and $h_{12}$ are obtained based on $h_1$. But both $h_{11}$ and $h_{12}$ have to be processed separately to generate $\hat{h_{11}}$ and $\hat{h_{12}}$, which doubles the computational cost. I do not see any explicit way to precompute the kernel and then use the synthesized kernel to derive both $\hat{h_{11}}$ and $\hat{h_{12}}$. This is the main reason that I thought the proposed method is not as efficient as MoE.

---

> > > ### Author Response · Authors · 2021-11-29
> > > **FLOPs clarification**
> > >
> > > Regarding FLOPs values. pBE *does* require more FLOPs than V-MoE. And indeed, this is due to the tiling of the features. However, please note that the tiling only happens towards the end of the network because of the "Last-$n$" structure of the underlying V-MoE. Thus, the increase in FLOPs compared to V-MoE is fairly small, especially when compared with a standard deep ensemble. (Please see Table 12 of our paper, which shows that pBE increases the FLOPs of V-MoE by ~15% to ~40%, and uses ~30% to ~42% fewer FLOPs than a deep ensemble of size 2, depending on the ViT family). Furthermore, please note that **all** of our figures include this additional computational cost for pBE.
> > >
> > > Regarding the precomputation of experts. In a standard V-MoE's experts layer, we first process each token to get routing weights, and then each token is routed to only the top-$k$ experts. In other words, we sparsely activate the *only* $k$ experts for each of the input tokens. In the pBE case, there are two differences: tiling and partitioning. Tiling means that there are twice as many tokens. However, these tokens are treated independently and thus the tiling has no impact on our ability to sparsely activate the experts---we simply need to activate twice as many experts. The partitioning simply means that each token can be routed to a reduced subset of the experts. This too does not have any impact on our ability to sparsely activate experts. This is all to say that the only difference in FLOPs for pBE vs V-MoE is the increased number of tokens that occurs before the first experts layer.
> > >
> > > Please let us know if there is still some confusion and apologies if the above description was all clear already. We are very confident that the FLOPs calculation should not be an issue, but we are not sure where the concerns are coming from. It seems that everyone is on the same page: pBE does require more FLOPs than V-MoE and this is due to the tiling that happens before the first experts layer. There is no other fundamental difference between pBE and V-MoE in terms of FLOPs cost. The additional FLOPs cost has been shown in all of our figures.

---

### Official Review · Reviewer_cawf · 2021-11-02

**Correctness:** 3
**Technical Novelty And Significance:** 2
**Empirical Novelty And Significance:** 2
**Recommendation:** 3
**Confidence:** 5

**Main Review:**

Strengths:

The paper provides confirmation that MoEs and ensembles have complementary (or additive) benefits. Moreover, the proposed approach (partitioned batch ensembles), which combines MoEs and ensembling, achieves the benefits of both techniques simultaneously.

The paper provides the first study of the robustness of vision sparse MoE’s (parallel submission).

Weaknesses:

The proposed approach is incremental. In the words of the authors, the method “introduces two changes to V-MoEs: (a) the partitioning of the experts and (b) the tiling of the representations.”

The paper focuses on the comparison of specific architectures and training procedures. In this regard, conclusions one may be able to draw from the paper are too constrained. For example, can the benefit of “static” ensembles be achieved by “adaptive” ensembling? While this paper’s results suggest perhaps not, I don’t think this is conclusive. [The distinction between static and adaptive may be generalized in such a way as to enable finer control on aspect and degree of adaptation with implications to, e.g., the diversity and uncertainty calibration at the output of the ensemble.] Further, e.g., (Wen et al. 2020) claim that “diversity of initialization entirely determines the diversity of ensembling system.” The present paper notes ways to achieve diversity given a particular distinction between static vs adaptive ensembling, and training procedure. There are other dimensions (e.g., initialization, example balancing) which have direct relevance to diversity and thus, uncertainty calibration, which are not explored in this paper.

Section 3.1 is an ablation study just like section 4.2 is. I find the discussion going back and forth to be detrimental to clarity and efficiency of communication.

Section 3.2 compares a naive multi-head method -- without specifying how it is trained -- with a “standard” vision (sparse) MoE. I don’t find the results there clear enough to be compelling. For example, whether the multi-head variant is able to provide diverse predictions is more likely to do with the training procedure than with the architecture -- this is related, e.g., to the “load balancing” during training used in (Riquelme et al. 2021). Besides, the claims in section 3.2 are conflicting: we are told naive multi-head improves ECE but also that a different strategy is required for uncertainty calibration.

The experimental results (section 5) are simply stated. For example, most of the time (static) ensembling helps but when it doesn’t (e.g., out of distribution performance) the paper provides no discussion or insight.

Some acronyms are used without introduction (the introduction appears after the first usage), e.g., OOD and NLL in page 4.

It’s not clear to me how the static ensembling portion of the proposed approach is “efficient” (section 4.1). For example, Batch Ensemble (Wen et al. 2020) generates the family of ensemble weights via element-wise multiplication of a shared matrix and “fast-weights.” In my understanding, in the case of the proposed approach any sharing of parameters is done within the ensemble members.

How are the dashed lines in figures 4 and 5 pareto frontiers?

**Summary Of The Paper:**

The paper investigates the benefits of combining (Sparse) Mixture of Experts (MoE) and ensembling. Sparse MoE’s employ conditional computation to reduce computational and environmental costs of DNNs while maintaining (or increasing) performance. On the other hand, ensembling models has been shown to achieve the highest robustness in the presence of dataset shift, e.g., higher accuracy and better estimation of uncertainty.

The present submission empirically demonstrates that Sparse MoE’s can be ensembled to attain the benefits of both techniques simultaneously, i.e., conditional computation (with its implications to scalability) together with robustness in the presence of dataset shift. The main components of the approach are: 1) Disjoint MoE’s as ensemble members. 2) Tiling of representations which enables all ensemble members to compute the output for a batch in a single forward pass.

A number of experiments compare predictive performance vs computational cost for the proposed approach (pBE), vision Sparse MoE’s (V-MoE) and Vision Transformers (ViT). The results show that, under some conditions, pBE displays the known benefits of Sparse MoE’s and ensembling, which leads to performance gains w.r.t. ViT and V-MoE under metrics like accuracy, negative log-likelihood (NLL) and expected calibration error (ECE).

**Summary Of The Review:**

Overall I lean towards rejection. While the paper presents experimental results likely not found elsewhere, in my perception, that appears to be the sole reason for the paper. Specifically, the paper presents experimental results on the robustness of V-MoE’s (vision sparse MoE’s) and adds ensembling on top of the V-MoE model to increase robustness. However, no further insight or development (e.g., in architecture or optimization) is provided.

---

> ### Author Response · Authors · 2021-11-15
> **Response to Reviewer cawf (part 1)**
>
> Thank you for your constructive comments, which lead us to run several additional experiments.
>
> > The paper provides the first study of the robustness of vision sparse MoE’s (parallel submission).
>
> We note that the “Scaling Vision with Sparse Mixture of Experts” is not a parallel submission, but this indeed is the first study of their robustness. It seems to have been accepted at NeurIPS 2021.
>
> > The proposed approach is incremental. In the words of the authors, the method “introduces two changes to V-MoEs: (a) the partitioning of the experts and (b) the tiling of the representations.”
>
> We do not believe that our approach is incremental. Firstly, we note that these two components of pBE (tiling and partitioning) interact in a non-linear way–on their own, neither significantly improves over the baseline (see Table 3). While the idea of tiling the inputs comes from BE, we do not use their fast weights approach for constructing ensemble members. Instead, we provide a novel algorithmic contribution in the form of partitioned experts. It is not obvious a priori that this should work. Furthermore, pBE employs a form of tiling tailored to the structure of “Last-n” V-MoE (see end of Section 4.1). Please see our general response to this point for more details.
>
> > The paper focuses on the comparison of specific architectures and training procedures. In this regard, conclusions one may be able to draw from the paper are too constrained.
>
> We focus on specific architectures and training procedures. However, we note that given the scale of our experiments a wider range of architectures and training procedures would not have been feasible. For some context, please note the substantial computational budgets required for training ViT and V-MoE upstream models, as described in Table B.5 of Riquelme at al. (2021). We also note that our training procedure was *intentionally* made as similar as possible to ViT and V-MoE. We did this to show that our findings apply in a setting that is close to the existing standard training regime and that no tricks are required for strong performance of pBE. Furthermore, we would argue that transformer-based architectures are widely used and gaining in popularity. Thus, improvements to their robustness and efficiency have wide applicability both for practitioners and researchers. Finally, we note that while sparse MoEs have found huge success in transformer architectures, they have seen relatively little success in other common architectures. For example, to the best of our knowledge there are no successful applications of sparse MoEs to ResNets. At the time we initiated this project V-MoE was, to the best of our knowledge, the first and only sparse MoE architecture available for vision.
>
> > For example, can the benefit of “static” ensembles be achieved by “adaptive” ensembling? While this paper’s results suggest perhaps not, I don’t think this is conclusive.
>
> Our paper makes no claims about whether the benefit of “static” ensembles can be achieved by “adaptive” ensembling. Our conclusion is simply that “static” and “adaptive” ensembles are (i) complementary axes, and (ii) “adaptive” ensembles are more cost-effective (see Figure 2-c). While our experimental setup in section 3 is fairly narrow, we believe that this conclusion is also supported by our very extensive experiments with deep ensembles of V-MoEs, which use multiple ViT families, datasets, and performance metrics. Are there other specific take-aways that you believe are limited by the set of architectures/training procedures?
>
> > [The distinction between static and adaptive may be generalized in such a way as to enable finer control on aspect and degree of adaptation with implications to, e.g., the diversity and uncertainty calibration at the output of the ensemble.]
>
> Unfortunately we are not sure what is meant by this statement. Could you please elaborate? Perhaps you were suggesting that it is possible to vary the sparsity of the experts in only some of the layers, in order to have finer control?

---

> > ### Author Response · Authors · 2021-11-15
> > **Response to Reviewer cawf (part 2)**
> >
> > > Further, e.g., (Wen et al. 2020) claim that “diversity of initialization entirely determines the diversity of ensembling system.” The present paper notes ways to achieve diversity given a particular distinction between static vs adaptive ensembling, and training procedure. There are other dimensions (e.g., initialization, example balancing) which have direct relevance to diversity and thus, uncertainty calibration, which are not explored in this paper.
> >
> > Other directions for achieving diversity, such as initialization and load balancing, are certainly interesting directions for further research. We believe that they are orthogonal to our work, in the sense that techniques for better initialization (for example) of ensemble members could be combined with pBE.
> > Following the comment of the reviewer, we ran additional experiments and we summarize them below. We also refer to the existing studies of the paper related to diversity.
> >
> > - **Initialization**:  In  Figure 22 (Appendix J.4) in the updated version of our paper, we have added an experiment showing the influence of the initialization of the experts. In particular, we add Gaussian noise with varying standard deviations to the initial weights of the expert MLPs. We notably show that more diverse initializations (larger standard deviations) do not translate to any clear performance gain. Note the experiment takes place **upstream**, since downstream the experts are already initialized (we just fine-tune them from the upstream checkpoint).
> >
> >   We also would like to draw the attention of the reviewer that our investigation into “upstream” deep ensembles of V-MoE (Appendix G) does explore the importance of initialization diversity. Our findings agree with Wen et al. (2020); we see that the ensembles made from different upstream models (and which therefore have more diverse initial weights at the beginning of finetuning) provide much more diverse predictions and thus better predictive performance.
> >
> > - **Example balancing**: In Figure 21 (Appendix J.3) in the updated version of our paper, we have added an analysis showing how the performance and diversity of the predictions vary with respect to the strength of the load balancing (i.e. example balancing) loss used by Riquelme et al., 2021. We thank the reviewer for that suggestion. In a nutshell, for both V-MoE and pBE the performance (NLL, Error, ECE) are pretty insensitive to the strength of this loss term. Similarly, the diversity (as measured by the KL) of the predictions of pBE mildly depends on the load balancing loss. Only when this loss becomes excessively large (4 orders of magnitude larger than standard values) do all the performance metrics plummet.
> >
> > - **Noise in the routing**: Our experiment already available in Table 3 shows that the only noise in the routing (“sigma” in the table) is not sufficient to drive up the predictive performance and the diversity of the predictions.
> >
> > > Section 3.1 is an ablation study just like section 4.2 is. I find the discussion going back and forth to be detrimental to clarity and efficiency of communication.
> >
> > Both sections 3 and 4.1 contain ablation studies. However, there is a conceptual separation between them. The ablations in section 3 pertain to sparse MoEs and ensembles more generally, while section 4.1 pertains to pBE specifically. The results in section 3 help to motivate pBE as a method, so we believe that it is best that they are placed before section 4. However, we want our work to be as clear as possible – accordingly, we'll rethink the order and ablations covered in each section, and we welcome specific suggestions that make the flow smoother.

---

> > > ### Author Response · Authors · 2021-11-15
> > > **Response to Reviewer cawf (part 3)**
> > >
> > > > Section 3.2 compares a naive multi-head method -- without specifying how it is trained -- with a “standard” vision (sparse) MoE. I don’t find the results there clear enough to be compelling. For example, whether the multi-head variant is able to provide diverse predictions is more likely to do with the training procedure than with the architecture -- this is related, e.g., to the “load balancing” during training used in (Riquelme et al. 2021).
> > >
> > > We thank the reviewer for pointing to that lack of details which we hope to have clarified in the updated manuscript, in appendix B.6.2. The training procedure for the naive multi-head architecture mimics that of the other architectures we consider (V-MoE, pBE). Compared with V-MoE, the only difference is that we use the average ensemble member cross entropy, described in appendix D.1, to handle multiple predictions.
> > >
> > > The naive multi-head model is also trained with a load balancing loss. At fine-tuning time, Riquelme et al. 2021 found that (as also confirmed by our rebuttal experiments in Appendix J.3) the load-balancing loss has only little impact on the performance. We also note that because the upstream model is a standard V-MoE, the naive multi-head model should benefit from diverse experts that became specialized while being pre-trained on the upstream task (e.g., see Figure 7 in Riquelme et al. 2021). We investigated more involved approaches based on MIMO (i.e., multi-head and multi-input) in Appendix F, and found that these also provide very low diversity. Furthermore, we tried several MIMO variants specialized for V-MoEs (described in Appendix F.3) without success. Thus, we believe that our results are well supported.
> > >
> > > > Besides, the claims in section 3.2 are conflicting: we are told naive multi-head improves ECE but also that a different strategy is required for uncertainty calibration.
> > >
> > > What we meant by “These results suggest that, *while prediction level ensembling can be beneficial for uncertainty calibration of MoEs, a different strategy is required.* ” was that while the calibration has been improved for multi-head, this came at the cost of decreased classification performance. We want to have both good error and calibration. This has been clarified in the updated manuscript, and now reads “Thus, while *prediction level ensembling can be beneficial for uncertainty calibration of MoEs*, a different strategy is required such that the error is not worse”. Thanks for pointing it out.
> > >
> > > > Some acronyms are used without introduction (the introduction appears after the first usage), e.g., OOD and NLL in page 4.
> > >
> > > Thanks for pointing out these issues. We have fixed them in the updated manuscript.
> > >
> > > > It’s not clear to me how the static ensembling portion of the proposed approach is “efficient” (section 4.1). For example, BE (Wen et al. 2020) generates the family of ensemble weights via element-wise multiplication of a shared matrix and “fast-weights.” In my understanding, in the case of the proposed approach any sharing of parameters is done within the ensemble members.
> > >
> > > Our method is efficient in two ways. Firstly, we do not increase the number of weights over the baseline V-MoE model, whereas a standard (inefficient) deep ensemble would increase the number of weights by a factor of M (the ensemble size). Shared parameters in our model include all parameters outside of the experts. In a given MoE layer, the parameters specific to an ensemble member, say the m-th member, are the parameters of all the E/M experts in the m-th part of the partition. This partitioning replaces the fast weights in BE. Secondly, by keeping the number of experts fixed, our method requires only a moderate increase in compute cost compared to standard V-MoE. The only overhead comes from the tiling happening after the first MoE layer after which point the model processes M times more inputs (see Tables 13 and 14 for example). Let us know if the distinction with respect to BE is not clear yet, and we’ll be happy to provide further details.
> > >
> > > > How are the dashed lines in figures 4 and 5 pareto frontiers?
> > >
> > > These are pareto frontiers in the sense that (of the methods we benchmarked) the dashed lines represent the best possible tradeoffs between predictive performance (e.g. LL, Error, ECE) and FLOPs. There are no points which increase performance and decrease FLOPs simultaneously over the points on this line; in other words, only the points in this line are not dominated by other configurations.
> > >
> > > To be more specific, we used Scipy’s [ConvexHull function](https://docs.scipy.org/doc/scipy/reference/generated/scipy.spatial.ConvexHull.html) to determine this set of points.

---

> > > > ### Author Response · Authors · 2021-11-15
> > > > **Response to Reviewer cawf (part 4)**
> > > >
> > > > > Overall I lean towards rejection. While the paper presents experimental results likely not found elsewhere, in my perception, that appears to be the sole reason for the paper. Specifically, the paper presents experimental results on the robustness of V-MoE’s (vision sparse MoE’s) and adds ensembling on top of the V-MoE model to increase robustness. However, no further insight or development (e.g., in architecture or optimization) is provided.
> > > >
> > > > We would like to emphasize that these results are both important, given the growing adoption of transformer and sparse MoE models in a wide range of settings, and challenging to arrive at, due to the computational requirements for experiments of this scale. This model, an efficient hybrid of MoEs and ensembling, is the first architecture of its kind for vision. We provide results not only for accuracy, but also robustness and calibration. Therefore, we disagree that there is no architectural development or insight.

---

### Official Review · Reviewer_KpkQ · 2021-11-03

**Correctness:** 4
**Technical Novelty And Significance:** 3
**Empirical Novelty And Significance:** 3
**Recommendation:** 6
**Confidence:** 3

**Main Review:**

1. PBEs share the attention layers and use an ensemble of sparse MOEs for the MLP layers of Vision Transformers. Any reasons for this design choice, and not going the other way, i.e. sharing the MLPs and using Sparse MOEs for the self-attention layers ?
2. Results in Table 2 suggest prediction level ensembling can be beneficial for uncertainty calibration. The Table reports the results for K = 2 and K = 4. Does this claim hold true for K > 4 as well ?
3. The multi-head MoE presented in Section 3.2 stacks the K selected expert predictions. Are these K predictions averaged in order to compute the NLL and Error scores reported in Table 2 ?
4. Fig 2 a: For every value of M, the log likelihood score increases till K=5, and drops at K=6. Any reasons behind this ?
5. The idea of Batch Ensembling and Tiling isn't new, and has been used in previous works [A].

[A] Yeming Wen, Dustin Tran, and Jimmy Ba. Batch ensemble: an alternative approach to efficient
ensemble and lifelong learning. In ICLR, 2019.


**Summary Of The Paper:**

The authors show empirically that Sparse MOEs and Ensembles have complementary features, and suggest that combining the two should lead to improved performance. Authors build on the Vision Transformer (ViT) for their experiments. To efficiently combine Sparse MOEs and Ensembles, the paper presents Partitioned Batch Ensembles (PBE), where the parameters of the self-attention layers are shared, and an ensemble of Sparse MOEs are used for the MLP layers of the Transformer blocks.



**Summary Of The Review:**

I only have minor concerns with the paper, and I have stated those concerns in the Main Review. Other than that, the paper is novel and the experiments demonstrate the usefulness of the proposed Partitioned Batch Ensembles (PBE).

---

> ### Author Response · Authors · 2021-11-15
> **Response to Reviewer KpkQ**
>
> Thank you for your constructive comments which pushed us to run/re-run various experiments and make the paper more clear in general.
>
> > PBEs share the attention layers and use an ensemble of sparse MOEs for the MLP layers of Vision Transformers. Any reasons for this design choice, and not going the other way, i.e. sharing the MLPs and using Sparse MOEs for the self-attention layers ?
>
> This was based on the architecture choices from Riquelme et al. (2021), which was in turn based on the empirical successes of Shazeer at al. (2017) and Lepikhin et al. (2020) who took the same approach in the domain of NLP. Fedus et al. (2021) did explore this design choice, see appendix A of their paper, and found that while it did provide some performance improvements it led to unstable training. On the other hand, [Yang et al. (2021)](https://arxiv.org/pdf/2105.15082.pdf) (see section 3.4 of their paper) provide contradictory results showing that MoE attention performs worse than a standard MoE. Since it isn’t clear that MoE attention provides benefits and since it makes training less stable, we chose not to investigate this choice further.
>
> > Results in Table 2 suggest prediction level ensembling can be beneficial for uncertainty calibration. The Table reports the results for K = 2 and K = 4. Does this claim hold true for K > 4 as well ?
>
> Yes, the trends reported for K=2 and K=4—namely, multi-head improves ECE but degrades other metrics while not inducing diverse predictions—seem to hold for K=8, as shown in Table 25 (Appendix J.1) of the updated manuscript. We will replace Table 2 with Table 25 in the camera-ready version of the paper. Note that the results for K=2 and K=4 are slightly different to those in the original table. This is because we have changed the experimental setup slightly; see details in Appendix J.1.
>
> > The multi-head MoE presented in Section 3.2 stacks the K selected expert predictions. Are these K predictions averaged in order to compute the NLL and Error scores reported in Table 2 ?
>
> Yes. Furthermore, training is done with the average ensemble member cross entropy as described for pBE in appendix D1. We have clarified this in appendix B.6.2.
>
> > Fig 2 a: For every value of M, the log likelihood score increases till K=5, and drops at K=6. Any reasons behind this ?
>
> This is simply due to noise in the training of the upstream model. At this point, the gains in NLL for larger values of K have been saturated and thus the noise in the performance of the upstream model has a significant effect on the final performance. The reason that this phenomenon is present for all values of M is that all downstream ensemble members are fine-tuned **from the same upstream model**. We note that, like Riquelme at al. (2021), we have observed that training noise is more pronounced in the smallest (i.e., S/32) models. We have rerun this experiment with two new random seeds. The new results, shown in Figure 25 (Appendix J.2) show a smoother improvement in LL as K increases. In particular, we see that there is nothing special about K=5 or K=6. Thanks for pushing us to make this more clear! We will replace Figure 2a with this plot for the camera-ready version of the paper.
>
> > The idea of Batch Ensembling and Tiling isn't new, and has been used in previous works [A].
>
> This is true. We were inspired by and aimed to build upon the success of BE. We hope that this was clear in our manuscript. Specifically, we are influenced by two aspects of BE: (i) the tiling of the inputs and (ii) the sharing of the parameters across ensemble members. However, while BE uses tiling before entering the model, regardless of its potential structure, pBE employs a form of tiling tailored to the structure of “Last-n” V-MoE (see end of Section 4.1). Moreover, while BE shares parameters across members via the introduction of fast weights, we consider a partitioning of the experts, which is unique to MoEs. As we demonstrate (see Table 3 of our paper), this combination of tiling and partitioning is crucial for good performance. Please see the general response for more detail on this point.
>
> > I only have minor concerns with the paper, and I have stated those concerns in the Main Review. Other than that, the paper is novel and the experiments demonstrate the usefulness of the proposed Partitioned BEs (PBE).
>
> We are thankful that you have recognised the novelty of our paper. We also thank you for the constructive comments, which we hope that we have satisfactorily addressed. We also hope that we have clarified any other concerns. If not, please let us know and we will do our best to remedy the problems. On the other hand, if you are satisfied please consider increasing your score.

---

### Author Response · Authors · 2021-11-15
**General Response**

We thank all of the reviewers for their time and constructive comments. We are pleased that Reviewer KpkQ found our work novel, that Reviewer 5srE notes the quality of the writing, that Reviewer KpkQ and Reviewer 5srE note the strength of our experimental evaluation, that Reviewer cawf notes that our experimental results cannot be found elsewhere, and that Reviewer cawf notes that our proposed approach achieves the performance benefits of both MoEs and ensembles. We address in comments to each review the concerns raised by each reviewer.

While Reviewer KpkQ noted the novelty and significance of our proposed method, both Reviewer cawf and Reviewer 5srE were concerned with a lack of novelty and/or significance, therefore we clarify the key novelties here.

The BE/MoE model is a novel architectural development; it is the first-in-class scalable hybrid of MoEs and ensembling in Computer Vision. The partitioning of experts in order to create ensemble-member specific parameters (rather than using vanilla BE’s fast weights) is a unique architectural component of our method that is only applicable to MoEs. Furthermore, the “standard” tiling used in BE (applied before entering the model, regardless of its possible structure) is here customized in the context of the "Last-n" V-MoE architecture, in order to increase efficiency (Sec. 4.1). We summarize the differences between pBE and BE in the table below.

More importantly, naively combining efficient ensemble approaches with V-MoE **does not work** well. Appendix F shows this.

We spent a significant effort to try and get MIMO and V-MoE (as well as ViT) to work well together, without success (see Appendix F). These results demonstrate that combining efficient ensemble methods with V-MoE is highly non-trivial, and it required novel algorithmic contributions to work well. A final point is that simplicity and novelty are two orthogonal axes that should not be conflated. We view the simplicity of pBE as a strength.

| | Shared parameters | Parameters specific to the m-th ensemble member | Tiling
---|---|---|---
Batch ensemble | Kernels in each dense (when applicable, conv.) layer | Pairs of fast-weights {u_m, v_m} and bias terms for each dense (when applicable, conv.) layers | At the very beginning of the model, regardless of its internal structure
Partitioned batch ensemble | All parameters outside the MoE layers, e.g., self-attention layers and final dense layer | In each MoE layer, the parameters of all the E/M experts in the m-th part of the partition and the router for those experts | At the first MoE layer (thus saving redundant computation for “Last-n” V-MoE)

Regarding significance, we believe that our experimental results, especially those evaluating the robustness of V-MoE, are significant and worth publishing regardless of the novelty, or lack thereof, in pBE. Furthermore, we believe that the strong empirical performance of pBE, made more appealing by its conceptual simplicity, makes pBE of interest to the general machine learning community. In particular, sparse deep networks are in their infancy and, given their huge representational capacity (by design), robustness results are needed to make sure these models do not overfit or memorize the training data.


**A note regarding our updates.** All updates to the text have been highlighted in red for ease of access. Similarly, all additional experiments and results have been added to a new Appendix J. All of these additional results will be incorporated into the main text (or appendices, where appropriate) for the camera-ready version.

---

> ### Comment · Reviewer_cawf · 2021-11-22
> **Re: Initial author response**
>
> I thank the authors for the detailed and comprehensive replies. I provide here a response to some items I had not addressed (stressed in the rebuttal or from the appendixes).
>
>
> > “We do not believe that our approach is incremental. Firstly, we note that these two components of pBE (tiling and partitioning) interact in a non-linear way–on their own, neither significantly improves over the baseline (see Table 3).”
>
> The results in Table 3 are not novel: tiling w/o partition is like duplicating one model (all outputs will be similar => low diversity); partition w/o tiling produces a single prediction (zero diversity). Thus, it is expected that the only method that enables diversity will have superior performance. In my view, the more interesting investigation is into how diversity is enabled and how it could be, e.g., further encouraged to increase robustness.
>
>
> > “naively combining efficient ensemble approaches with V-MoE does not work well.”
>
> > Appendix F (comparison to other ensembling approaches)
>
> It is well-known that more diversity among ensemble members leads to increased performance (e.g., Wen et al. 2020). In your experiments your method is achieving the higher KL but this is not investigated. It would be interesting to investigate what mechanism is pushing the KL up. Is it an aspect of the proposed method? Or is it an artifact of the experimental setup which could be incorporated into other methods as well (e.g., initialization, sample balancing, etc.)?
>
>
> > “Our findings agree with Wen et al. (2020); we see that the ensembles made from different upstream models (and which therefore have more diverse initial weights at the beginning of finetuning) provide much more diverse predictions and thus better predictive performance.”
>
> > Appendix J.3 (example balancing)
>
> The load balancing loss of (Riquelme et al. 2021) seeks to balance the number of example-to-expert assignments. While example-to-expert assignment is relevant to diversity, the load balancing loss used was not designed to pursue diversity. The authors may easily find several lines of work on achieving diversity in ensembles, e.g., through losses specifically designed to achieve diversity (see e.g., https://ieeexplore.ieee.org/abstract/document/8717641/references#references).
>
>
> > Appendix J.4 (Adding gaussian noise to upstream expert MLPs)
>
> This negative result does not provide much insight.
>
>
> In summary, my view of the submission remains mostly unchanged. The experiments presented are of interest but they need to be interpreted. Moreover, further investigation into what enables the proposed method to attain diversity (aka superior performance) -- and what (if anything) prevents previous ensemble methods from attaining similar diversity -- would be essential. Providing that kind of insight would significantly strengthen the present submission.

---

> > ### Author Response · Authors · 2021-11-25
> > **Additional investigation into ensemble diversity and individual model performance**
> >
> > Ensemble diversity and individual member performance. The following tables show the individual member performance for each of our efficient ensemble variants as well as upstream and downstream deep ensembles, for sizes M=2 and M=4. In each table, the first row is the combined ensemble, and the following rows are its individual members. The tables make it possible to view the impact of diversity through a different lens.
> >
> > We make the following key observations:
> > 1. For pBE, the gap between single members and ensembles is considerable. This is reminiscent of deep ensembles (both upstream and downstream) and in stark contrast with what we observe for the other efficient ensembles BE ViT and MIMO V-MoE.
> > 2. The diversity of pBE is comparable to that of upstream deep ensembles and much larger than downstream deep ensembles.
> >
> > For example, if we compare MIMO V-MoE (K=2) (BR=1) (M=2) and pBE (K=1) (M=2), the single members of pBE are all worse in NLL, ACC, and ECE than the single members of MIMO. However, the performance gap between the individual members and the ensemble is much larger for pBE than MIMO (where there is almost no difference). The diversity of the individual models in pBE is key to its strong performance. Thus, pBE outperforms MIMO.
> >
> > In short, this suggests that pBE approximates a deep ensemble of smaller V-MoE models. That is, with pBE, we are able to take advantage of the overparameterization of the experts to create a rich set of ensemble members within a single model. Each ensemble member has a large number of non-shared parameters, thus high induced diversity. In comparison, BE only has a few vectors specific to each member.

---

> > > ### Author Response · Authors · 2021-11-25
> > > **Tables**
> > >
> > > | model_type        |   im/nll |   im/acc |    im/ece |     im/kl |
> > > |:------------------|---------:|---------:|----------:|----------:|
> > > | BE ViT (M=2) average | 0.681832 | 0.815318 | 0.0209275 |
> > > | BE ViT (M=2) 0    | 0.692583 | 0.813157 | 0.024972  | 0         |
> > > | BE ViT (M=2) 1    | 0.69329  | 0.81316  | 0.0252599 | 0         |
> > >
> > > | model_type        |   im/nll |   im/acc |    im/ece |     im/kl |
> > > |:------------------|---------:|---------:|----------:|----------:|
> > > | BE ViT (M=4) average | 0.675471 | 0.815995 | 0.0174591 | 0.0349797 |
> > > | BE ViT (M=4) 0    | 0.689566 | 0.813335 | 0.0224964 | 0         |
> > > | BE ViT (M=4) 1    | 0.68997  | 0.813007 | 0.0228812 | 0         |
> > > | BE ViT (M=4) 2    | 0.690451 | 0.813327 | 0.022758  | 0         |
> > > | BE ViT (M=4) 3    | 0.69148  | 0.812572 | 0.0228684 | 0         |
> > >
> > > | model_type                    |   im/nll |   im/acc |    im/ece |       im/kl |
> > > |:------------------------------|---------:|---------:|----------:|------------:|
> > > | MIMO V-MoE (K=2) (BR=1) (M=2) average | 0.636294 | 0.830342 | 0.0275978 | 0.000199199 |
> > > | MIMO V-MoE (K=2) (BR=1) (M=2) 0    | 0.636307 | 0.830325 | 0.0276429 | 0           |
> > > | MIMO V-MoE (K=2) (BR=1) (M=2) 1    | 0.636396 | 0.83038  | 0.0276482 | 0           |
> > >
> > > | model_type        |   im/nll |   im/acc |    im/ece |     im/kl |
> > > |:------------------|---------:|---------:|----------:|----------:|
> > > | MIMO V-MoE (K=2) (BR=1) (M=4) average | 0.67164  | 0.822835 | 0.0368673 | 0.00149287  |
> > > | MIMO V-MoE (K=2) (BR=1) (M=4) 0    | 0.672062 | 0.822618 | 0.037052  | 0           |
> > > | MIMO V-MoE (K=2) (BR=1) (M=4) 1    | 0.672255 | 0.822905 | 0.0368738 | 0           |
> > > | MIMO V-MoE (K=2) (BR=1) (M=4) 2    | 0.672451 | 0.822818 | 0.0371108 | 0           |
> > > | MIMO V-MoE (K=2) (BR=1) (M=4) 3    | 0.672655 | 0.822682 | 0.0371601 | 0           |
> > >
> > > | model_type                  |   im/nll |   im/acc |    im/ece |    im/kl |
> > > |:----------------------------|---------:|---------:|----------:|---------:|
> > > | pBE (K=1) (M=2) average | 0.622445 | 0.833035 | 0.0181896 | 0.21728  |
> > > | pBE (K=1) (M=2) 0    | 0.671431 | 0.822595 | 0.0380803 | 0        |
> > > | pBE (K=1) (M=2) 1    | 0.682661 | 0.820628 | 0.0381708 | 0        |
> > >
> > > | model_type        |   im/nll |   im/acc |    im/ece |     im/kl |
> > > |:------------------|---------:|---------:|----------:|----------:|
> > > | pBE (K=1) (M=4) average | 0.623535 | 0.830105 | 0.0129002 | 0.163973 |
> > > | pBE (K=1) (M=4) 0    | 0.676705 | 0.819635 | 0.0336955 | 0        |
> > > | pBE (K=1) (M=4) 1    | 0.685498 | 0.817662 | 0.0336966 | 0        |
> > > | pBE (K=1) (M=4) 2    | 0.691262 | 0.81648  | 0.0345467 | 0        |
> > > | pBE (K=1) (M=4) 3    | 0.696657 | 0.815342 | 0.0348871 | 0        |
> > >
> > > | model_type           |   im/nll |   im/acc |    im/ece |    im/kl |
> > > |:---------------------|---------:|---------:|----------:|---------:|
> > > | Upstream V-MoE ens (K=1) (M=2) average | 0.587998 | 0.842576 | 0.01704   | 0.214004 |
> > > | Upstream V-MoE ens (K=1) (M=2) 0    | 0.640183 | 0.831828 | 0.0301545 | 0        |
> > > | Upstream V-MoE ens (K=1) (M=2) 1    | 0.644538 | 0.830252 | 0.0291547 | 0        |
> > >
> > > | model_type        |   im/nll |   im/acc |    im/ece |     im/kl |
> > > |:------------------|---------:|---------:|----------:|----------:|
> > > | Upstream V-MoE ens (K=1) (M=4) average | 0.560527 | 0.849012 | 0.0196817 | 0.214079 |
> > > | Upstream V-MoE ens (K=1) (M=4) 0    | 0.639201 | 0.832012 | 0.0297799 | 0        |
> > > | Upstream V-MoE ens (K=1) (M=4) 1    | 0.640523 | 0.83158  | 0.0302939 | 0        |
> > > | Upstream V-MoE ens (K=1) (M=4) 2    | 0.642221 | 0.831    | 0.0307415 | 0        |
> > > | Upstream V-MoE ens (K=1) (M=4) 3    | 0.646967 | 0.829532 | 0.0275019 | 0        |
> > >
> > > | model_type            |   im/nll |   im/acc |    im/ece |     im/kl |
> > > |:----------------------|---------:|---------:|----------:|----------:|
> > > | Downstream V-MoE ens (K=1) (M=2) average | 0.62018  | 0.835616 | 0.0230324 | 0.0734039 |
> > > | Downstream V-MoE ens (K=1) (M=2) 0    | 0.641565 | 0.831728 | 0.0303637 | 0         |
> > > | Downstream V-MoE ens (K=1) (M=2) 1    | 0.642633 | 0.83156  | 0.0301148 | 0         |
> > >
> > > | model_type        |   im/nll |   im/acc |    im/ece |     im/kl |
> > > |:------------------|---------:|---------:|----------:|----------:|
> > > | Downstream V-MoE ens (K=1) (M=4) average | 0.607057 | 0.83826  | 0.020543  | 0.07345   |
> > > | Downstream V-MoE ens (K=1) (M=4) 0    | 0.641304 | 0.831804 | 0.0298523 | 0         |
> > > | Downstream V-MoE ens (K=1) (M=4) 1    | 0.642375 | 0.831532 | 0.0301925 | 0         |
> > > | Downstream V-MoE ens (K=1) (M=4) 2    | 0.642767 | 0.831112 | 0.0306736 | 0         |
> > > | Downstream V-MoE ens (K=1) (M=4) 3    | 0.643479 | 0.83071  | 0.0307099 | 0         |

---

> > ### Author Response · Authors · 2021-11-25
> > **Expert partitioning: The mechanism pushing the KL up**
> >
> > In order to test our hypothesis in the comment "Additional investigation into ensemble diversity and individual model performance"—ensemble members with larger sets of non-shared parameters leads to high diversity—we have run an additional ablation experiment.
> >
> > In this experiment, we allow the partitions of the experts in pBE to have *some degree of overlap*. That is, the ensemble members can now share *some* experts. For example, with 32 total experts and an ensemble size of M=2, an overlap of 8 shared experts means that each expert has (32 - 8)/2 = 12 unique experts, and 12 + 8 = 20 experts in total. The results for an overlap of 0 experts (standard pBE) and for full overlap (“tiling without partitioning”) can be found in our Table 3.
> >
> > We see in the following table that as we increase the number of shared experts, we are directly decreasing the ensemble diversity and improving the NLL, ACC, and ECE. We also observe the same trends for K=1 (rather than K=2) experts.
> >
> > *We believe that this ablation study directly addresses the reviewer’s request for “further investigation into what enables the proposed method to attain diversity (aka superior performance)” and to ”investigate what mechanism is pushing the KL up”.*
> >
> > | model_type                      |   im/nll |   im/acc |    im/ece |     im/kl |
> > |:--------------------------------|---------:|---------:|----------:|----------:|
> > | pBE (M=2) (K=2) shared experts = 2  | 0.617 | 0.8345 | 0.016 | 0.167  |
> > | pBE (M=2) (K=2) shared experts = 4   | 0.622 | 0.8338 | 0.017 | 0.148  |
> > | pBE (M=2) (K=2) shared experts = 8   | 0.627 | 0.8333 | 0.021 | 0.122  |
> > | pBE (M=2) (K=2) shared experts = 16 | 0.639 | 0.8318 | 0.030  | 0.077 |

---

> > ### Author Response · Authors · 2021-11-25
> > **Diversity alone is not sufficient**
> >
> > While we do think that our investigations below into ensemble diversity and individual member performance are interesting, we would also like to discuss the fact that ensemble member diversity is not the only important factor in final generalization performance. Specifically, we note that adding more diversity for diversity's sake does not result in a better ensemble. For example, figure 6 of [Deep Ensembles: a Loss Landscape Perspective](https://arxiv.org/pdf/1912.02757.pdf) shows a diversity-accuracy tradeoff. Furthermore, Theorem 4 of [Learning under Model Misspecification: Applications to Variational and Ensemble methods](https://arxiv.org/pdf/1912.08335.pdf) tells us that in an optimal ensemble requires a trade-off between fitting the data well, and maintaining high diversity. [When does Diversity Help Generalization in Classification Ensembles?](https://arxiv.org/abs/1910.13631) also shows that ensemble diversity alone is not always helpful.

---

> > ### Author Response · Authors · 2021-11-25
> > **Replies to other points**
> >
> >  We would also like to respond to two comments:
> >
> > > The load balancing loss of (Riquelme et al. 2021) seeks to balance the number of example-to-expert assignments. While example-to-expert assignment is relevant to diversity, the load balancing loss used was not designed to pursue diversity. The authors may easily find several lines of work on achieving diversity in ensembles, e.g., through losses specifically designed to achieve diversity…
> >
> > We agree with the reviewer that the load-balancing loss was not designed to induce diversity. However, we believe it is an orthogonal research direction to further consider a diversity-inducing loss (e.g., [Mariet et al., 2020](https://ml4molecules.github.io/papers2020/ML4Molecules_2020_paper_23.pdf) in the context of BE) that could be applied to any method. Here, we show that all things being otherwise equal, our architecture, without any additional diversity loss, induces more diversity.
> >
> > > tiling w/o partition is like duplicating one model (all outputs will be similar => low diversity)
> >
> > This is not accurate, the experts, via noisy input-dependent assignments, make it possible to have different predictions for each tiled input. It was not straightforward to foresee that the noise in the routing was not sufficient to induce diversity (see our Table 3).

---

### Decision · Program_Chairs · 2022-01-20

**Decision:**

Reject

**Comment:**

This paper experiments with a combination of Sparse MoEs and Ensembles on the Vision Transformer (ViT), showing improved performance. To efficiently combine Sparse MoEs and Ensembles, the paper presents Partitioned Batch Ensembles (PBE), where the parameters of the self-attention layers are shared, and an ensemble of Sparse MOEs are used for the MLP layers of the Transformer blocks.

While reviewers agree that the proposed approach is interesting, they also point out several weaknesses, such as the limited novelty of the proposed method (a simple combination of existing techniques) and small experimental gains. They also pointed out several weakness related to the experimental part. While the authors responded in a very detailed manner to several of these points and presented several additional experiments, I feel this paper will benefit from consolidating all these new results and going through another round of reviews.